## 1 Anhydrite Dissolution Dynamics as a Hydrogeochemical

# **Tracer of Seismic-Fluid Coupling: Insights from the East**

## 3 Anatolian Fault Zone, Turkey

- Zebin Luo<sup>1</sup>, Xiaocheng Zhou<sup>2,3</sup>, Yueren Xu<sup>2</sup>, Peng Liang<sup>2</sup>, Huiping Zhang<sup>4</sup>, Jinlong
- Liang<sup>5</sup>, Zhaojun Zeng<sup>2</sup>, Yucong Yan<sup>3</sup>, Zheng Gong<sup>6</sup>, Shiguang Wang<sup>6</sup>, Chuanyou Li<sup>4</sup>,
- Zhikun Ren<sup>4</sup>, Jingxing Yu<sup>4</sup>, Zifa Ma<sup>4</sup>, Junjie Li<sup>4</sup>
- <sup>1</sup>School of Emergency Management, Xihua University, Chengdu 610039, China
- <sup>2</sup>United Laboratory of High-Pressure Physics and Earthquake Science, institute of earthquake forecasting,
- CEA, Beijing 100036, China
- <sup>3</sup>School of Earth Sciences and Resources, China University of Geosciences, Beijing 100083, China
- <sup>4</sup>Institute of Geology, China Earthquake Administration, Beijing, 100081, China
- <sup>5</sup>College of Earth and Planetary Sciences, Chengdu University of Technology, Chengdu 610059, China
- 13 <sup>6</sup> Institute of Geophysics, China Earthquake Administration, Beijing, 100081, China
- 14 Correspondence to: Xiaocheng Zhou (<u>zhouxiaocheng188@163.com</u>).
- 15 Abstract: Pre-seismic turbidity and salinity anomalies in groundwater were documented at HS04 and 16 HS14 monitoring wells along the East Anatolian Fault Zone (EAFZ) following the 2023 Mw 7.8 and Mw 7.6 Turkey earthquakes. By synthesizing hydrogeochemical datasets (2013-2023) with post-seismic 17 18 responses, we unravel fault-segmented groundwater evolution: (1) Northern Na-Cl and Na-HCO<sub>3</sub> type waters result from mixing of mantle-derived magmatic fluids (0-7% contribution) with shallow 19 20 groundwater, governed by volcanic rocks-carbonate dissolution; (2) Central-southern Ca-HCO3 and Ca-Na-HCO<sub>3</sub> systems reflect shallow circulation with localized inputs from evaporites (Increased SO<sub>4</sub><sup>2-</sup> 21 concentration caused by dissolution of anhydrite), ophiolites (Mg<sup>2+</sup> anomalies), and seawater. 22 PHREEQC simulation shows that the dissolve-precipitation equilibrium of anhydrite is sensitive to the 23 24 variation of water-rock reaction intensity in the Central-southern segments of EAFZ. Coseismic 25 permeability changes disrupt the solubility equilibria of anhydrite, driving hydrochemical anomalies. We 26 propose that seismic stress redistribution induces fracture network reorganization, thereby disrupting anhydrite solubility equilibria. Given its tectonic sensitivity and widespread occurrence, anhydrite 27 dissolution dynamics emerge as a potential tracer for hydrogeochemical monitoring in active fault zones. 28 29 Key words: Groundwater; Water-rock interaction; Seismic activity; PHREEQC; Anhydrite; East 30 Anatolian Fault Zone.

#### 1 Introduction

Active fault zones perturb subsurface hydrogeochemical equilibrium through dynamic rock-water interactions, generating diagnostic anomalies in groundwater chemistry that may serve as potential seismic precursors (Franchini et al., 2021; Ingebritsen and Manga, 2014; King et al., 2006; Luo et al., 2023; Poitrasson et al., 1999; Skelton et al., 2014; Tsunogai and Wakita, 1995; Wang et al., 2021). However, the diagnostic reliability of such hydrochemical signatures faces challenges. Climatic factors (e.g., precipitation variability and temperature fluctuations) can mask tectonic signals by altering waterrock reaction kinetics (Okan et al., 2018), while regional heterogeneity in lithology, fracture density, and hydrological circulation depth introduces substantial spatial variability in groundwater (Luo et al., 2023). This study investigates the hydrogeochemical characteristics of the seismically active East Anatolian Fault Zone (EAFZ) in eastern Turkey through a comprehensive 13-year observational dataset (2013-2023). By systematically analyzing groundwater circulation patterns and water-rock interaction processes along the fault system, we integrate post-seismic hydrochemical monitoring following the February 2023 Mw 7.8 and 7.6 earthquake sequence to delineate the relationship between hydrogeochemical anomalies and fault activity. Our findings aim to establish the relationship between groundwater anomalies and fault zone activities, thereby advancing methodologies for groundwaterbased seismic monitoring in active fault zone systems. The EAFZ, a ~500 km NE-SW trending left-lateral strike-slip system accommodating ~11 mm/yr of Anatolian-Arabian plate motion with reverse thrust components (Pousse - Beltran et al., 2020), has generated destructive seismic events throughout recorded history (Hubert-Ferrari et al., 2020; Simão et al., 2016; Sparacino et al., 2022; Tan et al., 2008). The 2023 twin earthquakes exemplify its capacity for massive stress release (Kwiatek et al., 2023; Ma et al., 2024; Wang et al., 2023b), producing coseismic surface ruptures exceeding 280 km with maximum slip of 7.2±0.72 m (Liang et al., 2024). Notably, marked hydrochemical anomalies (e.g., white water, turbidity and intermittent groundwater gushing) were detected at monitoring wells HS04 and HS14 both before and after the earthquake (Video 1 and 2), indicating fault-controlled fluid responses to seismic stress perturbations. Previous studies have identified three primary fluid sources within the EAFZ system: 1) mantle-derived magmatic fluids (Aydin et al., 2020; Italiano et al., 2013; Karaoğlu et al., 2019), 2) deeply circulated metamorphic waters (Yuce et al., 2014), and 3) Mediterranean seawater intrusion at its southern terminus

(Yuce et al., 2014). These studies provide sufficient data support for accurate understanding of EAFZ groundwater circulation. In this contribution, the EAFZ groundwater observation data over the past 13 years are compared with the groundwater chemical composition after the double earthquakes in 2023 to tracing the origin of geothermal fluid, restore the water-rock interaction process, and evaluate the influence of seismic activity on the geothermal fluid circulation process. This work provides new constraints on tectonic controls of deep fluid migration in active fault zone systems while advancing the application of hydrogeochemical monitoring in seismic hazard assessment.

Fig. 1. a: A brief Map of the eastern Mediterranean region from NASADEM (https://doi.org/10.5069/G93T9FD9). b: Geological map of EAFZ, modified from (van Hinsbergen et al., 2024). EF: Ecemiş Fault, SF: Sürgü Fault, MOF: Malatya-Ovacık Fault, GF: Göksün Fault, YGF: Yeşilgöz-Göksün Fault.

#### 2 Geologic background

72

Located at the intersection of Eurasia, Africa and Arabia, Turkey has a complex tectonic background 74 (Lanari et al., 2023; Simão et al., 2016). Here, the collision between the Arabian and Eurasian plates was 75 an important tectonic process that began in the early Miocene (~ 23 Ma) and continues to the this day 76 (van Hinsbergen et al., 2024). This collision caused plateau uplift, volcanic eruptions, sedimentary basin 77 formation, and large-scale strike-slip faults in eastern Turkey, including the EAFZ (Fig. 1) (Bilim et al., 78 2018; Karaoğlu et al., 2018; Karaoğlu et al., 2020; Whitney et al., 2023; Yönlü et al., 2017; Zhou et al., 79 2024). 80 The formation of the EAFZ is related to the northward subduction of a strong and thin lithospheric wedge 81 under the Arabian Plate (Nalbant et al., 2002; Sparacino et al., 2022). This subduction process led to the 82 formation of a stress concentration zone that eventually developed into a strike-slip fault that penetrated 83 the entire lithosphere, i.e. the EAFZ (Nalbant et al., 2002). In addition, because the African plate and the 84 Arabian plate are still moving northward, this fault zone is also accompanied by a certain thrust process, 85 which causes huge stresses at the plate margin (Ma et al., 2024; Över et al., 2023; Özkan et al., 2023; 86 Pousse - Beltran et al., 2020; Wang et al., 2023b; Whitney et al., 2023). 87 The stratigraphic composition of the East Anatolian fault zone is complex, including Nonmetamorphosed Tauride nappes and Metamorphosed Tauride nappes crystallization base, Cretaceous 88 89 ophiolites and Cretaceous-Paleogene plutons. It is overlaid by clastic deposits, lacustrine deposits (such 90 as: Ancient Amik Lake) and volcanic cover of Upper Eocene-Oligocene to Plio-Quaternay. Faults are 91 widely developed in study area, including East Anatolian Fault, Ecemis Fault, Sürgü Fault, Malatya-92 Ovacık Fault, Göksün Fault, Yeşilgöz-Göksün Fault etc. (van Hinsbergen et al., 2024). These faults has 93 been active for a long time and has a history of devastating earthquakes, including two in February 2023 94 (Mw 7.8 and Mw 7.6) (Fig. 1) (Carena et al., 2023; Kwiatek et al., 2023; Ma et al., 2024; Maden and Özt ürk, 2015; Över et al., 2023; Özkan et al., 2023; Pousse - Beltran et al., 2020; Tan et al., 2008; Wang et 95 96 al., 2023b). 97 The climate of the EAFZ is mainly a temperate continental climate with cold winters and hot and dry 98 summers. The average annual rainfall is between 200 mm and 600 mm, and is mainly winter rain. Due 99 to its inland location and low rainfall, the flow of the river is relatively small. The groundwater system 100 is relatively complex, and geothermal resources are mainly distributed near the fault zone and its controlled areas, including low or moderate temperature geothermal systems, which have great potential for development and utilization (Aydin et al., 2020; Güleç and Hilton, 2016; Inguaggiato et al., 2016; Karaoğlu et al., 2019).

#### 3 Sampling and analytical methods

16 samples of groundwater were collected in EAFZ, including hot springs, geothermal wells and river water. HS01-HS04 was collected from west to east along SF. HS07-HS16 was collected from north to south along EAFZ (Fig. 1). Detailed sample collection and testing methods can be found at Luo et al. (2023). In short, the water sample was taken with a 50 mL clean polyethylene bottle and the temperature and pH of the water were measured and recorded. Two samples were collected at each sampling site, one was added with ultrapure HNO<sub>3</sub> to analyse the cation content, and the other was used to analyse the anion content and isotopic composition. All samples need to be pre-treated with a 0.45 µm filter membrane to remove impurities before sampling. The Hydrogen and oxygen isotopes were determined by a Picarro L2140-I Liquid water and vapor isotope analyzer (relative to Vienna Standard Mean Ocean Water (V - SMOW)). Precisions on the measured  $\delta^{18}O$  and  $\delta D$  value was  $\pm 0.2\%$  (2SD) and  $\pm 1\%$  (2SD) respectively (Zeng et al., 2025). The cation (Li<sup>+</sup>, Na<sup>+</sup>, K<sup>+</sup>, Ca<sup>2+</sup> and Mg<sup>2+</sup>) and anion (F<sup>-</sup>, Cl<sup>-</sup>, NO<sub>3</sub><sup>-</sup> and SO<sub>4</sub><sup>2-</sup>) were analysed by Dionex ICS-900 ion chromatograph (Thermo Fisher Scientific Inc.) at the Earthquake Forecasting Key Laboratory of China Earthquake Administration, with the reproducibility within ±2% and detection limits 0.01 mg/L (Chen et al., 2015). HCO<sub>3</sub><sup>-</sup> and CO<sub>3</sub><sup>2-</sup> was determined by acid-base titration with a ZDJ-100 potentiometric titrator (reproducibility within ±2%). SiO<sub>2</sub> were analysed by inductively coupled plasma emission spectrometer Optima-5300 DV (PerkinElmer Inc.) (Li et al. 2021). Trace elements were analysed by Element XR ICP-MS at the Test Center of the Research Institute of Uranium Geology. Multielement standard solutions (IV-ICPMS 71A, IV-ICP-MS 71B and IV-ICP-MS 71D, iNORGANIC VENTURES) used for quality control. The analytical error margin of major cations and trace elements were less than 10%. Strontium isotope ratios (87Sr/86Sr) were determined through triple quadrupole ICP-MS (Agilent 8900 ICP-QQQ) with a precision of  $\pm 0.001$  (Liu et al., 2020).

#### 4 Results

Physical, chemical and isotopic compositions of groundwaters are listed in Table 1. The pH of the water samples varied from 7.03 to 11.72, and all the samples showed weakly alkaline characteristics except HS15 (pH=11.72). The effluent temperature of water sample is low (8.1–32.0°C), and the highest temperature is HS15 sample (32.0°C). HS08 is a river sample with the lowest temperature (8.1°C). SiO<sub>2</sub> varies from 0.38 mg/L to 84.64mg/L. HCO<sub>3</sub><sup>-</sup> (165.72–1854.30 mg/L) is the main anion. The concentration of SO<sub>4</sub><sup>2-</sup> range from 1.21 mg/L to 316.61 mg/L, and the concentration of SO<sub>4</sub><sup>2-</sup> in some samples is relatively high (e.g. HS01 (287.74 mg/L), HS03 (103.56 mg/L), HS04 (229.75 mg/L), HS14 (316.61 mg/L)). The concentration of Na<sup>+</sup> (0.42–88.93 mg/L), Cl<sup>-</sup> (0.97–75.92 mg/L) and B (3.62–1047.25 µg/L) varied synergistically. Ca<sup>2+</sup> (14.16–501.58 mg/L) is the main cation, followed by Mg<sup>2+</sup> (0.38–116.20 mg/L). The types of groundwater include Na-Cl-HCO<sub>3</sub>, Ca-HCO<sub>3</sub>, Ca-HCO<sub>3</sub>-SO<sub>4</sub> and Mg-HCO<sub>3</sub> (Fig. 2). The  $\delta$ <sup>18</sup>O and  $\delta$ D of samples varied from –11.30% to –6.55% and –65.43% to –34.43% respectively, which is near to the global meteoric water line (GMWL) (Craig, 1961) (Fig. 3), suggesting their meteoric water origin. The <sup>87</sup>Sr/<sup>86</sup>Sr varied from 0.7053 to 0.7135, showing the characteristics of multi-source region mixing.

Fig. 2. Piper plot of sampled groundwaters in EAFZ. The groundwaters are Na-Cl-HCO<sub>3</sub>, Ca-HCO<sub>3</sub>, Ca-HCO<sub>3</sub>-SO<sub>4</sub> and Mg-HCO<sub>3</sub> types. Literature data source (see Table S1 for details): (Aydin et al., 2020; Baba et al., 2019; Karaoğlu et al., 2019; Okan et al., 2018; Pasvanoglu, 2020; YASİN and YÜCE, 2023; Yuce et al., 2014)

The composition of trace elements in groundwaters are shown in Table 2. The contents of Sr (30.13–3244.88  $\mu$ g/L) and Ba (1.89–196.48  $\mu$ g/L) in the samples varied widely. Moreover, Sr and SO<sub>4</sub><sup>2-</sup> had obvious positive correlation. Box plot analysis showed that the Fluid-Mobile Element (FME) concentrations of B (3.62–1047.25  $\mu$ g/L), Li (0.33–89.93  $\mu$ g/L) and Rb (0.14–28.91  $\mu$ g/L) in some samples were greater than the median (Fig. S1). Enrichment coefficients (EF) normalized by Ti is used for groundwaters and rocks. The result shows that whether compared with schist, basalt or Andesite of EAFZ, trace elements in groundwaters are all in a state of enrichment, and some elements can even be enriched 100000 times (Fig. S2).

Fig. 3.  $\delta D$  and  $\delta^{18}O$  (%V-SMOW) values for groundwaters collected from EAFZ. The GMWL represents the global meteoric water line (Craig, 1961). The LMWL represents the Local meteoric water line (Aydin et al., 2020). The magmatic fluid distribution ( $\delta D = -20 \pm 10\%$ ,  $\delta^{18}O = 10 \pm 2\%$ ) from (Giggenbach, 1992). Literature data source is consistent with Fig. 2.

Table 1.Physical, Chemistry and isotopic compositions of groundwaters from the EAFZ. 160

| (mg/L) (mg/L) (mg/L) - 27.93 4.85 - 0.13 48.44 0.48 - 7.66 0.39 - 7.66 0.39 - 4.19 0.32 - 1.13 1.13 1.13 1.57 1.57 1.57 1.57 1.57 1.57 1.57 1.57 1.57 1.57 1.57 1.57 1.57 1.57 1.57 1.57 1.57 1.57 1.57 1.57 1.57 1.57 1.57 1.57 1.57 1.57 1.57 1.57 1.57 1.57 1.57 1.57 1.57 1.57 1.57 1.57 1.57 1.57 1.57 1.57 1.57 1.57 1.57 1.57 1.57 1.57 1.57 1.57 1.57 1.57 1.57 1.57 1.57 1.57 1.57 1.57 1.57 1.57 1.57 1.57 1.57 1.57 1.57 1.57 1.57 1.57 1.57 1.57 1.57 1.57 1.57 1.57 1.57 1.57 1.57 1.57 1.57 1.57 1.57 1.57 1.57 1.57 1.57 1.57 1.57 1.57 1.57 1.57 1.57 1.57 1.57 1.57 1.57 1.57 1.57 1.57 1.57 1.57 1.57 1.57 1.57 1.57 1.57 1.57 1.57 1.57 1.57 1.57 1.57 1.57 1.57 1.57 1.57 1.57 1.57 1.57 1.57 1.57 1.57 1.57 1.57 1.57 1.57 1.57 1.57 1.57 1.57 1.57 1.57 1.57 1.57 1.57 1.57 1.57 1.57 1.57 1.57 1.57 1.57 1.57 1.57 1.57 1.57 1.57 1.57 1.57 1.57 1.57 1.57 1.57 1.57 1.57 1.57 1.57 1.57 1.57 1.57 1.57 1.57 1.57 1.57 1.57 1.57 1.57 1.57 1.57 1.57 1.57 1.57 1.57 1.57 1.57 1.57 1.57 1.57                                                                                                                                                                                                                                                                                                                                                                                            | SiO <sub>2</sub> Li <sup>+</sup> Na <sup>+</sup> | $K^{+}$ | Mg <sup>2+</sup> Ca | Ca <sup>2+</sup> F- | CI         | NO <sub>3</sub> - | $SO_4^{2-}$ | $HCO_3$ | $CO3^{2-}$ | δD  | $\delta^{18}$ O | 875,1865 | 75.0   |
|----------------------------------------------------------------------------------------------------------------------------------------------------------------------------------------------------------------------------------------------------------------------------------------------------------------------------------------------------------------------------------------------------------------------------------------------------------------------------------------------------------------------------------------------------------------------------------------------------------------------------------------------------------------------------------------------------------------------------------------------------------------------------------------------------------------------------------------------------------------------------------------------------------------------------------------------------------------------------------------------------------------------------------------------------------------------------------------------------------------------------------------------------------------------------------------------------------------------------------------------------------------------------------------------------------------------------------------------------------------------------------------------------------------|--------------------------------------------------|---------|---------------------|---------------------|------------|-------------------|-------------|---------|------------|-----|-----------------|----------|--------|
| 36.518113         38.003517         S         03.23/2023         15.8         8.12         1565         20.70         -         27.93           37.173212         38.028567         S         03/23/2023         13.2         8.35         287         5.27         -         0.42           37.166040         38.038567         S         03/23/2023         13.2         7.12         1876         26.36         0.13         48.44           37.1669088         37.809271         S         03/23/2023         12.7         8.50         634         14.42         -         0.42           37.669088         37.809271         S         03/23/2023         12.7         8.50         634         14.42         -         7.66           38.056844         37.949260         S         03/23/2023         12.7         8.46         276         9.41         -         0.84           38.058844         37.949260         S         03/23/2023         18.0         8.46         275         15.15         8.21         48.49         -         1.13         48.44           38.058844         37.949022         S         03/23/2023         8.1         8.43         275         15.15         8.21                                                                                                                                                                                                      | (mg/L)                                           |         | (mg/L) (mg          | (mg/L) (mg/L)       | L) (mg/L)  | .) (mg/L)         | (mg/L)      | (mg/L)  | (mg/L)     | (%) | (%)             | 01/10    | Z2Z    |
| 37.173212         38.028567         S         03/23/2023         13.2         8.35         287         5.27         -         0.42           37.166040         38.031327         S         03/23/2023         13.2         7.12         1876         26.36         0.13         48.44           37.174886         38.033718         S         03/23/2023         15.0         7.03         2683         84.64         0.05         19.90           37.510811         37.609088         37.809271         S         03/23/2023         15.0         8.27         774         15.34         -         48.44           38.056844         37.90526         S         03/23/2023         15.0         8.27         774         15.34         -         4.19           38.051818         37.942560         S         03/23/2023         8.1         8.45         275         15.15         -         4.19           36.994384         37.940742         S         03/23/2023         16.3         8.24         5.5         5.5         1.13           36.54302         36.54302         S         03/23/2023         16.3         8.24         46.50         -         1.57           36.439440         36.672020 <td>1</td> <td></td> <td>75.69 253</td> <td>253.85 3.</td> <td>3.60 55.46</td> <td>- 9</td> <td>- 287.74</td> <td>670.01</td> <td>-</td> <td>1</td> <td>-9.81</td> <td>0.7065</td> <td>0.0001</td> | 1                                                |         | 75.69 253           | 253.85 3.           | 3.60 55.46 | - 9               | - 287.74    | 670.01  | -          | 1   | -9.81           | 0.7065   | 0.0001 |
| 37.166040         38.031327         S. 03/23/2023         13.2         7.12         1876         26.36         0.13         48.44           37.174886         38.033718         S. 03/23/2023         15.0         7.03         2683         84.64         0.05         19.90           37.174886         38.033718         S. 03/23/2023         15.0         7.03         2683         84.64         0.05         19.90           37.510811         37.700516         S. 03/23/2023         15.0         8.27         774         15.34         -         7.66           38.058844         37.349742         S. 03/23/2023         18.0         8.43         275         15.15         -         1.13           36.89339         37.340742         S. 03/23/2023         18.0         8.11         699         25.50         0.01         5.85           36.534302         36.892454         S. 03/23/2023         16.3         8.27         517         9.69         -         1.57           36.531328         36.531328         S. 03/23/2023         16.3         8.22         579         10.05         0.01         4.87           36.439440         36.63634         W         03/23/2023         18.2         8.21                                                                                                                                                                                   |                                                  | 2:      | 6.58 54             | 54.04 0.            | 0.40 1.33  | 3 5.06            | 5 6.37      | 178.53  | 1          | •   | •               | 0.7120   | 0.0003 |
| 37.174886         38.033718         5         03/23/2023         15.0         7.03         2683         84.64         0.05         19.90           37.669088         37.809271         S         03/23/2023         12.7         8.50         634         14.42         -         7.66           37.510811         37.942560         S         03/23/2023         15.0         8.27         774         15.34         -         4.19           38.056844         37.942560         S         03/23/2023         8.1         8.45         276         9.41         -         4.19           38.056844         37.942560         S         03/23/2023         8.1         8.45         275         15.15         -         0.84           38.058379         37.349742         S         03/23/2023         18.0         8.48         659         25.50         0.01         5.85           36.594384         37.460028         S         03/23/2023         16.3         8.27         517         9.69         -         1.57           36.51328         36.51328         S         03/23/2023         16.3         8.27         519         10.55         0.01         4.87           36.439440 <td< td=""><td>0.13</td><td></td><td>74.20 368</td><td>368.42 0.</td><td>0.50 30.85</td><td>5 30.13</td><td>3 103.56</td><td>1271.1</td><td>•</td><td>•</td><td>-9.33</td><td>0.7079</td><td>0.0002</td></td<>  | 0.13                                             |         | 74.20 368           | 368.42 0.           | 0.50 30.85 | 5 30.13           | 3 103.56    | 1271.1  | •          | •   | -9.33           | 0.7079   | 0.0002 |
| 37.669088         37.809271         S         03/23/2023         12.7         8.50         634         14.42         -         7.66           37.510811         37.700516         S         03/23/2023         15.0         8.27         774         15.34         -         4.19           38.056844         37.942560         S         03/23/2023         8.1         8.46         276         9.41         -         0.84           38.051818         37.939222         R         03/23/2023         18.0         8.11         699         25.50         0.01         5.85           36.894384         37.460028         S         03/23/2023         16.3         8.48         659         25.50         0.01         5.85           36.54302         36.892454         S         03/23/2023         16.3         8.27         517         9.69         -         1.57           36.54302         36.543040         S         03/23/2023         16.3         8.22         579         10.05         0.01         4.87           36.439440         36.672020         S         03/23/2023         18.2         8.21         1305         0.05         4.864           36.13823         S                                                                                                                                                                                                                  | 0.05                                             |         | 116.20 501          | 501.58 3.           | 3.70 9.29  | 9 3.33            | 3 229.75    | 1854.3  | •          | '   | -9.64           | 0.7132   | 0.0008 |
| 38.056844         37.700516         S         03/23/2023         15.0         8.27         774         15.34         -         4.19           38.056844         37.942560         S         03/23/2023         9.8         8.46         276         9.41         -         0.84           38.051818         37.942560         S         03/23/2023         8.1         8.43         275         15.15         -         0.84           36.808379         37.349742         S         03/23/2023         18.0         8.11         699         25.50         0.01         5.85           36.994384         37.460028         S         03/23/2023         16.3         8.27         517         9.69         -         1.57           36.51328         36.811041         S         03/23/2023         16.3         8.27         517         9.69         -         2.32           36.439440         36.672020         S         03/23/2023         18.2         8.22         579         10.05         0.01         4.87           36.13823         S         03.23/2023         23.5         8.21         1305         6.24         0.09         62.40           36.143759         36.383335         S                                                                                                                                                                                                         |                                                  |         | 25.88 103           | 103.61 0.           | 0.53 4.43  | 3 12.92           | 29.75       | 367.72  | •          | •   | -7.79           | 0.7091   | 0.0003 |
| 38.056844         37.942560         8         03/23/2023         9.8         8.46         276         9.41         -         0.84           38.051818         37.939222         R         03/23/2023         8.1         8.43         275         15.15         -         1.13           36.808379         37.349742         S         03/23/2023         18.0         8.41         699         25.50         0.01         5.85           36.994384         37.460028         S         03/23/2023         16.3         8.27         517         9.69         -         1.57           36.51328         36.811041         S         03/23/2023         16.9         8.32         489         46.50         -         2.32           36.439440         36.672020         S         03/23/2023         18.2         8.22         579         10.05         0.01         4.87           36.433823         36.383335         S         03/23/2023         23.5         8.21         1305         36.4         0.09         62.40           36.14759         36.23220         S         03/23/2023         24.5         8.45         100         9.61         48.64                                                                                                                                                                                                                                                 | •                                                |         | 54.08 100           | 100.99 0.           | 0.43 5.98  | 8 1.61            | 1.96        | 515.66  | •          | •   | -8.11           | 0.7100   | 0.0002 |
| 36.808379         37.939222         R         03/23/2023         8.1         8.43         275         15.15         -         1.13           36.808379         37.349742         S         03/23/2023         18.0         8.11         699         25.50         0.01         5.85           36.994384         37.460028         S         03/23/2023         16.3         8.48         659         31.29         -         1.57           36.551328         36.892454         S         03/23/2023         16.3         8.27         517         9.69         -         1.57           36.439440         36.672020         S         03/23/2023         18.2         8.22         579         10.05         0.01         4.87           36.439440         36.503634         W         03/23/2023         23.5         8.21         1305         36.64         0.09         62.40           36.143759         36.383355         S         03/23/2023         24.5         845         1100         32.57         0.01         88.93                                                                                                                                                                                                                                                                                                                                                                           | •                                                | 4.      | 4.62 55             | 55.11 0.            | 0.41 0.97  | 7 2.74            | 5.00        | 167.86  | •          | •   | -8.93           | 0.7135   | 0.0006 |
| 36.808379         37.349742         S         03/23/2023         18.0         8.11         699         25.50         0.01         5.85           36.994384         37.460028         S         03/23/2023         16.3         8.48         659         31.29         -         1.57           36.51302         36.811041         S         03/23/2023         16.9         8.32         489         46.50         -         2.32           36.439440         36.672020         S         03/23/2023         18.2         8.22         579         10.05         0.01         4.87           36.439440         36.503634         W         03/23/2023         23.5         8.21         1305         36.64         0.09         62.40           36.143759         36.233335         S         03/23/2023         24.5         845         1100         32.57         0.01         88.93                                                                                                                                                                                                                                                                                                                                                                                                                                                                                                                        |                                                  | 3       | 4.47 55             | 55.34 0.            | 0.44 1.06  | 6 3.83            | 3 5.69      | 165.72  | •          | •   | -9.26           | 0.7104   | 0.0004 |
| 36.994384         37.460028         S         03/23/2023         20.0         8.48         659         31.29         -         1.57           36.554302         36.592454         S         03/23/2023         16.3         8.27         517         9.69         -         2.32           36.51328         36.51328         36.81041         S         03/23/2023         16.9         8.32         489         46.50         -         2.31           36.439440         36.672020         S         03/23/2023         18.2         8.22         579         10.05         0.01         4.87           36.133823         36.503634         W         03/23/2023         23.5         8.21         1305         36.64         0.09         62.40           36.147159         36.233335         S         03/23/2023         24.5         845         1100         32.57         0.01         88.93                                                                                                                                                                                                                                                                                                                                                                                                                                                                                                            | 0.01                                             |         | 42.60 94            | 94.99 0.            | 0.52 6.80  | 0 8.87            | 7 93.44     | 344.96  | 1          | •   | -6.81           | 0.7076   | 0.0002 |
| 36.554302         36.892454         S         03/23/2023         16.3         8.27         517         9.69         -         2.32           36.521328         36.811041         S         03/23/2023         16.9         8.32         489         46.50         -         2.31           36.439440         36.672020         S         03/23/2023         18.2         8.22         579         10.05         0.01         4.87           36.373823         36.53634         W         03/23/2023         23.5         8.21         1305         36.64         0.09         62.40           36.163672         36.383335         S         03/23/2023         24.5         845         1100         32.57         0.01         88.93                                                                                                                                                                                                                                                                                                                                                                                                                                                                                                                                                                                                                                                                          | 1                                                | - L:    | 90.13               | 18.22 0.            | 0.35 3.80  | 0 7.53            | 3 2.76      | 459.47  | 1          |     | -6.71           | 0.7119   | 0.0003 |
| 36.521328         36.811041         S         03/23/2023         16.9         8.32         489         46.50         -         2.11           36.439440         36.672020         S         03/23/2023         18.2         8.22         579         10.05         0.01         4.87           36.373823         36.373823         36.383335         S         03/23/2023         23.5         8.21         1305         36.64         0.09         62.40           36.163672         36.383335         S         03/23/2023         32.0         11.72         589         0.38         0.02         48.64           36.147159         36.27370         S         03/23/2023         24.5         845         1100         32.57         0.01         88.93                                                                                                                                                                                                                                                                                                                                                                                                                                                                                                                                                                                                                                                   | ı                                                |         | 27.89 75            | 75.25 0.            | 0.45 4.39  | 9 9.25            | 5 12.11     | 312.24  | 1          | •   | -7.58           | 0.7107   | 0.0004 |
| 36.439440     36.672020     S     03/23/2023     18.2     8.22     579     10.05     0.01     4.87       36.373823     36.503634     W     03/23/2023     23.5     8.21     1305     36.64     0.09     62.40       36.163672     36.383335     S     03/23/2023     32.0     11.72     589     0.38     0.02     48.64       36.147159     36.273720     S     03/23/2023     24.5     8.45     1100     32.57     0.01     88.93                                                                                                                                                                                                                                                                                                                                                                                                                                                                                                                                                                                                                                                                                                                                                                                                                                                                                                                                                                             | ,                                                |         | 60.76               | 14.16 0.            | 0.52 6.13  | 3 14.55           | 5 4.27      | 307.98  | 1          | •   | -6.55           | 0.7110   | 0.0006 |
| 36.373823 36.503634 W 03/23/2023 23.5 8.21 1305 36.64 0.09 62.40 36.163672 36.383335 S 03/23/2023 32.0 11.72 589 0.38 0.02 48.64 36.147159 36.273720 S 03/23/2023 24.5 8.45 1100 32.57 0.01 88.93                                                                                                                                                                                                                                                                                                                                                                                                                                                                                                                                                                                                                                                                                                                                                                                                                                                                                                                                                                                                                                                                                                                                                                                                              | 0.01                                             |         | 30.35 81            | 81.56 0.            | 0.50 7.67  | 7 8.67            | 7 39.89     | 309.40  | 1          | •   | -7.30           | 0.7080   | 0.0002 |
| 36.163672 36.383335 S 03/23/2023 32.0 11.72 589 0.38 0.02 48.64<br>36.147159 36.273720 S 03/23/2023 24.5 845 1100 32.57 0.01 88.93                                                                                                                                                                                                                                                                                                                                                                                                                                                                                                                                                                                                                                                                                                                                                                                                                                                                                                                                                                                                                                                                                                                                                                                                                                                                             | 0.09                                             |         | 65.12 151           | 151.43 4.           | 4.33 75.92 | 2 34.60           | 316.61      | 300.15  | 1          | •   | -7.51           | 0.7053   | 0.0001 |
| 36 147159 36 273720 S 03/23/2023 24.5 8.45 1100 32.57 0.01 88.93                                                                                                                                                                                                                                                                                                                                                                                                                                                                                                                                                                                                                                                                                                                                                                                                                                                                                                                                                                                                                                                                                                                                                                                                                                                                                                                                               | 0.02                                             |         | 0.38 55             | 55.55 0.            | 0.41 48.71 | 1 5.28            | 3 1.21      | 1       | 154.61     | •   | -8.37           | 0.7070   | 0.0007 |
|                                                                                                                                                                                                                                                                                                                                                                                                                                                                                                                                                                                                                                                                                                                                                                                                                                                                                                                                                                                                                                                                                                                                                                                                                                                                                                                                                                                                                | 32.57 0.01 88.9                                  | 18.68   | 59.60 73            | 73.35 0.            | 0.72 67.11 | 1 43.51           | 75.90       | 484.37  | •          | '   | -7.33           | 0.7073   | 0.0002 |

Note: "-" represents below detection limit or undetected. "S" is Hot spring, "W" is Well water, "R" is river water.

Table 2. Trace elements compositions of groundwaters from the EAFZ.

| 2      | В                                                                                                                         | Al          | Ь          | Sc         | ij         | >           | Mn          | Fe          | 3          | ïZ           | Ga         | Rb      | Sr      | ¥      | Zr     | Nb     | Ba     | Hf     | Та     | Pb     | Th     | n      |
|--------|---------------------------------------------------------------------------------------------------------------------------|-------------|------------|------------|------------|-------------|-------------|-------------|------------|--------------|------------|---------|---------|--------|--------|--------|--------|--------|--------|--------|--------|--------|
| NO     | (hg/L)                                                                                                                    | (ng/L)      | (hg/L)     | (µg/L)     | (hg/L)     | (hg/L)      | (hg/L)      | (hg/L)      | (hg/L)     | (hg/L)       | (µg/L)     | (hg/L)  | (hg/L)  | (hg/L) | (hg/L) | (hg/L) | (hg/L) | (hg/L) | (hg/L) | (hg/L) | (hg/L) | (hg/L) |
| HS01   | 35.49                                                                                                                     | 10.02       | 66.41      | 0.04       | 0.20       | 0.23        | 369.58      | 34.43       | 0.40       | 3.40         | 0.03       | 1.68    | 1231.40 | 0.04   | 0.19   | 0.02   | 77.45  | 0.004  | 0.01   | 0.19   | 0.001  | 3.15   |
| HS02   | 3.62                                                                                                                      | 8.26        | 8.94       | 0.02       | 0.22       | 0.85        | 0.73        | 21.10       | 0.01       | 0.16         | 0.04       | 0.25    | 69.66   | 0.01   |        | 0.02   | 16.12  |        | 0.01   | 0.14   |        | 0.43   |
| HS03   | 1047.25                                                                                                                   | 8.23        | 11.86      | 0.08       | 0.19       | 0.56        | 0.80        | 23.29       | 0.03       | 4.22         | 0.04       | 5.95    | 691.57  | 0.01   | 0.01   | 0.01   | 5.52   | 0.001  | 0.01   | 0.10   |        | 1.32   |
| HS04   | 512.31                                                                                                                    | 6.75        | 12.88      | 0.58       | 0.22       | 0.19        | 890.21      | 563.31      | 4.06       | 19.67        | 0.01       | 28.91   | 1505.17 | 0.12   | 99.0   | 0.02   | 11.28  | 0.004  | 0.01   | 0.13   | 0.003  | 0.23   |
| HS05   | 43.88                                                                                                                     | 88.9        | 9.14       | 0.04       | 0.17       | 2.23        | 0.90        | 16.14       | 0.04       | 0.88         | 0.04       | 0.31    | 667.55  | 0.02   | 0.03   | 0.01   | 196.48 | 0.001  | 0.01   | 0.17   |        | 1.64   |
| 90SH   | 18.60                                                                                                                     | 4.50        | 8.79       | 0.03       | 0.18       | 2.74        | 19.0        | 13.54       | 0.02       | 6.23         | 0.01       | 0.37    | 213.59  | 0.02   | 0.03   | 0.01   | 38.11  | 0.001  | 0.01   | 0.15   |        | 0.51   |
| HS07   | 8.32                                                                                                                      | 12.99       | 10.51      | 0.01       | 0.20       | 2.09        | 3.58        | 81.59       | 0.02       | 0.37         | 0.11       | 0.49    | 53.27   | 0.03   | 0.01   | 0.01   | 3.48   | ,      | 0.01   | 0.26   | 0.004  | 0.32   |
| HS08   | 4.77                                                                                                                      | 12.27       | 8.89       | 0.03       | 0.18       | 2.85        | 1.05        | 12.52       | 0.02       | 0.26         | 0.01       | 0.44    | 55.78   | 90.0   |        | 0.01   | 1.89   |        |        | 0.10   |        | 0.26   |
| HS09   | 24.05                                                                                                                     | 8.48        | 4.56       | 0.04       | 0.27       | 0.50        | 0.99        | 45.62       | 0.01       | 0.81         | 0.02       | 0.62    | 70.796  | 0.02   |        | 0.01   | 105.53 |        |        | 0.15   |        | 0.49   |
| HS10   | 14.56                                                                                                                     | 8.37        | 9.74       | 0.03       | 0.23       | 0.73        | 0.62        | 19.86       | 0.02       | 89.0         | 0.01       | 0.19    | 96.74   | 90.0   |        |        | 7.85   |        |        | 0.16   |        | 0.02   |
| HS11   | 9.13                                                                                                                      | 8.17        | 13.04      | 0.02       | 0.18       | 0.64        | 2.58        | 134.71      | 0.03       | 2.05         | 0.01       | 0.36    | 263.61  | 0.02   | 0.01   | 0.01   | 22.37  | 0.001  |        | 0.11   |        | 0.53   |
| HS12   | 7.37                                                                                                                      | 28.55       | 23.54      | 0.03       | 0.30       | 1.24        | 2.51        | 49.33       | 0.14       | 5.73         | 0.05       | 0.14    | 34.78   | 0.09   |        |        | 38.75  |        |        | 0.18   | 0.001  | 0.03   |
| HS13   | 14.94                                                                                                                     | 10.65       | 10.86      | 0.02       | 0.47       | 09.0        | 15.09       | 805.45      | 0.07       | 1.27         | 0.05       | 96.0    | 592.95  | 0.02   | 0.01   | 0.01   | 146.07 |        |        | 0.17   |        | 1.01   |
| HS14   | 183.76                                                                                                                    | 17.48       | 7.06       | 0.07       | 0.14       | 2.50        | 2.94        | 12.72       | 0.04       | 11.66        | 0.00       | 11.25   | 3244.88 | 0.02   | 0.02   | 0.01   | 95.96  | 0.001  | 0.01   | 0.10   | 0.001  | 0.34   |
| HS15   | 4.34                                                                                                                      | 5.41        | 6.85       | 0.03       | 0.19       | 0.03        | 69.0        | 14.15       | 0.01       | 0.32         | 0.00       | 1.86    | 30.13   | 0.01   |        |        | 2.36   |        |        | 0.15   |        | 0.01   |
| HS16   | 491.19                                                                                                                    | 6.67        | 812.91     | 0.03       | 0.29       | 7.20        | 68.0        | 34.78       | 0.10       | 10.68        | 0.00       | 2.23    | 738.82  | 0.02   | 0.02   |        | 39.83  | 0.001  | 0.01   | 0.20   | 0.002  | 5.08   |
| 163 No | Note: "-" represents below detection limit or undetected. Hf and Ta are kept to 3 decimal places due to their low content | resents bel | ow detecti | on limit o | r undetect | ted. Hf and | l Ta are ke | pt to 3 dec | imal place | es due to tl | heir low c | ontent. |         |        |        |        |        |        |        |        |        |        |

Note: "-" represents below detection limit or undetected. Hf and Ta are kept to 3 decimal places due to their low content.

#### 5 Discussion

5.1 The origin of groundwater in different segments of EAFZ

Previous studies have documented abundant geothermal resources within the EAFZ, which is characterized by low or moderate temperature geothermal systems (Aydin et al., 2020; Baba et al., 2019). Both aqueous and gaseous geochemical signatures indicate mixing between deep-sourced mantle/crustal fluids and shallow groundwater reservoirs (Aydin et al., 2020; Italiano et al., 2013; Yuce et al., 2014). Yuce et al. (2014) proposed that geothermal fluids at the southwest end of the EAFZ are triggered by deep-rooted regional faults, with localized seawater intrusion. Analogously, there are deep components involved in the geothermal fluid circulation in the middle to east section of EAFZ. However, the source of deep components are thought to be controlled by magmatic activity rather than from deep-rooted regional faults (Aydin et al., 2020; Italiano et al., 2013; Karaoğlu et al., 2019). At the intersection of the EAFZ and the North Anatolian Fault Zones (NAFZ), which is also known as the Karliova triple junction, there is extensive volcanic activity that may have provided energy and components for the geothermal fluid cycle eastern segment of the EAFZ (Bilim et al., 2018; Karaoğlu et al., 2018; Karaoğlu et al., 2020). Furthermore, Italiano et al. (2013) suggested these volcanic activities may even contribute to geothermal fluids in the middle segment of the EAFZ. These findings collectively suggest multiple tectonic controls (volcanism, fault activity, and seawater intrusion) on EAFZ's geothermal systems. The February 2023 earthquake sequence (Mw 7.8 and 7.6) ruptured the central EAFZ segment. A critical question arises: Are the observed pre-seismic groundwater anomalies seismogenically linked to this seismic event? To address this, we conducted comparative analyses of post-seismic hydrochemical data against a decadal-scale (13-year) pre-seismic groundwater dataset, as detailed below:

5.1.1 Hydrogen and oxygen isotope characteristics of groundwaters

Hydrogen and oxygen isotopes serve as robust geochemical tracers for elucidating the origin of geothermal fluids groundwater. As illustrated in Fig. 3, the  $\delta D$  and  $\delta^{18}O$  compositions of groundwater in the EAFZ align closely with the GMWL (Craig, 1961), indicating predominant atmospheric precipitation recharge. Notably, groundwater in the southern EAFZ proximal to the Mediterranean Sea exhibits progressively heavier isotopic signatures toward the coast, consistent with recharge sourced from evaporated Mediterranean seawater. In contrast, northern groundwater displays distinct  $\delta^{18}O$  enrichment

deviating from local meteoric trends, indicative of mixing with deep-sourced magmatic fluids—a interpretation corroborated by widespread Quaternary volcanic activity in the northern sector (Fig. 3) (Bilim et al., 2018; Karaoğlu et al., 2018; Karaoğlu et al., 2020). Conversely, central and southern groundwater samples exhibit isotopic signatures decoupled from magmatic inputs, reflecting the absence of active deep-seated magma reservoirs in these segments.

#### 5.1.2 Major ion characteristics of groundwaters

The groundwater chemistry exhibits distinct spatial heterogeneity across the EAFZ segments. Northern groundwaters are significantly enriched in Na+, K+, and Cl- (Na-Cl and Na-HCO<sub>3</sub> type), whereas central and southern segments display Ca-Mg-HCO3 type waters, with localized Ca-SO4 and Na-Cl anomalies (Fig. 2). These hydrochemical disparities likely reflect fundamentally distinct recharge sources and circulation pathways. As discussed earlier, magmatic fluid contributions are evident in northern groundwaters. Chloride serves as a key tracer for magmatic input (Luo et al., 2023; Pan et al., 2021). In the eastern EAFZ, Clconcentrations span 0.4-2500 mg/L, markedly higher than central/southern values. Given the segment's inland setting, seawater intrusion is negligible, suggesting Cl- enrichment primarily originates from magmatic fluids. Notably, Na<sup>+</sup>/Cl<sup>-</sup> molar ratios deviate from theoretical mixing trends, with Na<sup>+</sup> excesses implicating additional sodium sources (e.g., albite dissolution), to be detailed in Section 5.2. This interpretation aligns with petrological and geophysical evidence of active magmatism in the eastern EAFZ (Bilim et al., 2018; Karaoğlu et al., 2018; Karaoğlu et al., 2020; Maden and Öztürk, 2015; Oyan, 2018). Integrated H-O isotopic, major ion, and volcanic activity data collectively support a mixing model between meteoric water and magmatic fluids in the northern EAFZ. In contrast, central and southern groundwaters exhibit lower Na<sup>+</sup> and Cl<sup>-</sup> concentrations, with sporadic anomalies attributable to evaporite dissolution or limited seawater influence (Table 1). The Ca-Mg-HCO<sub>3</sub> dominance, coupled with isotopic signatures, reflects shallow circulation systems (<5 km depth) devoid of significant deep tectonic/magmatic inputs (Table S2). Ca<sup>2+</sup> likely derives from calcite, dolomite, or plagioclase weathering, while Mg<sup>2+</sup> sources include dolomite and serpentinite. Pre-seismic turbidity at HS14 (Video 1) may indicate earthquake-induced disruption of water-rock equilibria. However, the geothermal gases in the centre and south segment of EAFZ exhibit mantle-like  $\delta^{13}C_{CO_2}$ 

(-5.6% to -0.2%) and elevated  ${}^{3}\text{He}/{}^{4}\text{He}$  ratios (Rc/Ra = 0.44–4.41), contrasting with the absence of

deep fluid signatures in groundwater (Italiano et al., 2013). Actually, this decoupling results from fundamentally distinct migration mechanisms. Groundwater circulation operates as a shallow crustal system dominated by meteoric recharge, structurally confined by fault architecture. Conversely, geothermal gases predominantly represent deep-seated fluids, with their high mobility and low density enabling efficient ascent through fractures. This explains why mantle/crustal signals are preserved in gases but attenuated in aqueous phases.

To further constrain groundwater source area, we have calculated the thermal reservoir temperature of EAFZ groundwater, and the results are shown in Table S2. Due to the low water-rock interaction degree and diversity of rock types in this area, cations in water are difficult to reach water-rock equilibrium (Fig. 4). Hence, most of the cationic thermometer estimates are too large or too small, which can only be used as a reference for thermal reservoirs. Fortunately, SiO<sub>2</sub> thermometers are relatively suitable for estimating the reservoir temperature. As can be seen from Table S2, the reservoir temperatures range from 19.81°C to 128.09 °C (Quartz, no steam loss), which belongs to the low or moderate temperature geothermal systems. Using the circulation depth calculation formula, the maximum circulation depth is estimated to be 4.4km (HS04) (Table S2).

Fig. 4. Na-K-Mg ternary diagram of groundwaters in EAFZ. Literature data source is consistent with Fig. 2.

## 5.1.3 87Sr/86Sr characteristics of groundwaters

Radiogenic strontium isotopes (87Sr/86Sr) serve as robust tracers of groundwater provenance. The measured 87Sr/86Sr ratios (0.7053–0.713) across EAFZ groundwaters reflect multi-source mixing processes. Central-southern groundwaters integrate signatures from: Shallow aquifers: Inheriting Sr from local lithologies (ophiolites) (Oyan, 2018); Modern seawater: 87Sr/86Sr = 0.7092–0.7096 (Mediterranean seawater) (Banner, 2004; Bernat et al., 1972); River inputs: Enriched ratios (>0.710) from silicate weathering. Binary mixing models using 87Sr/86Sr vs. Ca/Sr ratios (Fig. 5) quantify source contributions: Carbonate weathering dominates, consistent with Ca-HCO₃ hydrochemical type; Ophiolite contributions 

Fig. 5.  $^{87}$ Sr/ $^{86}$ Sr vs. Ca/Sr of groundwaters in the EAFZ. The mixing-boundary lines are built with the following end members: Mediterranean Sea water Ca = 411ppm, Sr = 8.30ppm  $^{87}$ Sr/ $^{86}$ Sr = 0.7092 (Banner, 2004; Bernat et al., 1972); Cretaceous Kızıldağ ophiolite CaO = 9.7%, Sr = 1088.10ppm  $^{87}$ Sr/ $^{86}$ Sr = 0.7032 (Oyan, 2018); Shallow groundwater (HS08) Ca = 55.34ppm, Sr = 0.06ppm  $^{87}$ Sr/ $^{86}$ Sr = 0.7150 (Affected by silicate weathering); Evaporite CaO = 29.5%, Sr = 149ppm  $^{87}$ Sr/ $^{86}$ Sr = 0.7085 (Güngör Yeşilova and Baran, 2023).

Fig. 6. Characteristics of chemical components of groundwaters in the EAFZ, during water-rock interaction. The dashed line is the numerical simulation result of PHREEQC. a: Ca<sup>2+</sup> vs SO<sub>4</sub><sup>2-</sup>, b: Na<sup>+</sup> vs Cl<sup>-</sup>, c: Na<sup>+</sup> vs HCO<sub>3</sub><sup>-</sup>+Cl<sup>-</sup> and d: Na<sup>+</sup> vs HCO<sub>3</sub><sup>-</sup>. The simulation calculations are detailed in Supporting Information Part 1. Literature data source is consistent with Fig. 2.

5.2 The groundwater circulation in different segments of EAFZ

## 5.2.1 Water-rocks interaction

Pre-seismic whitish discoloration and turbidity anomalies observed at HS04 and HS14 groundwater monitoring stations likely reflect seismically induced perturbations to water-rock equilibrium (Video 1 and 2). To validate this hypothesis, we conducted numerical simulations of water-rock interaction processes across distinct segments of EAFZ, aiming to reconstruct their hydrochemical evolution.

Fig. 6 indicates pronounced disparities in groundwater chemistry between northern and central-southern segments. As discussed, elevated Na<sup>+</sup> and Cl<sup>-</sup> concentrations in northern groundwaters suggest magmatic fluid contributions. During ascent, these deep-sourced Na-Cl rich fluids mix with shallow groundwater while reacting with surrounding rocks. To quantify magmatic mixing ratios and reaction pathways, we first characterized dominant lithologies in the northern EAFZ—basalt, basaltic andesite, and sedimentary

cover (clastics and carbonates). CIPW norm calculations were employed to estimate mineral abundances, followed by PHREEQC-based reactive transport modeling (Parkhurst and Appelo, 2013) (see Supplementary File 1 for parameters). Simulation results (Fig. 6) demonstrate that linear correlations between Na<sup>+</sup> and (HCO<sub>3</sub><sup>-+</sup> Cl<sup>-</sup>) arise from magmatic NaCl fluid-carbonate interactions, with magmatic contributions accounting for 0–7% of total mixing.

In contrast, central–southern groundwaters lack magmatic signatures but exhibit Ca<sup>2+</sup>–SO<sub>4</sub><sup>2-</sup> covariation indicative of anhydrite dissolution (Fig. 6). Central segment waters reflect mixed carbonate- anhydrite controls (30% anhydrite contribution), while southern systems are dominated by anhydrite-derived solutes (100%), sourced from extensive evaporite deposits of the paleo–Amik Lake. Silica–enthalpy mixing models estimate reservoir temperatures of 234°C (HS04) and 155°C (HS04) (Fig. 7a), under which anhydrite saturation indices confirm its dissolution dominance (Fig. 7b). Notably, HS14—located

20 km from the paleo-Amik Basin-displayed prominent pre-seismic turbidity anomalies, likely

triggered by earthquake-driven disruption of anhydrite equilibrium. Coseismic changes in temperature,

pressure, fracture density, and circulation depth may have enhanced evaporite dissolution, increasing

Saturation index of Anhybrite  $SiO_2(mg/L)$ Enthalpy (J/g) T(°C)

groundwater salinity.

Fig. 7. a: Silica-enthalpy model of groundwaters in EAFZ. b: Temperature versus variation of anhydrite saturation indices of groundwaters in EAFZ. The enthalpies and reservoir temperatures of sample HS04 and HS14 are 981 J/g, 234 °C and 648 J/g, 156 °C respectively. The blue diamond is sample HS08, which is river water. At reservoir temperature, the anhydrite in HS04 and HS14 samples is saturated, indicating that anhydrite dissolution occurs during the water-rock reaction.

5.2.2 Contribution of mantle degassing to EAFZ groundwater circulation

Geochemical studies of EAFZ geothermal gases indicate significant mantle degassing (Fig. 8), where sulfur volatiles (e.g., SO<sub>2</sub> and H<sub>2</sub>S) ascend through fault conduits and oxidize upon mixing with shallow

groundwater, ultimately mobilizing as  $SO_4^{2^-}$  in thermal fluids. Consequently, mantle-derived sulfur contributions to groundwater sulfate inventories cannot be disregarded. Lacking  $O_2$  was detected in EAFZ geothermal gases suggested that the dissolved oxygen may have been consumed (Italiano et al., 2013; Yuce et al., 2014). However, it is important to note that  $H_2S$ ,  $H_2$ , and  $CH_4$  can all react with oxygen. Thermodynamic calculations indicate that  $CH_4$  is more favorable than  $H_2S$  in oxidation reactions ( $\Delta G^\circ$   $CH_4 = -818.1$  kJ/mol,  $\Delta G^\circ$   $H_2S = -494.2$  kJ/mol, at 298 K and 1atm). In actual geothermal systems, however, the depletion of  $H_2S$  is more commonly observed than the depletion of  $CH_4$ . We propose the following possible explanations: 1) Oxidation of  $H_2S$ : While thermodynamic calculations predict  $CH_4$  oxidation first, a small amount of  $H_2S$  might still be oxidized simultaneously with  $CH_4$ . Due to the much lower concentration of  $H_2S$  in geothermal systems compared to  $CH_4$ ,  $H_2S$  is consumed more quickly, leaving  $CH_4$  with a higher residual concentration. 2) Exogenous  $CH_4$  Supply: In addition to mantle-derived  $CH_4$ , other sources of  $CH_4$ , such as biogenic  $CH_4$  and thermogenic  $CH_4$  (e.g., serpentinization), may contribute to the geothermal system. These external sources could increase the concentration of  $CH_4$  in the geothermal fluids.

Fig. 8. Helium isotope ratios (R/Ra, Ra = air  $^3$ He/ $^4$ He = 1.39 × 10<sup>-6</sup>) versus  $^4$ He/ $^2$ 0Ne ratios for EAFZ gas samples. The mixing-boundary lines are built with the following end members: Air R/Ra = 1 and  $^4$ He/ $^2$ 0Ne = 0.318; mantle R/Ra = 8 and  $^4$ He/ $^2$ 0Ne = 1000; continental crust R/Ra = 0.02 and  $^4$ He/ $^2$ 0Ne = 1000 (Sano and

Wakita, 1985). Literature data source from (D'Alessandro et al., 2018; Inguaggiato et al., 2016; Italiano et al., 2013; YASİN and YÜCE, 2023; Yuce et al., 2014; Yuce and Taskiran, 2013).

However, previous studies have shown that the geothermal gas in the southern segment of EAFZ has more crustal source components than northern segment (Fig. 8). Furthermore, isotopic evidence confirms substantial biogenic and serpentinization-derived CH<sub>4</sub> inputs (Italiano et al., 2013; Yan et al., 2024), whereas H<sub>2</sub>S remains below detection thresholds. This implies that while H<sub>2</sub>S may transiently influence redox cycling, its low abundance limits long-term impacts. Instead, post-seismic SO<sub>4</sub><sup>2-</sup> surges likely originate from shallow evaporite dissolution (anhydrite) or low-temperature metamorphic anhydrite hydration—processes amplified by coseismic fracture propagation and fluid remobilization.

#### 5.3 Geothermal fluid circulation model in the EAFZ

As discussed above, EAFZ's geothermal fluid circulation model is shown in the Fig. 9. Beginning in the Late Cretaceous, as the New Tethys Ocean closed, Arabia-Eurasia collision zone have accommodated ~350 km of convergence, making crust up to 45 km thick, and causing >2 km of uplift (Yönlü et al., 2017). Arabian lithospheric mantle extends 50~150 km north beneath Anatolian crust (Whitney et al., 2023). Subsequently, the "roll back" and "slab break" occurred, resulting in extensive volcanic and devastating earthquakes, including those of February 6, 2023 in East Anatolian Plateau (Zhou et al., 2024). The collision of the Eurasian and Arabian plates caused Anatolian microplate was extruding westwards, which lead to EAFZ at a high strike-slip rate of ~11 mm/yr (Pousse - Beltran et al., 2020), and accompanied by counterclockwise rotation with a rotation rate of 1.053 ±0.015°/Ma (Simão et al., 2016). In this tectonic context, EAFZ remains active for a long time. Paleoseismic studies have shown that EAFZ has had many large earthquakes in its history (Carena et al., 2023; Hubert-Ferrari et al., 2020; Sparacino et al., 2022; Tan et al., 2008; Yönlü et al., 2017), with the largest magnitude reaching Mw 8.2 (Carena et al., 2023). Fault that cut through the crust provide channels for material and energy to rise up from mantle, which makes EAFZ geothermal gas contain a high proportion of mantle-derived compositions (Aydin et al., 2020; Italiano et al., 2013; Yuce et al., 2014). However, the transport of geothermal gas and geothermal water appears to be decoupled. On the one hand, deep geothermal fluid stays deep under the influence of gravity and less diffusive, compare to geothermal gas. On the other hand, the geothermal fluid was diluted due to the infiltration of a large amount of shallow cold water after the double earthquakes in February 2023 (Mw 7.8 and Mw 7.6). Our interpretation can better explain the lack of deep fluid signal in the groundwater studied in this study. Subsequently, at a depth of 4km, gas-water interaction process was experienced. Finally rose to the surface and discharged into the atmosphere. On the contrary, the circulating groundwater has undergone complex water-rock interaction processes such as anhydrite, calcite, dolomite, anorthite and serpentinization (Fig. 9).

Fig. 9. The genesis model of the geothermal fluids in the EAFZ. The deep geothermal fluid was diluted due to the infiltration of a large amount of shallow cold water. In the shallow crust, gas-water interaction process and water-rock interaction processes were experienced. The gases rose to the surface and discharged into the atmosphere. The circulating groundwater has undergone complex such as anhydrite, calcite, dolomite, anorthite and serpentinization.

5.4 The relationship between geothermal fluid and earthquake forecasting

Earthquake forecasting is a grand goal pursued by human beings, but also one of the most difficult goals. Various physical, chemical and biological techniques are used for earthquake forecasting (Bayrak et al., 2015; Güleç et al., 2002; Kwiatek et al., 2023; Luo et al., 2024; Luo et al., 2023; Miller et al., 2004; Nalbant et al., 2002; Skelton et al., 2014; Tsunogai and Wakita, 1995; Wakita et al., 1980). As a link between the shallow (crust) and the deep (mantle), geothermal fluids can react to various diseases just like human blood. In earlier studies, researchers found that the anomaly of chemical indicators in geothermal fluids could be used for earthquake forecasting e.g., (Güleç et al., 2002; King et al., 2006;

Miller et al., 2004; Perez et al., 2008; Poitrasson et al., 1999; Tsunogai and Wakita, 1995), but due to limited technology and funding, such research requiring long-term and large-scale monitoring is difficult to carry out (Ingebritsen and Manga, 2014). With the advancement of technology, more and more automated equipment and the development of 5G communication technology make long-term automatic monitoring possible, e.g., (Barbieri et al., 2021; Boschetti et al., 2022; Franchini et al., 2021; Liang et al., 2023; Luo et al., 2024; Luo et al., 2023; Skelton et al., 2014; Wang et al., 2023a). However, before geothermal fluid is really used in earthquake prediction, there is a problem that must be solved (i.e. to understand the relationship between geothermal fluid and earthquake). Its essence is to restore the origin and evolution process of geothermal fluid (Boschetti et al., 2022). For a long time, researchers have been searching for the information of the deep fluid in the fault zone, trying to link the earthquake with the deep fluid activity (Liang et al., 2023; Luo et al., 2023; Yan et al., 2024). However, deep information is easily changed during upward migration, and sometimes even lacks deep information, just like the EAFZ groundwater in this study (Fig. 6). This seems to limit the ability of groundwater to be used for earthquake prediction. In fact, chemical anomalies related to seismic activity can still be found in some shallow circulating groundwater (e.g., SO<sub>4</sub><sup>2-</sup>) (Luo et al., 2023). Moreover, the shallower water-rock interactions are more sensitive to the environment. Anhydrite are widely distributed in nature, and its formation is related to evaporite or hydrothermal metasomatism. Dissolution and precipitation of anhydrite are often observed in groundwater. Its solubility is greatly affected by environmental conditions (temperature, pH, pressure surrounding rock condition etc.) and they are potential indicators of tectonic activity. After the 2023 Mw 7.8 and 2023 Mw 7.6 earthquake, in the absence of deep fluid signals, we observed anhydrite dissolution at central-southern segments of EAFZ, which are likely to have been affected by seismic activity (Fig. 6). Similar SO<sub>4</sub><sup>2-</sup> anomalies have also been found in the eastern Tibetan Plateau (Li et al., 2021; Luo et al., 2023) and southeast China (Wang et al., 2021). Therefore, we suggest that anhydrite can be used as a potential tectonic activity index. However, although anhydrite's potential as a tectonic activity proxy is significant, its shallow crustal occurrence renders it susceptible to climatic perturbations (e.g., rainfall, evaporation). As evidenced in Fig. 6, post-seismic SO<sub>4</sub><sup>2-</sup> and Ca<sup>2+</sup> concentrations show no statistically significant deviations from background levels during quiescent periods, underscoring the challenge of filtering out climatic noise. While statistical correlations tentatively position anhydrite dissolution as a fault activity indicator,

advancing this paradigm requires: Long-term, high-resolution monitoring to disentangle tectonic vs. meteoric signals; Mechanistic models integrating fracture permeability dynamics with anhydrite solubility kinetics.

This study's key contribution lies in establishing fault-driven permeability changes as a viable driver of anhydrite dissolution. We propose a novel conceptual framework for fault activity monitoring via groundwater systems—one that prioritizes reactive minerals in shallow water-rock interactions over traditional deep fluid signals.

### **6 Conclusions**

Segmented groundwater provenance: Northern groundwaters represent mixing between mantle-derived magmatic fluids (0–7%) and shallow meteoric waters, while central-southern systems are dominated by carbonate-evaporite weathering with localized seawater/halite inputs.

Tectono-Climatic controls on water-rock interactions: Plagioclase-carbonate dissolution dominates northern segments, whereas anhydrite dissolution (30–100%) in central-southern segments correlates with fault permeability changes. Seismically enhanced fracture networks amplify evaporite dissolution, driving hydrochemical anomalies.

Anhydrite as a tectonic activity tracer: Despite climatic noise, anhydrite dissolution kinetics exhibit stress-state sensitivity. Their ubiquity and rapid stress response position anhydrite as a potential tracer for real-time fault activity monitoring.

- Code and data availability. All water data are listed in the text or in the Supporting Information.
- **Supplement.** See Supporting Information.
- Authorship contributions. Zebin Luo: Conceptualization, Methodology, Software, Writing-Original
- Draft, Writing-Review and Editing. Xiaocheng Zhou: Conceptualization, Validation. Yueren Xu:
- Investigation. Peng Liang: Investigation. Huiping Zhang: Investigation. Jinlong Liang: Validation.
- Zhaojun Zeng: Investigation. Yucong Yan: Investigation. Zheng Gong: Investigation. Shiguang Wang:
- Investigation. Chuanyou Li: Investigation. Zhikun Ren: Investigation. Jingxing Yu: Investigation.
- **Zifa Ma:** Investigation. **Junjie Li:** Investigation.
- Competing Interests. The authors declare that they have no known competing financial interests or
- personal relationships that could have appeared to influence the work reported in this paper.
- **Acknowledgements.** We would like to thank the Associate Editor Prof. Heng Dai, Walter D'Alessandro
- Giovanni Martinelli, Hafidha Khebizi and another anonymous reviewers for their constructive comments,
- suggestions and corrections. We also thank Dr. Yinchun Wang Dr. Renjie Li and Dr. Yi Yu for discussion,
- Dr. Shiqi Zhang for her help for diagram drawing.
- Financial support. The work was funded by National Key Research and Development Project
- (2024ZD1000503, 2023YFC3012005-1), Central Public-interest Scientific Institution Basal Research
- Fund (CEAIEF20240405, CEAIEF2022030200, CEAIEF2022030205), the National Natural Science
- Foundation of China (41673106, 4193000170), IGCP Project 724.

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
