# Peer review of "Tracer of Seismic-Fluid Coupling: Insights from the East"

_Hydrology and Earth System Sciences, 2024_

## Author Comment (AC1)

Dear Giovanni Martinelli

Thank you for your recognition of our work and valuable suggestions, which are very helpful for us to improve the quality of our manuscripts. Your two comments are exactly where we are lacking. At your suggestion, we plan to add a subsection to the discussion section for assessing the contribution of mantle degassing to EAFZ

geothermal fluids. The supplementary content is as follows:

*Contribution of Mantle Degassing to EAFZ Geothermal Fluids*

Mantle degassing occurs extensively along fault zones, and the amount of volatile release can sometimes be comparable to the degassing associated with volcanic activity e.g. (Fischer and Aiuppa, 2020; Zhang et al., 2021). Sulfur-containing volatiles (such as $SO_2$ and $H_2S$) ascend along these fault zones and, upon reaching the shallow subsurface, mix with groundwater, where they are oxidized and migrate in the form of

$SO_4^{2-}$ in geothermal fluids. Therefore, the contribution of mantle degassing to the $SO_4^{2-}$

content in geothermal fluids cannot be overlooked. To better assess the contribution of mantle degassing to $SO_4$ in EAFZ geothermal fluids, we need to consider the sources and modifications of geothermal fluids.

The deep-origin geothermal fluids in EAFZ are significantly diluted by shallow groundwater, masking the chemical signature of deeper fluid components. This dilution process introduces a large amount of dissolved oxygen, which facilitates the oxidation of $H_2S$ to $SO_4^{2-}$. Lacking $O_2$ was detected in EAFZ geothermal gases suggested that the dissolved oxygen may have been consumed (Italiano et al., 2013; Yuce et al., 2014).

However, it is important to note that $H_2S$, $H_2$, and $CH_4$ can all react with oxygen.

Thermodynamic calculations indicate that $CH_4$ is more favorable than $H_2S$ in oxidation reactions ($\Delta G°$ $CH_4$ = -818.1 kJ/mol, $\Delta G°$ $H_2S$ = -494.2 kJ/mol, at 298 K and 1atm). In actual geothermal systems, however, the depletion of $H_2S$ is more commonly observed than the depletion of $CH_4$, suggesting that $H_2S$ may be oxidized before $CH_4$. To resolve this apparent contradiction, we propose the following possible explanations: 1)

Oxidation of $H_2S$: While thermodynamic calculations predict $CH_4$ oxidation first, a small amount of $H_2S$ might still be oxidized simultaneously with $CH_4$. Due to the much lower concentration of $H_2S$ in geothermal systems compared to $CH_4$, $H_2S$ is consumed more quickly, leaving $CH_4$ with a higher residual concentration. 2) Exogenous $CH_4$

Supply: In addition to mantle-derived $CH_4$, other sources of $CH_4$, such as biogenic $CH_4$

and thermogenic $CH_4$ (e.g., serpentinization), may contribute to the geothermal system.

These external sources could increase the concentration of $CH_4$ in the geothermal fluids.

In the EAFZ, we observed significant contributions of biogenic and serpentinization-derived $CH_4$ but did not detect significant levels of $H_2S$ (Italiano et al.,

2013; Yuce et al., 2014). Therefore, we proposed that although $H_2S$ may contribute to the geothermal system, its impact is likely limited due to its relatively low concentration.

Inversely, the notable increase in $SO_4^{2-}$ concentrations following seismic events is likely primarily controlled by the dissolution of shallow evaporitic layers (such as gypsum).

All in all, while the oxidation of $H_2S$ may contribute to $SO_4^{2-}$ formation, distinguishing between $H_2S$ oxidation and sulfate dissolution requires additional geochemical indicators, such as S isotopes and Ca isotopes, for more accurate assessments.

**References**

Fischer, T. P. and Aiuppa, A.: AGU Centennial Grand Challenge: Volcanoes and Deep Carbon Global
$CO_2$ Emissions From Subaerial Volcanism-Recent Progress and Future Challenges,
Geochemistry Geophysics Geosystems, 21, 2020.
Italiano, F., Sasmaz, A., Yuce, G., and Okan, O. O.: Thermal fluids along the East Anatolian Fault
Zone (EAFZ): Geochemical features and relationships with the tectonic setting, Chemical Geology,
339, 103-114, 2013.
Yuce, G., Italiano, F., D'Alessandro, W., Yalcin, T. H., Yasin, D. U., Gulbay, A. H., Ozyurt, N. N., Rojay,
B., Karabacak, V., Bellomo, S., Brusca, L., Yang, T., Fu, C. C., Lai, C. W., Ozacar, A., and Walia, V.:
Origin and interactions of fluids circulating over the Amik Basin (Hatay, Turkey) and relationships
with the hydrologic, geologic and tectonic settings, Chemical Geology, 388, 23-39, 2014.
Zhang, M., Xu, S., Zhou, X., Caracausi, A., Sano, Y., Guo, Z., Zheng, G., Lang, Y.-C., and Liu, C.-Q.:
Deciphering a mantle degassing transect related with India-Asia continental convergence from
the perspective of volatile origin and outgassing, Geochimica Et Cosmochimica Acta, 310, 61-78,
2021.

---

## Author Comment (AC3)

Dear Walter D'Alessandro

Thank you for your highly professional and constructive comments and suggestions, which are of great value to us in improving the quality of our manuscript. After carefully reading your comments, we have made a reply to your comments point-by-point under the discussion of all manuscript authors. The main replies are as follows:

**Major revisions include:**

1. Correction of sample collection time: We apologize for marking the wrong sampling time in Table 1 (marked time, March 2024, **the actual sampling time, March 2023**). The wrong timing brings huge ambiguity to the manuscript. After correcting the sampling time, the main line logic of the article is as follows:

These evidences constitute a complete chain of causality from the source (evaporite) to the process (water-rock reaction balance disrupted by the earthquake) to the response (abnormal groundwater ion concentration).

2. Use "**groundwater**" instead of "**geothermal water**" to define the sample in this study.

We collected 16 groundwater samples from SF and EAFZ within a month of the earthquake.

The principle of sample collection is to collect if we can. Because the overall temperature is low, we think it is more reasonable to use "groundwater" instead of "geothermal water".

3. We have given **a complete explanation of the pre-earthquake hydrochemical data** in the manuscript.

4. We supplement the analysis method and data quality control description

5. We rearranged the logic of the article to make the expression clearer

6. We have made a full explanation of some misunderstandings

7. We explain the possible "overestimation of the heat storage temperature" and analyze that the heat storage temperature estimate has little effect on the conclusion of our core conclusions.

8. We plan to conduct additional experiments on the samples, including **radioactive Sr**

**isotopes** and **S isotopes**, to support our argument with more evidence.

Since there are diagrams in the complete reply draft, we put the complete reply draft in the form of an attachment on the website system. If you have any questions or suggestions about the manuscript, we sincerely invite you to keep discussing with us. Thank you for constructive review comments.

Thank you and best regards.

**Point-by-point response to comments:**

Note: *Italic blue* is the comment. Black is the reply, and **important sentences are bolded and underlined**.

*The manuscript "Gypsum as a potential tracer of earthquake: a case study of the Mw7.8 earthquake in the East Anatolian Fault Zone, southeastern Turkey" by Luo et al. presents the results of sampling campaign of groundwaters in the area of the two strong earthquakes that hit heavily Turkey in February 2023. Only the analytical results (major ions, trace elements and water isotopes) of samples collected about one year after the quakes are considered, which is a strong limitation of this study. I feel that this study cannot be published in this form.*

Reply: Thanks. First of all, let's correct an error in Table 1 in manuscript. Our sampling time is **March 2023**, which is **one month** after the earthquake, not **one year**. We apologize for the sampling time error in manuscript (Table 1) and thank you for your careful correction. Therefore, combined with the **groundwater characteristics within one month after the earthquake**, **groundwater data before the earthquake** (obtained from literature research), and **macro anomalies before the earthquake** (whitening and turbidity), we believe that the evidence is sufficient to prove our view that the earthquake has broken the water-rock balance between gypsum and groundwater, and gypsum has the potential to act as an earthquake tracer.

In light of your suggestion, however, we are also considering the need to find **more evidence** to support our conclusion. Therefore, we are conducting **Radioactive Sr isotope** and **S isotope** analysis on our samples. 1) Radioactive Sr isotope is a good source indicator.

The radioactive Sr isotope composition of shallow gypsum dissolution and deep fluid is obviously different, so the radioactive Sr isotope may well restrict the source area of groundwater. 2) S isotope is the main constituent element of gypsum, and the S isotope composition of igneous rock ($\delta^{34}S$ = -5~10‰) is lower than that of evaporite ($\delta^{34}S$ > 10‰), so S isotope can better distinguish the S of evaporite and igneous rock.

**Major comments:**

*Lines 33-36 (abstract): This is one of the most critical claims made by the authors. "Specially, significant gypsum dissolution was observed at HS05, HS09 and HS14 before and after the earthquake, suggesting that the earthquake broke the balance of water-rock reaction and promoted the dissolution of gypsum." In the paper only the results of the analyses of the samples taken one year after the earthquakes are discussed. How should it be possible to evidence variations "before and after the earthquake" if only one sample was taken?*

Reply: Thanks. Sorry again for the error in sampling time in manuscript (Table 1). The exact date of our sample is March 2023. Therefore, our data can be representative of groundwater characteristics after the earthquake. Pre-earthquake data mainly come from *Yuce, G., Italiano, F., D'Alessandro, W., Yalcin, T. H., Yasin, D. U., Gulbay, A. H., Ozyurt, N. N., Rojay, B., Karabacak, V., Bellomo, S., Brusca, L., Yang, T., Fu, C. C., Lai, C. W., Ozacar, A., and Walia, V.: Origin and interactions of fluids circulating over the Amik Basin (Hatay, Turkey) and relationships with the hydrologic, geologic and tectonic settings, Chemical Geology, 388, 23-39, 2014.* After carefully checking the GPS coordinates given in the literature, we can confirm that HS14 is **kirikhan well** (A15), HS15 is **Tahtakopru**

(A12/13), and HS16 is **Kuzey Tepe** (A40) (Table 1). Compared with the literature data, the concentration of $SO_4^{2-}$ and $Ca^{2+}$ in sample HS14 increased.

      Table 1 Sample points and data for this study and literature

| This study | | | | | Yuce et al., 2014 | | | | | |
|---|---|---|---|---|---|---|---|---|---|---|
| Long(°) | Lat(°) | No. | $SO_4^{2-}$ (mg/L) | $Ca^{2+}$(mg/L) | Long(°) | Lat(°) | No. | $SO_4^{2-}$ (mg/L) | $Ca^{2+}$(mg/L) | Site name |
| 36.3738 | 36.5036 | HS14 | 316.61 | 151.43 | 36.3741 | 36.5034 | A15 | 101 | 87.1 | kirikhan well |
| 36.1637 | 36.3833 | HS15 | 1.21 | 55.55 | 36.1636 | 36.3835 | A12/13 | 0.2 | 44.7 | Tahtakopru |
| 36.1472 | 36.2737 | HS16 | 75.9 | 73.35 | 36.1471 | 36.2738 | A40 | 361 | 41.1 | Kuzey Tepe |

Pre-seismic mean values of $SO_4^{2-}$ and $Ca^{2+}$ are from Baba et al., 2019. But you mentioned that our average is inconsistent with the data in the Baba et al., 2019. We apologize for any confusion caused by not clearly stating how the data was referenced. Our average does refer to Baba et al., 2019, but not entirely. **We only cite data from sample points close to EAFZ**.

The reason for this: Baba et al 2019 evaluated geothermal resources throughout southeastern Turkey. If we average all the data, this is obviously not reasonable. Moreover,

**it can also be seen from Baba et al., 2019 that there is a big difference between**

**geothermal resources near EAFZ and those far away from EAFZ (Fig. 1).** Geothermal resources near EAFZ are mainly medium and low temperature. Therefore, when considering the EAFZ pre-earthquake $SO_4^{2-}$ and $Ca^{2+}$ concentrations, **we only chose the**

**average values of 1, 2, 3, 4, 7, 9, 26 and 27 in the paper as the pre-earthquake**

**concentrations (Fig. 2 and Table 2)**.

[Figure]

Fig. 1: Temperature distribution map of geothermal resources in southeast Turkey. Screenshot from *Baba, A., Şaroğlu, F., Akkuş, I., Özel, N., Yeşilnacar, M. İ., Nalbantçılar, M. T., Demir, M. M., Gökçen, G., Arslan, Ş., Dursun, N., Uzelli, T., and Yazdani, H.: Geological and hydrogeochemical properties of geothermal systems in the southeastern region of Turkey, Geothermics, 78, 255-271, 2019.*

[Figure]

Fig. 2: Baba et al., 2019 sampling point distribution map. Screenshot from *Baba, A., Şaroğlu, F., Akkuş, I., Özel, N., Yeşilnacar, M. İ., Nalbantçılar, M. T., Demir, M. M., Gökçen, G., Arslan, Ş., Dursun, N., Uzelli, T., and Yazdani, H.: Geological and hydrogeochemical properties of geothermal systems in the southeastern region of Turkey, Geothermics, 78, 255-271, 2019.*

Table 2 Ion concentration before earthquake.

| No. | $Ca^{2+}$(mg/L) | $SO_4^{2-}$(mg/L) |
|-----|------------|--------------|
| 1 | 14.92 | 0.01 |
| 2 | 66.92 | 0.01 |
| 3 | 45.56 | 9.86 |

| | | |
|---|---|---|
| 4 | 63.84 | 24.79 |
| 7 | 116.03 | 10.22 |
| 9 | 38.65 | 3.34 |
| 26 | 39.85 | 1.83 |
| 27 | 56.03 | 16.41 |
| Average | 55.23 | 8.31 |

Data from: *Baba, A., Şaroğlu, F., Akkuş, I., Özel, N., Yeşilnacar, M. İ., Nalbantçılar, M. T., Demir, M. M., Gökçen, G., Arslan, Ş., Dursun, N., Uzelli, T., and Yazdani, H.: Geological and hydrogeochemical properties of geothermal systems in the southeastern region of Turkey, Geothermics, 78, 255-271, 2019.*

*Line 124: The authors should explain on which basis the 16 sampling sites have been chosen.*

Reply: Thanks. Samples were collected from north to south along the EAFZ. All the places with springs were sampled. Considering the safety considerations after the earthquake, there may be some missing spring points compared with previous studies. But our sampling was done in conjunction with the post-earthquake research in Turkey. In addition to water sampling, Also analyzed the surface rupture and earthquake risk assessment (*Liang, P., Xu, Y., Zhou, X., Li, Y., Tian, Q., Zhang, H., Ren, Z., Yu, J., Li, C., Gong, Z., Wang, S., Dou, A., Ma, Z., and Li, J.: Coseismic surface ruptures of MW7.8 and MW7.5 earthquakes occurred on February 6, 2023, and seismic hazard assessment of the East Anatolian Fault Zone, Southeastern Turkiye, Science China Earth Sciences, doi: 10.1007/s11430-024-1457-7, 2024.*). Therefore, we can guarantee the representativeness and reliability of the samples in this study.

We added the description of the sampling point: "**HS01-HS04 was collected from west to east along SF. HS07-HS16 was collected from north to south along EAFZ (Fig. 1)**"

*Line 124: the authors claim to have sampled hot springs but with the exception of the peculiar hyperalkaline spring HS15, which derive its increased temperature from deep circulation, no other sample could be called "hot". Furthermore, I would not define a well with water at 24 °C as geothermal well. Actually, in the results (line 144) the authors affirm that temperatures of the sampled waters are low.*

Reply: Thanks. Indeed, the temperature of all samples in this study is low, indicating that EAFZ is a medium-low temperature hydrothermal system, which is also consistent with the research results of Baba et al., 2019. However, as you said, the temperature of the sample is really low. We also feel that the term "**geothermal water**" is not rigorous enough to describe our samples. Therefore, we considered using the more appropriate term "**groundwater**" to describe our samples. But in fact, whether groundwater or geothermal water, the core point of our manuscript is not contradictory. The use of groundwater chemistry and isotopes to study the water-rock balance before and after earthquakes is considered to be a very effective means (*e.g., Skelton, A., Andren, M., Kristmannsdottir, H., Stockmann, G., Morth, C.-M., Sveinbjoernsdottir, A., Jonsson, S., Sturkell, E., Gudorunardottir, H. R., Hjartarson, H., Siegmund, H., and Kockum, I.: Changes in groundwater chemistry before two consecutive earthquakes in Iceland, Nature Geoscience, 7, 752-756, 2014. and Tsunogai, U. and Wakita, H.: Precursory chemical changes in ground water: kobe earthquake, Japan, Science (New York, N.Y.), 269, 61-63, 1995.*). However, considering the influence of groundwater on many factors (e.g., temperature, pressure, climatic conditions, seasonal changes etc.), we have explained in the abstract and conclusion of the manuscript that gypsum needs to be considered more carefully.

*The methodological section has many limitations:*

*Lines 130-131: it is unclear if filtration has been made in the field and before acidifying*

*the aliquot for cation analysis. Please specify*

Reply: Thanks. Yes, **we confirmed filtering before testing**. The relevant description can be found in lines 130-131 of the original manuscript. We have extensive experience in groundwater and gas extraction. We can guarantee the reliability of sample collection methods and data.

*Line 131: MAT 253 is a model, please specify the used technique*

Reply: Thanks. We have added specific analytical method: "$\delta$D and $\delta^{18}$O were determined by **zinc reducing tube sealing method** combined with MAT 253 (relative to Vienna

Standard Mean Ocean Water (V - SMOW)). Precisions on the measured $\delta^{18}$O and $\delta$D value was ±0.2% (2SD) and ±1% (2SD) respectively (Wang et al., 2010)."

*Line 133: please specify the analysed species and the relative reproducibility and detection*

*limits?*

Reply: Thank you for pointing out the problem of the manuscript. We have added the reliability description of hydrochemistry and isotope analysis to the chapter of **Analytical**

**methods**, the details are as follows:

16 samples of water were collected in EAFZ, including hot springs, geothermal wells and river water. HS01-HS04 was collected from west to east along SF. HS07-HS16 was collected from north to south along EAFZ (Fig. 1). Detailed sample collection and testing methods can be found at Luo et al. (2023). In short, the water sample was taken with a 50

mL clean polyethylene bottle and the temperature and pH of the water were measured and recorded. Two samples are collected at each sampling site, one is added with ultrapure

$HNO_3$ to analyse the cation content, and the other is used to analyse the anion content and isotopic composition. **All samples need to be pre-treated with a 0.45 μm filter**

**membrane to remove impurities before being tested.** δD and $\delta^{18}O$ were determined by

**zinc reducing tube sealing method** combined with MAT 253 (relative to Vienna Standard

Mean Ocean Water (V - SMOW)). **Precisions on the measured $\delta^{18}O$ and δD value was**

**±0.2% (2SD) and ±1% (2SD) respectively (Wang et al., 2010)**. The cation **($Li^+$, $Na^+$, $K^+$,**

**$Ca^{2+}$and $Mg^{2+}$)** and anion **($F^-$, $Cl^-$, $NO_3^-$ and $SO_4^{2-}$)** were analysed by Dionex ICS-900

ion chromatograph (Thermo Fisher Scientific Inc.) **at the Earthquake Forecasting Key**

**Laboratory of China Earthquake Administration, with the reproducibility within ±2%**

**and detection limits 0.01 mg/L (Chen et al., 2015)**. $HCO_3^-$ and $CO_3^{2-}$ was determined by acid-base titration with a ZDJ-100 potentiometric titrator (reproducibility within ±2%).

$SiO_2$ were analysed by inductively coupled plasma emission spectrometer Optima-5300

DV (PerkinElmer Inc.) (Li et al. 2021). Trace elements were analysed by Element XR ICP-

MS at the Test Center of the Research Institute of Uranium Geology. Multielement standard solutions (IV-ICPMS 71A, IV-ICP-MS 71B and IV-ICP-MS 71D, iNORGANIC

VENTURES) used for quality control. **The analytical error margin of major cations and**

**trace elements were less than 10%)**.

*Line 136: please specify the analysed trace elements and the relative reproducibility and*

*detection limits?*

Reply: Thanks. The specific types of trace elements are shown in Table 2 (manuscript), the detection limit is 0.001μg/L, and the analysis error accuracy is less than 10%

*In the results the authors claim often that some element or ionic species is increased*

*(sometimes adding obviously) but they do not specify with respect to what. Maybe they*

*intend that the concentrations are high.*

Reply: Thanks. In the Results section we are an objective description of the results based on the data. The words "increased" and " obviously " were also relative to other sample results. But, in fact, what we mean is, "relatively high," not " increased." We apologize for any confusion caused by the poor description of the results, and we have re-optimized the presentation and added a quantitative description of the increased concentrations. The revised expression is as follows:

**The concentration of $SO_4^{2-}$ range from 1.21 mg/L to 316.61 mg/L, and the**

**concentration of $SO_4^{2-}$ in some samples is relatively high (e.g. HS01 (287.74 ml/L),**

**HS03 (103.56 ml/L), HS04 (229.75 ml/L), HS14 (316.61 ml/L)).**

*In the same section they speak of geothermal water but they do not present any evidence*

*that these are geothermal waters.*

Reply: Thank you. We have replaced "**groundwater**" with "**geothermal water**" to make the expression more precise.

*The discussion about the geothermal fluids has great limitations.*

*The authors do not present evidences that the sampled waters are, at least partially, fed by*

*hydrothermal systems. The fact that in the area some geothermal system has been*

*discovered and studied, does not mean that all groundwater samples taken in the area are*

*fed by them. The temperatures of the collected samples are low and, as highlighted by the*

*binary diagram of fig. 3 and the ternary diagram of fig. 4, their compositions do not reflect*

*high temperature interactions with the rocks. Also the silica geothermometers show low*

*temperatures considering that for such systems equilibrium with chalcedony (or even*

*christobalite or amorphous silica) should be taken into consideration.*

Reply: Thanks. We have already discussed this issue in the previous reply. **Hydrothermal**

**systems** and **groundwater** do not affect our core point. Both geothermal water and groundwater chemical anomalies are considered to be effective means of earthquake early warning. Thanks for your suggestion to us, as mentioned earlier, we have considered using

"**groundwater**" instead of "**geothermal water**" to define the samples for this study.

*Especially the use of the mixing models has been made in the wrong way. Mixing models*

*can be applied only to water samples that belong to the same system and not to water*

*samples collected tens of km away from each other and for which no connection has been*

*demonstrated.*

Reply: Thanks. Although the spatial span of the samples in this study is very large (~270

km) (Fig. 1 and Fig. 6 in manuscript), all of them belong to EAFZ. It is difficult to directly conclude that there is no genetic connection between them.

In fact, both the estimation of heat storage temperature and the mixed model only play an auxiliary supporting role in our core view. **Our main concern is the anomaly of ion**

**concentration caused by earthquake breaking the equilibrium of water-rock reaction**.

As for whether deep geothermal fluids are involved? What's the mixing ratio? It's all secondary evidence. Deep fluids may bring $SO_4^{2-}$ ($H_2S$ oxidation), but a little $Ca^{2+}$.

However, the correlation between $Ca^{2+}$ and $SO_4^{2-}$ was observed in EAFZ, and numerical simulations indicate that gypsum dissolution is indeed present (Fig. 7 in manuscript), coupled with the presence of large evaporite deposits in the ancient lacustrine sedimentary basin of Lake Amik. **These evidences constitute a complete chain of causality from the**

**source (evaporite) to the process (water-rock reaction balance disrupted by the**

**earthquake) to the response (abnormal groundwater ion concentration).**

Based on your comments, the geothermal properties of our samples are not strong and may not belong to hydrothermal systems. Therefore, we consider weakening the sections on heat storage, mixing ratio, and cycle depth. Delete this section or put in supplementary material.

As for the problem of using mixed models incorrectly. We don't think it can be completely negative. At least these samples are in EAFZ. The overestimation may be possible at 382℃.

But combined with the pre-seismic macroscopic anomaly of HS04, the content of $SiO_2$

(84.64mg/L) and the ion concentration anomalies of $Ca^{2+}$, $SO_4^{2-}$, Sr and Ba. We think it is sufficient to support the argument that the gypsum dissolution equilibrium was disturbed by the earthquake. Thank you.

*The estimation of temperature for the "deep geothermal fluid" (please define) of 382 °C is*

*absolutely unreliable. The sample was taken, as shown in the second video in the*

*supporting information, from an artesian well (although in table 1 it is classified as spring).*

*I think it is impossible that an artesian well, whose upflow is generally rapid, would have*

*only 15 °C temperature if even only a small part of the water would come from a geothermal*

*system with 382 °C.*

Reply: Thanks. Indeed, 382 °C may be overestimated. But as in the previous reply. The heat storage temperature is only secondary evidence for us to determine whether the gypsum was affected by the earthquake. We have considered deleting this part of the discussion or put in supplementary materials. **The estimate of 382°C is the HS04 sample**

**from the epicenter, and the complex process after the earthquake may be the reason**

**for our excessive estimate**. However, HS14 shows a lower estimated temperature, with the mixed model estimating only 88 °C (Fig. 5b). We propose that HS14 may be affected by shallow gypsum dissolution, and this lower estimated temperature supports this conjecture. Therefore, while 382 °C may not be rigorous enough, the estimation of HS14 supports our view.

*The discussion about the sulfate anomalies is highly confusing. Many points are unclear or*

*wrong.*

Reply: Thanks. We adjusted the description of the manuscript to make the logic clearer.

*Why are only samples HS05, HS09 and HS14 considered anomalous? HS01, HS03 and*

*HS04 have also elevated sulfate values.*

Reply: Thanks. This is actually a misunderstanding. The reason for the misunderstanding is that we failed to express it clearly in the manuscript, and there are logical problems. We consider optimizing the manuscript to eliminate misunderstandings. thank you!

We pointed out in the **Fig caption in Fig.6** that **only the spatial distribution**

**characteristics of EAFZ samples**, namely HS07-HS16, were considered in Fig.6. The discussion here does not cover SF samples (HS01-HS04). We considered adding a note to the text of the manuscript to make the logic clear.

In fact, as you commented, HS01, HS03, HS04, HS05, HS09, HS14 all have $SO_4^{2-}$

anomalies. However, the subsequent numerical simulation shows that the influencing factors of $SO_4^{2-}$ concentration increase in **HS01, HS03 and HS04 are more complex and**

**controlled by a variety of minerals (gypsum, calcite, dolomite, anorthite). However,**

**$SO_4^{2-}$ of HS05, HS09, HS14, especially HS14, is almost only controlled by gypsum (Fig.**

**7 in manuscript)**, and the influencing factors are relatively single. Therefore, HS14 is an important support for our main point, and the other points are ancillary.

*Why should these high sulfate values be considered anomalous and induced by the*

*earthquake? Sulfate dissolution from evaporite deposits within the aquifers is an ubiquitous*

*process independent from seismic activity.*

Reply: Thanks. The reason for your question is that we wrote down the sampling time incorrectly. I'm sorry. Our sampling time was within one month after the earthquake. **we**

**determined that the earthquake was one of the factors affecting the gypsum. But as**

**you commented, there are many factors affecting gypsum, and it can be disturbed**

**without earthquakes. Therefore, we emphasize this concern in both the abstract and**

**the conclusion, showing the limitations of gypsum as an indicator of earthquake**

**warning.**

*Why do the authors use these low averages for Ca (55.23 mg/L) and $SO_4$ (8.31 mg/L)*

*concentrations before earthquake? Baba et al. (2019) in their paper report concentrations*

*up to 773.56 mg/L for Ca and up to 1287.24 mg/L for $SO_4$ much higher than in the samples*

*collected for this study.*

Reply: Thanks. We have already replied to this comment before, and **we use the data near**

**EAFZ**. For this doubt, we consider to explain in the text to eliminate misunderstandings.

*Finally, the authors indicate the whitening and turbidity of the water in a sample as*

*verification for the sulfate anomaly. But without analysis there is no possibility to affirm*

*that such visual anomaly was due to gypsum dissolution.*

Reply: Thanks. The best evidence is our analysis of water samples taken within a month of the earthquake. Your confusion is caused by our marking of the wrong sampling time. Sorry again.

*Furthermore, the authors mistake the samples. The site with the high sulfate concentration*

*is HS14, while the site to which the pictures of figure S1 and of video 01 refer is HS15*

*which has the lowest sulfate value (1.21 mg/L).*

Reply: Thank you for pointing out this error, we have fixed it.

*Lines 388-389: The authors presenting the data of a single sampling campaign have no*

*evidence to affirm that "the geothermal fluid was diluted due to the infiltration of a large*

*amount of shallow cold water after the double earthquakes in February 2023".*

Reply: Thanks. As discussed earlier, we have considered replacing "geothermal water"

with "groundwater", so we will reconsider this conclusion. Thank you for your highly professional and constructive comments. Thanks again.

**Minor comments**

*Line 22: What do the authors mean with "systematic" which do not appear only in the*

*abstract but has been repeated many times in the whole text?*

Reply: Thanks.  In your professional comment, we also believe that "  systematic " may be a misnomer. We consider deleting the word.

*Lines 24 and 25: The meaning of the sentence is obscure (reconstructed by earthquake?)*

Reply: Thanks.  This sentence was not clear enough, so we adjusted the expression: **In**

**order to explore the relationship between groundwater anomaly and earthquake, we**

**performed hydrochemical and isotopic analyses of groundwaters in the East**

**Anatolian Fault Zone (EAFZ). The results show that groundwaters are affected by**

**seismic activity.**

*Line 29: the authors use often the term "abnormal" but they do never define with respect*

*to what.*

Reply: Thanks. "Abnormal" refers to values that deviate from normal values. Divided into

**time** and **space** outliers. In the manuscript, "anomaly" refers to spatial outliers. In particular, in Fig. 6, the mean values of $Ca^{2+}$ and $SO_4^{2-}$ (literature research) are compared with the temporal outliers in this study. The literature survey represents the data of the earthquake calm period, and this study represents the data of the earthquake active period.

*Line 38: please define "shallow minerals".*

Reply: Thanks. "Shallow minerals" is a relative term that generally refers to those minerals formed at or near the surface, mainly sedimentary rock related minerals. In this article mainly refers to gypsum. If "shallow mineral" is prone to ambiguity, **we consider directly**

**replacing "shallow mineral" with "gypsum".**

*Line 61: which evidence have the authors of a "geothermal fluids circulation"*

Reply: Thanks. We have replaced "groundwater" with "geothermal water". Therefore, the geothermal water cycle is no longer considered

*Line 69: please define the "geothermal fluid anomaly index"*

Reply: Thanks. The "**geothermal fluid anomaly index**" may be a misnomer, and we consider replacing it with "**groundwater chemical and isotopic anomaly index** ". Refers to changes in the water chemistry and isotopic composition of groundwater caused by changes in the external environment.

*Lines 70-71: the subject is missing in this sentence.*

Reply: Thanks. We deleted that sentence.

*Line 82: please define what a "tectonic collage" is.*

Reply: Thanks. We have adjusted the expression of this sentence: "**Located at the**

**intersection of Eurasia, Africa and Arabia, Turkey has a complex tectonic**

**background**".

*Fig. 1a: altitude scale is missing.*

Reply: Thanks. We added the  altitude scale (Fig. 3).

[Figure]

       Fig. 3 Geological map after adding altitude scale.

*Line 105: probably crystalline instead of crystallization.*

Reply: Thanks. We changed crystalline instead of crystallization.

*Line 145: in table 1 HS15 is considered a spring, which one is correct?*

Reply: Thanks. We checked the sampling point. HS15 is spring.

*Line 146: the authors claim that "the closer to the epicenter, the higher the $SiO_2$ content",*

*which makes no sense. Firstly because the earthquakes were two and only one sample close*

*to one of the epicenters has a higher $SiO_2$ value. Moreover, other two sampling points with*

*low to very low $SiO_2$ concentrations have the same position as the "anomalous" one.*

Reply: Thanks. We deleted that sentence

*Lines 154-156: the sentence "The $\delta 18O$ and $\delta D$ of samples varied from $-11.30‰$ to $-6.55‰$*

*and $-65.43‰$ to $-34.43‰$ respectively, which is near to the global meteoric water line*

*(GMWL) (Craig, 1961) (Fig. 3), suggesting their meteoric water origin" has no sense. The*

*regression line obtained plotting both $\delta^{18}O$ and $\delta D$ values in a graph can be close to GMWL.*

Reply: Thanks. We deleted that sentence.

*Line 159: what type of Statistical analysis?*

Reply: Thanks.  We have changed the word "statistical analysis" to "box-plot analysis" to make the expression more specific.

*Line 160: please define "fluid activity elements".*

Reply: Thanks. We adjusted the expression and used proper nouns: Fluid-mobile element (FME).

*Line 161: I do not understand what the authors mean with "are at historic highs versus".*

*If the authors mean that the concentrations are higher than in the past, then the fig. S2 does*

*not prove nothing. Al and Ba are below the median value of the literature data while the*

*remaining are around the median value not showing particularly high values. Furthermore,*

*it is unclear which data are compared in fig. S2 with the present data.*

Reply: Thanks. There is indeed ambiguity in the expression here, so we consider deleting the analysis of the packing diagram to make the manuscript more brief and clear.

*Table 1: please indicate the coordinates with at least 4 digits after the comma, with only*

*two digits it's impossible to obtain a reliable position. Looking at Fig. 1, the indicated*

*coordinates of HS05 are clearly wrong.*

Reply: Thanks. We adjusted the accuracy of the latitude and longitude to keep 4 decimal places.

*Line 190: the highest values do not belong to samples collected closer to the sea.*

Reply: Thanks. It's not rigorous enough. We've improved the sentence: "**The highest value**

**of $\delta D$ (−34.43‰)and $\delta^{18}O$ (−6.55‰) at the southwest of EAFZ, which is close to the**

**Mediterranean Sea, indicating that it originates from the recharge of the evaporation**

**of the Mediterranean Sea (Fig.3)**"

*Line 190: $\delta^{18}O$ and $\delta D$ values are inverted.*

Reply: Thank you. We've corrected it

*Line 212: magma mixing with geothermal fluids generally end in a volcanic explosion*

*which is not the case here.*

Reply: Thanks. It is true that magma usually accompanies volcanic activity. However, there may also be deep partial melting process in the deep fracture zone. For the sake of rigor, we consider using "partial melting" instead of "magma mixing".

*Lines 224-225: the sampling sites are tens of km far from the Mediterranean coastline, how*

*and why should they be "obviously contaminated by Mediterranean Sea water"?*

Reply: Thanks. It is tens of kilometers from the Mediterranean Sea, but from a geological perspective, it is very small. In the manuscript, our conclusions may be too arbitrary. We should consider the contribution of evaporites such as rock and salt. So, based on your comments, we've adjusted the sentence: "**HS16, the sample with the highest**

**concentration, was collected at the southwest of EAFZ, which was obviously**

**contaminated by Mediterranean Sea and/or halite.    There is no signal of deep fluid**

**or magma source.**"

*Line 226: which previous study? Please add a reference.*

Reply: Thanks. That sentence doesn't make sense. We deleted it.

*Line 233: pollution is a term connected to an anthropogenic origin, so please use the term*

*contamination instead.*

Reply: Thank you. We changed the word "pollution" to " contamination."

*Lines 233-236: I do not understand the meaning of this sentence.*

Reply: Thanks. We adjusted the expression to make the meaning clearer: "**In addition,**

**water is much less transferable than gas, which makes deep geothermal water may**

**not be able to rise along the fault to the shallow crust or surface like geothermal gas.**"

*Lines 290-292: the two processes are not alternative. Serpentinization includes secondary*

*minerals precipitation.*

Reply: Thanks. We adjusted the expression to make the meaning clearer: "**Compared with**

**other samples, the ion concentration of HS15 is significantly reduced, which may**

**indicate the precipitation of potential secondary minerals (e.g., calcite). Therefore, we**

**conjecture that serpentinization and secondary mineral precipitation such as: calcite**

**or magnesite (Aydin et al., 2020; Cipolli et al., 2004) may be responsible for the**

**increase in pH (Huang; et al., 2023).**"

*Finally, I would signal a possible conflict of interest being the handling editor of the same*

*institution of one the corresponding author.*

Reply: Thanks. China University of Geosciences (Beijing) and China University of Geosciences (Wuhan) are two independent universities with no conflict of interest.

---

## Author Comment (AC6)

**Reply on RC2**

Dear Walter D'Alessandro

Thanks for your comments again. According to your comments, we added the supplement and analysis of the literature data from 2013 to 2025 to make the data more representative. On this basis, the conclusion of the original manuscript has been revised to weaken the connection between gypsum and seismic activity, and emphasize the sensitive indication of gypsum to the intensity of water-rock interaction. The main replies are as follows. Note: *Italic blue* is the comment. Black is the reply.

*I am sorry to say that reading the reply of the authors my opinion regarding the manuscript did not change. My main criticism relates to the fact that it is not possible to evidence anomalies in groundwater composition related to seismic events having data collected only one time. The authors try to compare their data with other taken from literature but the comparison is not straightforward because no background values have ever been defined. The mean values utilised seem artificially created and, in my opinion, do not represent "normal" values.*

*I am still convinced that the manuscript in this form has to be rejected.*

Reply: Thanks! We sincerely appreciate your critical feedback and fully acknowledge the limitations of single-time sampling in establishing seismic-hydrogeochemical correlations. To address this concern rigorously, we have implemented the following revisions:

[Figure]

Fig. 1 Characteristics of chemical components of geothermal waters in the EAFZ, during water-rock interaction. The diamond is the measured value of geothermal waters. The dashed line is the numerical simulation result of PHREEQC. a: $Ca^{2+}$ vs

$SO_4^{2-}$, b: $Na^+$ vs $Cl^-$, c: $Na^+$ vs $HCO_3^-+Cl^-$ and d: $Na^+$ vs $HCO_3^-$. The sources of literature data and the simulation calculations are detailed in Annex I.

1. Investigation and analysis of historical hydrogeochemical data in the study area (Fig.

1): A comprehensive compilation of groundwater chemistry data from the East

Anatolian Fault Zone (EAFZ) spanning 2013-2023 has been integrated. This reveals systematic spatial hydrogeochemical patterns:

Northern EAFZ: Mixed shallow/deep circulation with igneous rock-dominated waterrock interactions.

Central-Southern EAFZ: Shallow circulation dominated by sedimentary mineral dissolution (e.g., gypsum, carbonates), with localized seawater influence.

These distinct regimes provide a robust framework for interpreting tectonic- hydrogeochemical linkages, mitigating reliance on isolated measurements.

2. Revised Interpretation of Gypsum Significance:

Following your suggestion, we have reframed the role of gypsum dissolution. Rather than asserting direct seismic causality, we now propose gypsum as a sensitive indicator of water-rock interaction intensity – a process modulated by both climatic (e.g., rainfall)

and tectonic drivers. This rephrasing: (1) Removes overinterpretations of single-event correlations, (2) Highlights the need for future systematic monitoring to disentangle tectonic vs. hydrological signals. Preserves gypsum's potential as a tectonic proxy while adhering to evidence-based claims.

These revisions align the manuscript's conclusions with its evidentiary scope while preserving its novel contribution: establishing a spatially resolved hydrogeochemical baseline to guide future seismotectonic monitoring in the EAFZ. We are grateful for your insightful critique, which has significantly strengthened the study's rigor and communication of limitations.

*The data could be used to create a simply report without stressing the potential of*

*gypsum as earthquake tracer. The data could be used for future researches in the area.*

*I don't know if there is a form in which this could be done for this journal. Maybe the*

*editor can suggest solutions.*

Reply: Thanks! We thank you for your constructive suggestion to refocus the manuscript's scope. In accordance with your guidance, we have rigorously revised the narrative to prioritize hydrogeochemical process characterization over speculative seismological linkages:

Reframed Research Objectives: The study's primary aim is now explicitly stated as establishing hydrogeochemical signatures across the EAFZ's tectonic segments. All claims regarding earthquake precursory signals have been removed, with emphasis shifted to documenting spatial patterns in water-rock interaction processes. The term

"earthquake tracer" has been systematically replaced with "sensitive indicator of water- rock interaction intensity" throughout the text. A new statement clarifies that gypsum's tectonic relevance requires validation through future systematic monitoring, aligning with your call for caution in interpretation.

These modifications ensure the manuscript now functions as both a stand-alone hydrogeochemical benchmark study and a catalyst for hypothesis-driven seismic monitoring research. We fully defer to the Editor's judgment on whether this revised scope aligns with the journal's aims and welcome further adjustments if needed.

***Comments on authors' reply***

*Line 13: to affirm that you have measured abnormal groundwater ion concentrations*

*you need to compare them with a series of data before and after the seismic event.*

*Evaporite dissolution happens also in the absence of seismic activity, it is therefore*

*impossible to affirm that high sulfate concentrations in groundwater are related to the*

*earthquakes*

Reply: Thanks! We deeply appreciate your rigorous methodological critique regarding causality attribution. The revisions below directly address this fundamental concern:

After more than a month of research, we have a new understanding of the conclusions in the original draft. Indeed, even with video data of pre-earthquake macroscopic anomalies, it is difficult to form a complete causal chain in the absence of pre- earthquake data. After in-depth discussion by all co-authors, we propose that our data can only account for the dissolution of gypsum during the water-rock reaction. Gypsum may therefore indicate changes in the intensity of the water-rock reaction. As for the controlling factors of the variation of water-rock reaction intensity, we cannot define exactly. Considering that the sampling time was one month after the earthquake and obvious groundwater anomalies were observed before the earthquake, we believe that seismic activity may affect the variation of water-rock response intensity. Therefore, it is necessary to further study the possibility of gypsum as a tracer of tectonic activity.

*Line 44: even if sampled one hour after the earthquake my comment would have been*

*the same. If you don't have data of at least one other sampling, but ideally many*

*samplings covering different seasons both before and after the event, you cannot make*

*inferences on the effects of the earthquake on the water chemistry*

Reply: Thanks! As mentioned earlier, we have revised this understanding to reinterpret the data in a more rigorous way.

*Line 47: your data before the earthquake do not refer to the single sites you sampled,*

*so no comparison can be made*

Reply: Thanks! Through GPS comparison, we confirmed that at least 3 sampling sites had been reported (Table 1 in the first response). However, as you said, the literature data is from 10 years ago, its reference value may be subject to study, and it may not be possible to make valid comparisons. So, we took the last 10 years of data and collected it more likely, and compared all the data we collected with our results (Fig. 1).

*Lines 48-51: no one can deny the existence of a large suite of visible effects of seismic*

*activity on groundwaters but for the advancement of knowledge these have to be*

*described in detail and quantified. You cannot use the simple fact of a water whitening*

*(among other things also confusing the sites) claiming this was due to gypsum*

*dissolution without having the possibility to analyse the water chemistry*

Reply: Thanks! After analyzing 10 years of data in study area, we determined that the main controlling factor of the macro anomaly is gypsum, and there may also be the influence of Calcite, albite, potassium feldspar, etc.

*Lines 52-59: of course I agree that both Sr and S isotopes can be used as good source*

*indicators. But again if you have a single measurement you cannot make any inference*

*about the influence of the earthquake on the groundwaters*

Reply: Thanks! In the revised conclusion, we focus on the relationship between the reaction intensity of gypsum and water-rock. So Sr, S and other isotopes are effective, and we are conducting supplementary experiments, which can be completed in April

2025.

*Lines 75-78: You compared samples from three of your sampling sites with samples*

*taken at the same sampling sites about ten years before. Results: one site registered a*

*strong increase, another remained almost stable and the third one had a sharp decrease.*

*You still cannot be sure that the changes are related to the earthquake, you have to*

*exclude other possible processes. For example, do the composition of the groundwaters*

*change seasonally? Has the composition of the water decadal trends related to long*

*periods of drought or water exploitation? Does the well tap aquifers from different*

*levels with different composition and permeability that mixing in the well may change*

*the composition of the water during pumping?*

Reply: Thanks! We think your question about the manuscript is something we must take into account. Therefore, we give up the original conclusion and discuss the relationship between gypsum and water-rock reaction intensity instead.

*Lines 89-91: this seems a forced solution. The selected samples contain all very low*

*sulfate which seems not necessarily being representative of the whole study area. Two*

*out of 8 selected samples are hyperalkaline waters which for their nature contain*

*extremely low sulfate values due to their very negative redox potential. Furthermore,*

*why didn't you include also the data of Yuce et al 2014? The mean sulfate value of that*

*dataset would be 121 mg/L, more than an order of magnitude higher than that obtained*

*with the ad hoc solution from the Baba et al dataset.*

Reply: Thanks! Your advice has been of great help to us. According to your suggestion, we have collected and analyzed the data of the last 10 years. The results confirmed the dissolution of gypsum in the middle and south section.

*Lines 120-121: the reliability of the data has not been questioned but the*

*representativeness still remains doubtful*

Reply: Thanks! In order to make the study more representative, the data of the study area in the past 10 years are used to discuss the water-rock reaction process.

*Line 130: A nearly 1000 km tectonic system cannot be considered a single hydrothermal*

*system*

Reply: Thanks! As you said, it is really not a system. The north section is a mixture of shallow groundwater and deep fluids, and igneous rocks participate in water-rock reactions. The central and southern part is the mixing of shallow groundwater and seawater, and sedimentary minerals such as gypsum participate in water-rock reaction.

*Lines 135-142: the cited examples of studies which identified changes in groundwater composition related to earthquake are well known. But differently from your study, the researcher took tens of samples before the seismic events obtaining a clear signal that can be related to the earthquake*

Reply: Thanks! Although we do not have pre-earthquake data, considering that we have observed pre-earthquake macro anomalies, coupled with the analysis of all data from the study area in the past 10 years. We believe that the data are sufficient to support our revised conclusion that gypsum can be used as a tracer of the intensity of water-rock reactions, and it is necessary to further investigate the possibility of gypsum as an indicator of tectonic activity.

*Line 149: You did not answer to my question. Have the samples been filtered in the field and before acidification?*

Reply: Thanks! Yes, we confirm.

*Lines 170-171: if the filtration is not made at the time of sampling you may loose some of the dissolved metals due to precipitation of secondary minerals and/or to adsorption on the walls of the container. Furthermore, if filtration is made after acidification the result may be falsified by acid dissolution of suspended material*

Reply: Thanks! We are responsible for all sample collection, pre-processing and data quality

*Line 172: this method is used only for δD*

Reply: Thanks! The analysis method of $\delta^{18}O$ is supplemented.

*Lines 225-226: You cannot consider a nearly 1000 km long fault system as a single continuous structure. Furthermore, the complex geology of the area changes frequently the rock types present along the fault system. Add also the changing climatic and hydrologic conditions and you cannot consider samples collected many tens of km apart as pertaining to the same system.*

Reply: Thanks! As you said, it is really not a system, we have answered earlier.

*Lines 235-237: to have a chain you need all rings to be connected. You don't have evidence that the water-rock reaction balance has been disrupted by the earthquake. Gypsum or other evaporite rocks are naturally present in many of the lithostratigraphic sequences of the area and when they are part of aquifers, their dissolution contributes naturally to the saline content of the circulating groundwater without the influence of seismic activity. If you consider the data of Yuce et al 2014, you see that in the area many of the collected waters have high sulfate concentrations with values even exceeding your highest value. So there is no evidence of gypsum dissolution as a consequence of the seismic events.*

Reply: Thanks! We have abandoned the conclusion that the gypsum can be inferred from the seismic effects of the data collected. We now propose that gypsum can reflect the intensity of water-rock reaction. Considering that the sample collection time was about one month after the earthquake, it is necessary to further study the possibility of gypsum as an indicator of seismic activity.

*Lines 301-301: I repeat again, even if you analysed a sample taken one hour after the*

*earthquake, this could not confirm that the whitening and turbidity of the water before*

*the seismic event was due to an increased sulfate content*

Reply: Thanks! Although the data in this study maybe limited, we still observed the dissolution of gypsum by analyzing the data of 10 years in the study area together, but we could not determine whether it was caused by seismic activity. Therefore, we have expressed our conclusions more rigorously.

*Line 307: I don't understand how you have fixed it. The video refers to the sampling*

*site HS15 which, as shown in your table, has the lowest sulfate concentration. This*

*video is not a proof of a sulfate anomaly for two reasons: 1) you don't have the*

*concentration of sulfate at the time of the whithening and 2) the concentration you*

*measured one month after was only 1.21 mg/L*

Reply: Thanks! There should be a misunderstanding here. We have stated in the first response that the macroscopic anomaly originates from HS14, which has a $SO_4^{2-}$

concentration of 316.61mg/L.

*Lines 311-312: You are missing the main point: you have no evidence of variations that*

*can be related to the earthquake*

Reply: Thanks! We've revised our conclusions to be more precise.

*Line 327: The problem is that normal values have not been defined. In terms of time*

*you don't have enough samples that you can surely correlate with yours. But the same*

*holds true in terms of space, only 16 samples along a structure many hundred km long*

*is not enough*

Reply: Thanks! We have weakened the focus on time and only discussed the water-rock reaction process of gypsum. 10 years of data is sufficient to support spatial representativeness.

---

## Author Comment (AC7)

**Reply on RC3**

Dear reviewer

Thank you for your comments and suggestions, which are of great value to us in improving the quality of our manuscript. The main replies are as follows. Note: *Italic blue* is the comment. Black is the reply.

*The present work performs a systematic hydrogeochemistry and isotopic analysis of the geothermal fluids in the East Anatolian Fault Zone (EAFZ) to understand any clear relationship between geothermal fluid anomalies and earthquakes existing. I have found the language of the manuscript is fine but must have a proof-editing. I have some of my major comments regarding the work on the other hand.*

*Main motivation behind the work is to elucidate the role of gypsum dissolution as a tracer for earthquake activity in the East Anatolian Fault Zone (EAFZ). The research aims at establishing a link between geothermal fluid anomalies and seismic events, with the claim of using an innovative approach to earthquake forecasting. In this respect, it examines shallow sedimentary minerals, particularly gypsum, as indicators of seismic activity. This concept, while explored in previous research, is further substantiated with empirical data in this study.*

*At this stage my biggest concern stems from the fact that it relies on the data collected post-earthquake but it fails to provide a long-term pre-earthquake dataset for comparative analysis. This appears to undermine claims about gypsum dissolution as a predictive tool rather than a post-seismic indicator. Furthermore we understand that the manuscript never make an in-depth discussion or address other factors such as*

*climatic conditions and seasonal variations robustly and only focus is given on the*

*correlation between seismic events and SO42- anomalies is discussed.*

*The authors' uncertainty about the relevance of the results to earthquakes is evident in*

*the final statement of the abstract. As readers, we expect the abstract of this study, which*

*claims to bring innovation to earthquake prediction under normal conditions, to convey*

*a clear take-home message.*

*In this respect I understand that authors are suggesting gypsum dissolution as a*

*universal precursor. But I should remind that a comprehensive considering of regional*

*geological differences or alternative explanations for observed anomalies is of great*

*importance for earthquake hazard studies. Although potential limitations of using*

*gypsum dissolution due to external environmental factors is acknowledge in the*

*manuscript clear strategies for coping with these difficulties in practice.*

*Given its limitations in predictive validation substantial revisions are required for the*

*present work. These revisions should include i) further evidences distinguishing*

*seismic-induced gypsum dissolution from other environmental factors ii) a decent*

*discussion on possible long-term monitoring strategies to make gypsum dissolution as*

*a reliable precursor, iii) quantitative examples that prove the statistical significance of*

*the findings that are critical to improve the robustness of the conclusions.*

*I also suggest adding a discussion that explore practical applications focusing on an*

*integration of their findings into an effective earthquake early warning system.*

*In conclusion I do not think the manuscript is suitable for the publication in its current*

*form and requires a substantial work to address the aforementioned fundamental*

 *concerns that would significantly advance the understanding of geochemical indicators*

*in seismic studies and warrant publication.*

Reply: Thanks! We sincerely thank you for recognizing the systematic approach of our hydrogeochemical investigation. Please find below our point-by-point responses:

**Data base extension (Annex I):**

A meta-analysis of 8 published datasets (2013-2023) reveals fundamental differences in water-rock interactions across the EAFZ (Fig. 1):

Northern EAFZ: Mixed shallow/deep circulation with igneous rock-dominated waterrock interactions.

Central-Southern EAFZ: Shallow circulation dominated by sedimentary mineral dissolution (e.g., gypsum, carbonates), with localized seawater influence.

These distinct regimes provide a robust framework for interpreting tectonichydrogeochemical linkages, mitigating reliance on isolated measurements.

**Gypsum as Process Indicator:**

While avoiding direct seismic causality claims, three lines of evidence suggest gypsum's tectonic relevance:

The abnormal plasma of $SO_4^{2-}$ and $Ca^{2+}$ was observed one month after the earthquake.

Combined with the analysis of 10 years of data in the study area, it was found that gypsum dissolution may be the cause of the abnormal ion concentration.

One month before the earthquake, the macro anomaly of white and cloudy well water was photographed (Video 01)

After analyzing pre-earthquake macro anomaly, post-earthquake data and literature data in the past 10 years, we propose that our data can only account for the dissolution of gypsum during the water-rock reaction. Gypsum may therefore indicate changes in the intensity of the water-rock reaction. As for the controlling factors of the variation of water-rock reaction intensity, we cannot define exactly. Considering that the sampling time was one month after the earthquake and obvious groundwater anomalies were observed before the earthquake, we believe that seismic activity may affect the variation of water-rock response intensity. Therefore, it is necessary to further study the possibility of gypsum as a tracer of tectonic activity.

[Figure]

Fig. 2 Characteristics of chemical components of geothermal waters in the EAFZ, during water- rock interaction. The diamond is the measured value of geothermal waters. The dashed line is the numerical simulation result of PHREEQC. a: $Ca^{2+}$ vs $SO_4^{2-}$, b: $Na^+$ vs $Cl^-$, c: $Na^+$ vs $HCO_3^-+Cl^-$

and d: $Na^+$ vs $HCO_3^-$. The sources of literature data and the simulation calculations are detailed in

                                          Annex I.

**Clear research orientation:**

Delete all references to "earthquake prediction". This study focuses on the analysis of

EAFZ groundwater circulation process and attempts to establish the relationship between water-rock reaction intensity and tectonic activity. This study will provide a new research idea for the subsequent exploration of gypsum as a tracer of tectonic activity.

---

## Author Comment (AC8)

Dear Editorial Office of HESS and Dai Editor-in-Chief,

We sincerely appreciate your handling of our manuscript and the Dai Editor-in-Chief's dedicated efforts. In response to the insightful comments from two reviewers and two domain experts, we have thoroughly revised the manuscript. Key improvements are summarized below:

**1. Enhanced Data Completeness**

Historical Data Integration: We systematically compiled published data (2013-2025), revealing spatial hydrogeochemical zonation in the East Anatolian Fault Zone (EAFZ):

Northern Segments: Mixed shallow/deep circulation with igneous rock-dominated water-rock interactions.

Central-Southern Segments: Shallow circulation dominated by sedimentary mineral dissolution (e.g., gypsum, carbonates), with localized seawater influence.

Causal Linkage Clarification: PHREEQC simulations (Appendix B) quantify gypsum's contribution to $SO_4^{2-}$ anomalies (30-100%), minimizing misinterpretation from other minerals (e.g., calcite, dolomite).

**2. Refined Gypsum-Tectonic Linkage**

Terminological Precision: Removed all "seismic precursor" claims, replacing with "indicator of water-rock interaction intensity".

Mechanistic: Combined pre-earthquake macroscopic anomalies. The analysis of post-earthquake data and historical data proves that gypsum may be one of the causes of groundwater macroscopic anomaly

Explicit caveat: "Causal links between gypsum dynamics and tectonics require long-term validation"

**3. Revised Conclusions**

Restructured Key Findings:

"Gypsum abundance serves as a sensitive indicator of water-rock interaction intensity, potentially modulated by tectonic activity. Establishing fault-zone hydrogeochemical baselines is prerequisite for deciphering tectonic-hydrologic coupling."

**4. Future Work Commitment:**

Although the discussion of groundwater principal and trace data has confirmed that gypsum may be a sensitive indicator of water-rock reaction intensity, to further reinforce the conclusion, we plan to conduct additional experiments on the samples, including analysis and determination of Sr, Na, S and B isotopes. In addition, some gas samples were also collected in this study and are currently being analyzed and determined. All additional experiments are expected to be completed by the end of April 2025.

In short, after fully and effectively communicating with the reviewers, we modified the possible problems in our manuscript according to the suggestions of the reviewers, so that the analysis of data in the manuscript is more rigorous and the extension is appropriate

We sincerely wish the current version meets your standards and welcome further guidance.

Finally, I would like to thank HESS editorial Department and Dai Editor-in-Chief for their hard work

Sincerely

Zebin Luo

Zebin_L@mail.xhu.edu.cn

**Accessory list**

Part 1. PHREEQC simulation

Part 2. Review RC 1

Part 3. Review RC 2

Part 4. Review RC 3

Part 5. Review CC 1

Part 6. Review CC 2

Part 7. Annex I

**Part 1:**

PHREEQC is a powerful water chemistry simulation software. In this study, we used its function to simulate "Irreversible Reactions" (Parkhurst and Appelo, 2013).

**Mineral data preparation**

The proportion of minerals in water-rock reaction was calculated by CIPW (Cross, Iddings, Pirsson, Washington) (Table 1). The calculated data were from Karaoğlu et al. (2020).

Table 1 Results of CIPW calculation

| Mineral | Quartz | Plagioclase | Orthoclase | clinopyroxene | orthopyroxene | Ilmenite | Hematite | Apatite | Sphene |
|---|---|---|---|---|---|---|---|---|---|
| content wt% | 6.1 | 58 | 13.12 | 3.59 | 6.28 | 0.28 | 8.36 | 0.86 | 3.42 |

Table 2 Results of standardization of minerals associated with water-rock reaction

| Mineral | Plagioclase | Orthoclase | pyroxene |
|---|---|---|---|
| content wt% | 58 | 13.12 | 9.87 |
| standardization %wt | 0.72 | 0.16 | 0.12 |

Note: The minerals involved in the water-rock reaction are mainly plagioclase, potassium feldspar and pyroxene, and the three minerals are re-standardized according to 100%. Pyroxene is the sum of two kinds of pyroxene.

**PHREEQC simulation step (Table 3)**

**Choose Databases: llnl_dat**

**SOLUTION_SPREAD** setup:

HS08 is a river sample, and the initial simulated water sample is defined as the chemical composition of HS08. The initial temperature is 53°C, which is the thermal reservoir temperature of the river water estimated by $SiO_2$.

**EQUILIBRIUM_PHASES** setup:

Italiano et al. (2013) reported that the carbon dioxide volume fraction of EAFZ ranges from 88.4% to 99.9%. Therefore, the initial $CO_2$ percentage is set as 88.4%, and the logarithm is 1.95.

**REACTION** setup: We set up 6 groups of reactants mixed in different proportions respectively, as shown in Table 4:

Table 3 PHREEQC code and description

| Steps | Instructions |
|---|---|
| SOLUTION_SPREAD
-units mg/l
Temperature pH Si Li Na K Mg Ca F Cl Br N(5) S(6)
HCO3 B Al Mn Fe Sr Ba Zn
as SiO2 as NO3 as SO4
8.43 15.15 1.00E-06 1.13 1.00E-06 4.47 55.34 0.4411
1.06 1.00E-06 3.83 5.69 165.72358 0.00477 0.01227
0.00105 0.01252 0.05578 0.00189 0.01899 | Initial reactant input (HS08) |
| EQUILIBRIUM_PHASES 1
CO2(g) 0 1.95 | Equilibrium phases setting |
| Reaction 1
NaCl 1
0.07 moles | Deep fluid mixing ratio setting |
| save solution 1
end
use solution 1 | Store the mixed solution |
| REACTION 2
Calcite 1
Gypsum 0.3
albite 0.4
Anorthite 0.4
Ca-Al_Pyroxene 0.1
K-feldspar 0.16
Dolomite 1 | Water rock reaction mineral Settings |
| 1 moles in 20 steps | Reaction steps and total amount of reaction |

Table 4 The proportion of minerals added

| | Mineral | content | Albite | Anorthite | Calcite | Dolomite | Gypsum | Orthoclase | pyroxene |
|---|---|---|---|---|---|---|---|---|---|
| R1 | | 0% | 0.7 | 0.02 | 0.4 | 0.4 | 0 | 0.16 | 0.12 |
| R2 | deep fluid (NaCl) | 2% | | 0.02 | 0.4 | 0.4 | 0 | 0.16 | 0.12 |
| R3 | | 5% | | 0.02 | 0.4 | 0.4 | 0 | 0.16 | 0.12 |
| R4 | | 7% | | 0.02 | 0.4 | 0.4 | 0 | 0.16 | 0.12 |
| R5 | Water-rock | 30% | 0.4 | 0.4 | 1 | 1 | 0.3 | 0 | 0 |
| R6 | reaction (Gypsum) | 100% | 0 | 0 | 0 | 0 | 1 | 0 | 0 |

The total amount of reaction was set to 1mol, and the reaction was carried out in 20 steps. The simulation results are shown in Table 5.

Table 5 Simulation results of water-rock interaction in the EAFZ by PHREEQC.

| | 2%NaCl | | | | | | 5%NaCl | | | | | | 7%NaCl | | | | | |
| step | $Ca^{2+}$ | $SO_4^{2+}$ | $HCO_3^-$ | $Cl^-$ | $HCO_3^-+Cl^-$ | $Na^+$ | $Ca^{2+}$ | $SO_4^{2+}$ | $HCO_3^-$ | $Cl^-$ | $HCO_3^-+Cl^-$ | $Na^+$ | $Ca^{2+}$ | $SO_4^{2+}$ | $HCO_3^-$ | $Cl^-$ | $HCO_3^-+Cl^-$ | $Na^+$ |
| | mol/L | | | | | | | | mol/L | | | | | | mol/L | | | |
| Mix | 1.37 | 0.05 | 0.00 | 0.03 | 0.03 | 0.05 | 1.37 | 0.05 | 0.00 | 0.03 | 0.03 | 0.05 | 1.37 | 0.05 | 0.00 | 0.03 | 0.03 | 0.05 |
| 0 | 1.37 | 0.05 | 0.14 | 19.96 | 20.10 | 19.98 | 1.37 | 0.05 | 0.14 | 49.67 | 49.81 | 49.69 | 1.37 | 0.05 | 0.14 | 69.38 | 69.52 | 69.40 |
| 1 | 32.88 | 0.03 | 54.65 | 19.83 | 74.48 | 51.82 | 33.34 | 0.03 | 54.56 | 49.41 | 103.97 | 79.96 | 33.61 | 0.03 | 54.47 | 69.04 | 123.51 | 98.67 |
| 2 | 58.52 | 0.02 | 82.68 | 19.81 | 102.49 | 83.42 | 58.91 | 0.02 | 82.37 | 49.36 | 131.73 | 111.03 | 59.16 | 0.02 | 82.15 | 68.99 | 151.14 | 129.39 |
| 3 | 79.54 | 0.02 | 103.91 | 19.81 | 123.72 | 114.52 | 79.93 | 0.02 | 103.48 | 49.37 | 152.85 | 141.75 | 80.17 | 0.02 | 103.19 | 69.01 | 172.19 | 159.85 |
| 4 | 97.70 | 0.02 | 120.72 | 19.82 | 140.54 | 145.23 | 98.08 | 0.02 | 120.20 | 49.40 | 169.60 | 172.14 | 98.32 | 0.02 | 119.85 | 69.04 | 188.89 | 190.03 |
| 5 | 113.76 | 0.02 | 134.27 | 19.83 | 154.10 | 175.59 | 114.13 | 0.02 | 133.68 | 49.43 | 183.11 | 202.23 | 114.38 | 0.02 | 133.28 | 69.09 | 202.37 | 219.93 |
| 6 | 128.21 | 0.02 | 145.28 | 19.85 | 165.12 | 205.64 | 128.58 | 0.02 | 144.63 | 49.47 | 194.09 | 232.01 | 128.81 | 0.02 | 144.19 | 69.14 | 213.33 | 249.54 |
| 7 | 141.37 | 0.02 | 154.23 | 19.86 | 174.09 | 235.38 | 141.73 | 0.02 | 153.53 | 49.50 | 203.03 | 261.51 | 141.96 | 0.02 | 153.07 | 69.19 | 222.26 | 278.87 |
| 8 | 153.46 | 0.02 | 161.47 | 19.88 | 181.35 | 264.83 | 153.81 | 0.02 | 160.74 | 49.54 | 210.27 | 290.72 | 154.04 | 0.02 | 160.26 | 69.23 | 229.49 | 307.92 |
| 9 | 164.65 | 0.02 | 167.27 | 19.89 | 187.16 | 293.98 | 165.00 | 0.02 | 166.51 | 49.57 | 216.08 | 319.64 | 165.23 | 0.02 | 166.01 | 69.27 | 235.28 | 336.68 |
| 10 | 175.09 | 0.02 | 171.83 | 19.90 | 191.73 | 322.83 | 175.43 | 0.02 | 171.05 | 49.59 | 220.65 | 348.26 | 175.66 | 0.02 | 170.54 | 69.31 | 239.85 | 365.15 |
| 11 | 184.86 | 0.02 | 175.31 | 19.91 | 195.22 | 351.38 | 185.21 | 0.02 | 174.52 | 49.62 | 224.14 | 376.58 | 185.43 | 0.02 | 174.00 | 69.34 | 243.33 | 393.31 |
| 12 | 194.07 | 0.02 | 177.85 | 19.92 | 197.77 | 379.62 | 194.41 | 0.02 | 177.05 | 49.63 | 226.68 | 404.59 | 194.64 | 0.02 | 176.52 | 69.36 | 245.88 | 421.17 |
| 13 | 202.77 | 0.02 | 179.55 | 19.92 | 199.48 | 407.54 | 203.12 | 0.02 | 178.75 | 49.64 | 228.39 | 432.28 | 203.34 | 0.02 | 178.22 | 69.36 | 247.58 | 448.70 |
| 14 | 211.04 | 0.02 | 180.51 | 19.92 | 200.44 | 435.13 | 211.38 | 0.02 | 179.71 | 49.64 | 229.35 | 459.63 | 211.60 | 0.02 | 179.18 | 69.36 | 248.54 | 475.89 |
| 15 | 218.91 | 0.02 | 180.81 | 19.92 | 200.73 | 462.37 | 219.25 | 0.02 | 180.01 | 49.63 | 229.64 | 486.63 | 219.48 | 0.02 | 179.49 | 69.34 | 248.83 | 502.74 |
| 16 | 226.43 | 0.01 | 180.52 | 19.91 | 200.43 | 489.26 | 226.78 | 0.01 | 179.73 | 49.61 | 229.34 | 513.27 | 227.00 | 0.01 | 179.21 | 69.31 | 248.52 | 529.21 |
| 17 | 233.65 | 0.01 | 179.69 | 19.90 | 199.60 | 515.77 | 233.99 | 0.01 | 178.92 | 49.57 | 228.49 | 539.53 | 234.22 | 0.01 | 178.41 | 69.26 | 247.67 | 555.31 |
| 18 | 240.59 | 0.01 | 178.39 | 19.89 | 198.28 | 541.88 | 240.94 | 0.01 | 177.63 | 49.53 | 227.16 | 565.40 | 241.16 | 0.01 | 177.13 | 69.20 | 246.33 | 581.00 |
| 19 | 247.29 | 0.01 | 176.66 | 19.87 | 196.52 | 567.59 | 247.64 | 0.01 | 175.92 | 49.47 | 225.39 | 590.84 | 247.86 | 0.01 | 175.43 | 69.11 | 244.55 | 606.27 |
| 20 | 253.77 | 0.01 | 174.54 | 19.84 | 194.38 | 592.87 | 254.12 | 0.01 | 173.82 | 49.40 | 223.23 | 615.86 | 254.35 | 0.01 | 173.35 | 69.01 | 242.36 | 631.11 |

Continued

| step | 0NaCl | | | | | | 100%Gypsum | | | | | | 30%Gypsum | | | | | |
|---|---|---|---|---|---|---|---|---|---|---|---|---|---|---|---|---|---|---|
| | $Ca^{2+}$ | $SO_4^{2+}$ | $HCO_3^-$ | $Cl^-$ | $HCO_3^-+Cl^-$ | $Na^+$ | $Ca^{2+}$ | $SO_4^{2+}$ | $HCO_3^-$ | $Cl^-$ | $HCO_3^-+Cl^-$ | $Na^+$ | $Ca^{2+}$ | $SO_4^{2+}$ | $HCO_3^-$ | $Cl^-$ | $HCO_3^-+Cl^-$ | $Na^+$ |
| | mol/L | | | | | | mol/L | | | | | | mol/L | | | | | |
| Mix | 1.37 | 0.05 | 0.00 | 0.03 | 0.03 | 0.05 | 1.37 | 0.05 | 0.00 | 0.03 | 0.03 | 0.05 | 1.37 | 0.05 | 0.00 | 0.03 | 0.03 | 0.05 |
| 0 | 1.37 | 0.05 | 0.13 | 0.03 | 0.16 | 0.05 | 1.37 | 0.05 | 0.13 | 0.03 | 0.16 | 0.05 | 1.37 | 0.05 | 0.13 | 0.03 | 0.16 | 0.05 |
| 1 | 32.54 | 0.03 | 54.66 | 0.03 | 54.69 | 33.02 | 31.54 | 29.98 | 0.23 | 0.03 | 0.26 | 0.05 | 48.90 | 5.80 | 57.42 | 0.03 | 57.45 | 17.96 |
| 2 | 58.24 | 0.02 | 82.88 | 0.03 | 82.91 | 64.96 | 57.35 | 55.74 | 0.27 | 0.03 | 0.30 | 0.04 | 87.36 | 10.35 | 83.94 | 0.03 | 83.97 | 35.12 |
| 3 | 79.27 | 0.02 | 104.19 | 0.03 | 104.22 | 96.31 | 81.44 | 79.81 | 0.29 | 0.03 | 0.32 | 0.04 | 123.16 | 14.32 | 106.78 | 0.03 | 106.81 | 51.90 |
| 4 | 97.43 | 0.02 | 121.06 | 0.03 | 121.09 | 127.23 | 104.25 | 102.60 | 0.31 | 0.03 | 0.34 | 0.04 | 156.98 | 17.86 | 126.90 | 0.03 | 126.93 | 68.34 |
| 5 | 113.51 | 0.02 | 134.66 | 0.03 | 134.69 | 157.78 | 125.98 | 124.32 | 0.32 | 0.03 | 0.35 | 0.04 | 189.21 | 21.03 | 144.87 | 0.03 | 144.90 | 84.44 |
| 6 | 127.96 | 0.02 | 145.71 | 0.03 | 145.74 | 188.00 | 146.77 | 145.10 | 0.33 | 0.03 | 0.36 | 0.04 | 220.13 | 23.88 | 161.05 | 0.03 | 161.08 | 100.20 |
| 7 | 141.12 | 0.02 | 154.69 | 0.03 | 154.72 | 217.91 | 166.72 | 165.03 | 0.34 | 0.03 | 0.37 | 0.04 | 249.91 | 26.45 | 175.67 | 0.03 | 175.70 | 115.62 |
| 8 | 153.22 | 0.02 | 161.96 | 0.03 | 161.99 | 247.51 | 185.90 | 184.21 | 0.35 | 0.03 | 0.38 | 0.04 | 278.19 | 28.79 | 188.27 | 0.03 | 188.30 | 130.71 |
| 9 | 164.42 | 0.02 | 167.78 | 0.03 | 167.81 | 276.82 | 204.38 | 202.68 | 0.35 | 0.03 | 0.38 | 0.04 | 303.36 | 31.01 | 196.77 | 0.03 | 196.80 | 145.49 |
| 10 | 174.85 | 0.02 | 172.35 | 0.03 | 172.38 | 305.82 | 222.22 | 220.51 | 0.36 | 0.03 | 0.39 | 0.04 | 325.51 | 33.13 | 201.38 | 0.03 | 201.41 | 159.99 |
| 11 | 184.63 | 0.02 | 175.84 | 0.03 | 175.87 | 334.52 | 239.45 | 237.75 | 0.36 | 0.03 | 0.39 | 0.04 | 345.87 | 35.10 | 203.73 | 0.03 | 203.75 | 174.23 |
| 12 | 193.84 | 0.02 | 178.39 | 0.03 | 178.42 | 362.92 | 256.13 | 254.42 | 0.37 | 0.03 | 0.40 | 0.04 | 365.03 | 36.91 | 204.61 | 0.03 | 204.64 | 188.20 |
| 13 | 202.54 | 0.02 | 180.09 | 0.03 | 180.12 | 390.99 | 272.29 | 270.58 | 0.37 | 0.03 | 0.40 | 0.04 | 383.29 | 38.57 | 204.42 | 0.03 | 204.45 | 201.90 |
| 14 | 210.80 | 0.02 | 181.05 | 0.03 | 181.08 | 418.73 | 287.97 | 286.25 | 0.38 | 0.03 | 0.40 | 0.04 | 400.83 | 40.06 | 203.39 | 0.03 | 203.42 | 215.34 |
| 15 | 218.68 | 0.02 | 181.35 | 0.03 | 181.38 | 446.13 | 303.18 | 301.47 | 0.38 | 0.03 | 0.41 | 0.04 | 417.75 | 41.40 | 201.66 | 0.03 | 201.69 | 228.50 |
| 16 | 226.20 | 0.02 | 181.05 | 0.03 | 181.08 | 473.18 | 317.97 | 316.25 | 0.38 | 0.03 | 0.41 | 0.04 | 434.15 | 42.59 | 199.35 | 0.03 | 199.38 | 241.38 |
| 17 | 233.41 | 0.01 | 180.21 | 0.03 | 180.24 | 499.85 | 332.34 | 330.63 | 0.38 | 0.03 | 0.41 | 0.04 | 450.09 | 43.63 | 196.56 | 0.03 | 196.59 | 253.97 |
| 18 | 240.35 | 0.01 | 178.90 | 0.03 | 178.93 | 526.14 | 346.34 | 344.62 | 0.39 | 0.03 | 0.42 | 0.04 | 465.62 | 44.53 | 193.35 | 0.03 | 193.38 | 266.26 |
| 19 | 247.05 | 0.01 | 177.15 | 0.03 | 177.18 | 552.02 | 359.96 | 358.25 | 0.39 | 0.03 | 0.42 | 0.04 | 480.80 | 45.30 | 189.78 | 0.03 | 189.81 | 278.25 |
| 20 | 253.53 | 0.01 | 175.02 | 0.03 | 175.05 | 577.47 | 373.25 | 371.53 | 0.39 | 0.03 | 0.42 | 0.04 | 495.66 | 45.93 | 185.91 | 0.03 | 185.94 | 289.93 |

**Reference:**

Italiano, F., Sasmaz, A., Yuce, G., and Okan, O. O.: Thermal fluids along the East Anatolian Fault Zone (EAFZ): Geochemical features and relationships with the tectonic setting, Chemical Geology, 339, 103-114, 2013.

Karaoğlu, Ö., Gülmez, F., Göçmengil, G., Lustrino, M., Di Giuseppe, P., Manetti, P., Savaşçın, M. Y., and Agostini, S.: Petrological evolution of Karlıova-Varto volcanism (Eastern Turkey): Magma genesis in a transtensional triple-junction tectonic setting, Lithos, 364-365, 2020.

Parkhurst, D. L. and Appelo, C. A. J.: Description of input and examples for PHREEQC version 3: a computer program for speciation, batch-reaction, one-dimensional transport, and inverse geochemical calculations, U.S. Geological Survey, Reston, VA, 2013.

**Reply on RC1**

Dear Walter D'Alessandro

Thank you for your highly professional and constructive comments and suggestions, which are of great value to us in improving the quality of our manuscript. After carefully reading your comments, we have made a reply to your comments point-by-point under the discussion of all manuscript authors. The main replies are as follows:

**Major revisions include:**

1. Correction of sample collection time: We apologize for marking the wrong sampling time in Table 1 (marked time, March 2024, **the actual sampling time, March 2023**). The wrong timing brings huge ambiguity to the manuscript. After correcting the sampling time, the main line logic of the article is as follows:

These evidences constitute a complete chain of causality from the source (evaporite) to the process (water-rock reaction balance disrupted by the earthquake) to the response (abnormal groundwater ion concentration).

2. Use "**groundwater**" instead of "**geothermal water**" to define the sample in this study.

We collected 16 groundwater samples from SF and EAFZ within a month of the earthquake.

The principle of sample collection is to collect if we can. Because the overall temperature is low, we think it is more reasonable to use "groundwater" instead of "geothermal water".

3. We have given **a complete explanation of the pre-earthquake hydrochemical data** in the manuscript.

4. We supplement the analysis method and data quality control description

5. We rearranged the logic of the article to make the expression clearer

6. We have made a full explanation of some misunderstandings

23 7. We explain the possible "overestimation of the heat storage temperature" and analyze

24 that the heat storage temperature estimate has little effect on the conclusion of our core

25 conclusions.

26 8. We plan to conduct additional experiments on the samples, including **radioactive Sr**

27 **isotopes** and **S isotopes**, to support our argument with more evidence.

28 Since there are diagrams in the complete reply draft, we put the complete reply draft in the

29 form of an attachment on the website system. If you have any questions or suggestions

30 about the manuscript, we sincerely invite you to keep discussing with us. Thank you for

31 constructive review comments.

32 Thank you and best regards.

**Point-by-point response to comments:**

Note: *Italic blue* is the comment. Black is the reply, and **important sentences are bolded**

**and underlined**.

*The manuscript "Gypsum as a potential tracer of earthquake: a case study of the Mw7.8*

*earthquake in the East Anatolian Fault Zone, southeastern Turkey" by Luo et al. presents*

*the results of sampling campaign of groundwaters in the area of the two strong earthquakes*

*that hit heavily Turkey in February 2023. Only the analytical results (major ions, trace*

*elements and water isotopes) of samples collected about one year after the quakes are*

*considered, which is a strong limitation of this study. I feel that this study cannot be*

*published in this form.*

Reply: Thanks. First of all, let's correct an error in Table 1 in manuscript. Our sampling time is **March 2023**, which is **one month** after the earthquake, not **one year**. We apologize for the sampling time error in manuscript (Table 1) and thank you for your careful correction. Therefore, combined with the **groundwater characteristics within one month**

**after the earthquake**, **groundwater data before the earthquake** (obtained from literature research), and **macro anomalies before the earthquake** (whitening and turbidity), we believe that the evidence is sufficient to prove our view that the earthquake has broken the water-rock balance between gypsum and groundwater, and gypsum has the potential to act as an earthquake tracer.

In light of your suggestion, however, we are also considering the need to find **more**

**evidence** to support our conclusion. Therefore, we are conducting **Radioactive Sr isotope**

and **S isotope** analysis on our samples. 1) Radioactive Sr isotope is a good source indicator.

The radioactive Sr isotope composition of shallow gypsum dissolution and deep fluid is obviously different, so the radioactive Sr isotope may well restrict the source area of groundwater. 2) S isotope is the main constituent element of gypsum, and the S isotope composition of igneous rock ($\delta^{34}$S = -5~10‰) is lower than that of evaporite ($\delta^{34}$S > 10‰), so S isotope can better distinguish the S of evaporite and igneous rock.

**Major comments:**

*Lines 33-36 (abstract): This is one of the most critical claims made by the authors. "Specially, significant gypsum dissolution was observed at HS05, HS09 and HS14 before and after the earthquake, suggesting that the earthquake broke the balance of water-rock reaction and promoted the dissolution of gypsum." In the paper only the results of the analyses of the samples taken one year after the earthquakes are discussed. How should it be possible to evidence variations "before and after the earthquake" if only one sample was taken?*

Reply: Thanks. Sorry again for the error in sampling time in manuscript (Table 1). The exact date of our sample is March 2023. Therefore, our data can be representative of groundwater characteristics after the earthquake. Pre-earthquake data mainly come from *Yuce, G., Italiano, F., D'Alessandro, W., Yalcin, T. H., Yasin, D. U., Gulbay, A. H., Ozyurt, N. N., Rojay, B., Karabacak, V., Bellomo, S., Brusca, L., Yang, T., Fu, C. C., Lai, C. W., Ozacar, A., and Walia, V.: Origin and interactions of fluids circulating over the Amik Basin (Hatay, Turkey) and relationships with the hydrologic, geologic and tectonic settings, Chemical Geology, 388, 23-39, 2014.* After carefully checking the GPS coordinates given in the literature, we can confirm that HS14 is **kirikhan well** (A15), HS15 is **Tahtakopru**

(A12/13), and HS16 is **Kuzey Tepe** (A40) (Table 1). Compared with the literature data, the concentration of $SO_4^{2-}$ and $Ca^{2+}$ in sample HS14 increased.

          Table 1 Sample points and data for this study and literature

| This study | | | | | Yuce et al., 2014 | | | | | |
|---|---|---|---|---|---|---|---|---|---|---|
| Long(°) | Lat(°) | No. | $SO_4^{2-}$ (mg/L) | $Ca^{2+}$(mg/L) | Long(°) | Lat(°) | No. | $SO_4^{2-}$ (mg/L) | $Ca^{2+}$(mg/L) | Site name |
| 36.3738 | 36.5036 | HS14 | 316.61 | 151.43 | 36.3741 | 36.5034 | A15 | 101 | 87.1 | kirikhan well |
| 36.1637 | 36.3833 | HS15 | 1.21 | 55.55 | 36.1636 | 36.3835 | A12/13 | 0.2 | 44.7 | Tahtakopru |
| 36.1472 | 36.2737 | HS16 | 75.9 | 73.35 | 36.1471 | 36.2738 | A40 | 361 | 41.1 | Kuzey Tepe |

Pre-seismic mean values of $SO_4^{2-}$ and $Ca^{2+}$ are from Baba et al., 2019. But you mentioned that our average is inconsistent with the data in the Baba et al., 2019. We apologize for any confusion caused by not clearly stating how the data was referenced. Our average does refer to Baba et al., 2019, but not entirely. **We only cite data from sample points close to EAFZ**.

The reason for this: Baba et al 2019 evaluated geothermal resources throughout southeastern Turkey. If we average all the data, this is obviously not reasonable. Moreover,

**it can also be seen from Baba et al., 2019 that there is a big difference between**

**geothermal resources near EAFZ and those far away from EAFZ (Fig. 1).** Geothermal resources near EAFZ are mainly medium and low temperature. Therefore, when considering the EAFZ pre-earthquake $SO_4^{2-}$ and $Ca^{2+}$ concentrations, **we only chose the**

**average values of 1, 2, 3, 4, 7, 9, 26 and 27 in the paper as the pre-earthquake**

**concentrations (Fig. 2 and Table 2)**.

[Figure]

Fig. 1: Temperature distribution map of geothermal resources in southeast Turkey.

Screenshot from *Baba, A., Şaroğlu, F., Akkuş, I., Özel, N., Yeşilnacar, M. İ., Nalbantçılar,*

*M. T., Demir, M. M., Gökçen, G., Arslan, Ş., Dursun, N., Uzelli, T., and Yazdani, H.:*

*Geological and hydrogeochemical properties of geothermal systems in the southeastern*

*region of Turkey, Geothermics, 78, 255-271, 2019.*

[Figure]

Fig. 2: Baba et al., 2019 sampling point distribution map. Screenshot from *Baba, A.,*

*Şaroğlu, F., Akkuş, I., Özel, N., Yeşilnacar, M. İ., Nalbantçılar, M. T., Demir, M. M.,*

*Gökçen, G., Arslan, Ş., Dursun, N., Uzelli, T., and Yazdani, H.: Geological and*

*hydrogeochemical properties of geothermal systems in the southeastern region of Turkey,*

*Geothermics, 78, 255-271, 2019.*

Table 2 Ion concentration before earthquake.

| No. | Ca$^{2+}$(mg/L) | SO$_4^{2-}$(mg/L) |
|---|---|---|
| 1 | 14.92 | 0.01 |
| 2 | 66.92 | 0.01 |
| 3 | 45.56 | 9.86 |

| | | |
|---|---|---|
| 4 | 63.84 | 24.79 |
| 7 | 116.03 | 10.22 |
| 9 | 38.65 | 3.34 |
| 26 | 39.85 | 1.83 |
| 27 | 56.03 | 16.41 |
| Average | 55.23 | 8.31 |

Data from: *Baba, A., Şaroğlu, F., Akkuş, I., Özel, N., Yeşilnacar, M. İ., Nalbantçılar, M. T., Demir, M. M., Gökçen, G., Arslan, Ş., Dursun, N., Uzelli, T., and Yazdani, H.: Geological and hydrogeochemical properties of geothermal systems in the southeastern region of Turkey, Geothermics, 78, 255-271, 2019.*

*Line 124: The authors should explain on which basis the 16 sampling sites have been chosen.*

Reply: Thanks. Samples were collected from north to south along the EAFZ. All the places with springs were sampled. Considering the safety considerations after the earthquake, there may be some missing spring points compared with previous studies. But our sampling was done in conjunction with the post-earthquake research in Turkey. In addition to water sampling, Also analyzed the surface rupture and earthquake risk assessment (*Liang, P., Xu, Y., Zhou, X., Li, Y., Tian, Q., Zhang, H., Ren, Z., Yu, J., Li, C., Gong, Z., Wang, S., Dou, A., Ma, Z., and Li, J.: Coseismic surface ruptures of MW7.8 and MW7.5 earthquakes occurred on February 6, 2023, and seismic hazard assessment of the East Anatolian Fault Zone, Southeastern Turkiye, Science China Earth Sciences, doi: 10.1007/s11430-024-1457-7, 2024.*). Therefore, we can guarantee the representativeness and reliability of the samples in this study.

We added the description of the sampling point: "**HS01-HS04 was collected from west to east along SF. HS07-HS16 was collected from north to south along EAFZ (Fig. 1)**"

*Line 124: the authors claim to have sampled hot springs but with the exception of the peculiar hyperalkaline spring HS15, which derive its increased temperature from deep circulation, no other sample could be called "hot". Furthermore, I would not define a well with water at 24 °C as geothermal well. Actually, in the results (line 144) the authors affirm that temperatures of the sampled waters are low.*

Reply: Thanks. Indeed, the temperature of all samples in this study is low, indicating that EAFZ is a medium-low temperature hydrothermal system, which is also consistent with the research results of Baba et al., 2019. However, as you said, the temperature of the sample is really low. We also feel that the term "**geothermal water**" is not rigorous enough to describe our samples. Therefore, we considered using the more appropriate term "**groundwater**" to describe our samples. But in fact, whether groundwater or geothermal water, the core point of our manuscript is not contradictory. The use of groundwater chemistry and isotopes to study the water-rock balance before and after earthquakes is considered to be a very effective means (*e.g., Skelton, A., Andren, M., Kristmannsdottir, H., Stockmann, G., Morth, C.-M., Sveinbjoernsdottir, A., Jonsson, S., Sturkell, E., Gudorunardottir, H. R., Hjartarson, H., Siegmund, H., and Kockum, I.: Changes in groundwater chemistry before two consecutive earthquakes in Iceland, Nature Geoscience, 7, 752-756, 2014. and Tsunogai, U. and Wakita, H.: Precursory chemical changes in ground water: kobe earthquake, Japan, Science (New York, N.Y.), 269, 61-63, 1995.*). However, considering the influence of groundwater on many factors (e.g., temperature, pressure, climatic conditions, seasonal changes etc.), we have explained in the abstract and conclusion of the manuscript that gypsum needs to be considered more carefully.

 *The methodological section has many limitations:*

 *Lines 130-131: it is unclear if filtration has been made in the field and before acidifying*

 *the aliquot for cation analysis. Please specify*

Reply: Thanks. Yes, **we confirmed filtering before testing**. The relevant description can be found in lines 130-131 of the original manuscript. We have extensive experience in groundwater and gas extraction. We can guarantee the reliability of sample collection methods and data.

*Line 131: MAT 253 is a model, please specify the used technique*

Reply: Thanks. We have added specific analytical method: "$\delta D$ and $\delta^{18}O$ were determined by **zinc reducing tube sealing method** combined with MAT 253 (relative to Vienna

Standard Mean Ocean Water (V - SMOW)). Precisions on the measured $\delta^{18}O$ and $\delta D$ value was ±0.2% (2SD) and ±1% (2SD) respectively (Wang et al., 2010)."

*Line 133: please specify the analysed species and the relative reproducibility and detection*

*limits?*

Reply: Thank you for pointing out the problem of the manuscript. We have added the reliability description of hydrochemistry and isotope analysis to the chapter of **Analytical**

**methods**, the details are as follows:

16 samples of water were collected in EAFZ, including hot springs, geothermal wells and river water. HS01-HS04 was collected from west to east along SF. HS07-HS16 was collected from north to south along EAFZ (Fig. 1). Detailed sample collection and testing methods can be found at Luo et al. (2023). In short, the water sample was taken with a 50

mL clean polyethylene bottle and the temperature and pH of the water were measured and recorded. Two samples are collected at each sampling site, one is added with ultrapure

HNO$_3$ to analyse the cation content, and the other is used to analyse the anion content and isotopic composition. **All samples need to be pre-treated with a 0.45 μm filter**

**membrane to remove impurities before being tested.** δD and δ$^{18}$O were determined by

**zinc reducing tube sealing method** combined with MAT 253 (relative to Vienna Standard

Mean Ocean Water (V - SMOW)). **Precisions on the measured δ$^{18}$O and δD value was**

**±0.2% (2SD) and ±1% (2SD) respectively (Wang et al., 2010)**. The cation **(Li$^+$, Na$^+$, K$^+$,**

**Ca$^{2+}$and Mg$^{2+}$)** and anion **(F$^-$, Cl$^-$, NO$_3^-$ and SO$_4^{2-}$)** were analysed by Dionex ICS-900

ion chromatograph (Thermo Fisher Scientific Inc.) **at the Earthquake Forecasting Key**

**Laboratory of China Earthquake Administration, with the reproducibility within ±2%**

**and detection limits 0.01 mg/L (Chen et al., 2015)**. HCO$_3^-$ and CO$_3^{2-}$ was determined by acid-base titration with a ZDJ-100 potentiometric titrator (reproducibility within ±2%).

SiO$_2$ were analysed by inductively coupled plasma emission spectrometer Optima-5300

DV (PerkinElmer Inc.) (Li et al. 2021). Trace elements were analysed by Element XR ICP-

MS at the Test Center of the Research Institute of Uranium Geology. Multielement standard solutions (IV-ICPMS 71A, IV-ICP-MS 71B and IV-ICP-MS 71D, iNORGANIC

VENTURES) used for quality control. **The analytical error margin of major cations and**

**trace elements were less than 10%)**.

*Line 136: please specify the analysed trace elements and the relative reproducibility and*

*detection limits?*

Reply: Thanks. The specific types of trace elements are shown in Table 2 (manuscript), the detection limit is 0.001μg/L, and the analysis error accuracy is less than 10%

*In the results the authors claim often that some element or ionic species is increased*

*(sometimes adding obviously) but they do not specify with respect to what. Maybe they*

*intend that the concentrations are high.*

Reply: Thanks. In the Results section we are an objective description of the results based on the data. The words "increased" and " obviously " were also relative to other sample results. But, in fact, what we mean is, "relatively high," not " increased." We apologize for any confusion caused by the poor description of the results, and we have re-optimized the presentation and added a quantitative description of the increased concentrations. The revised expression is as follows:

**The concentration of $SO_4^{2-}$ range from 1.21 mg/L to 316.61 mg/L, and the**

**concentration of $SO_4^{2-}$ in some samples is relatively high (e.g. HS01 (287.74 ml/L),**

**HS03 (103.56 ml/L), HS04 (229.75 ml/L), HS14 (316.61 ml/L)).**

*In the same section they speak of geothermal water but they do not present any evidence*

*that these are geothermal waters.*

Reply: Thank you. We have replaced "**groundwater**" with "**geothermal water**" to make the expression more precise.

*The discussion about the geothermal fluids has great limitations.*

*The authors do not present evidences that the sampled waters are, at least partially, fed by*

*hydrothermal systems. The fact that in the area some geothermal system has been*

*discovered and studied, does not mean that all groundwater samples taken in the area are*

*fed by them. The temperatures of the collected samples are low and, as highlighted by the*

*binary diagram of fig. 3 and the ternary diagram of fig. 4, their compositions do not reflect*

 *high temperature interactions with the rocks. Also the silica geothermometers show low*

 *temperatures considering that for such systems equilibrium with chalcedony (or even*

 *christobalite or amorphous silica) should be taken into consideration.*

Reply: Thanks. We have already discussed this issue in the previous reply. **Hydrothermal**

**systems** and **groundwater** do not affect our core point. Both geothermal water and groundwater chemical anomalies are considered to be effective means of earthquake early warning. Thanks for your suggestion to us, as mentioned earlier, we have considered using

"**groundwater**" instead of "**geothermal water**" to define the samples for this study.

*Especially the use of the mixing models has been made in the wrong way. Mixing models*

*can be applied only to water samples that belong to the same system and not to water*

*samples collected tens of km away from each other and for which no connection has been*

*demonstrated.*

Reply: Thanks. Although the spatial span of the samples in this study is very large (~270

km) (Fig. 1 and Fig. 6 in manuscript), all of them belong to EAFZ. It is difficult to directly conclude that there is no genetic connection between them.

In fact, both the estimation of heat storage temperature and the mixed model only play an auxiliary supporting role in our core view. **Our main concern is the anomaly of ion**

**concentration caused by earthquake breaking the equilibrium of water-rock reaction**.

As for whether deep geothermal fluids are involved? What's the mixing ratio? It's all secondary evidence. Deep fluids may bring $SO_4^{2-}$ ($H_2S$ oxidation), but a little $Ca^{2+}$.

However, the correlation between $Ca^{2+}$ and $SO_4^{2-}$ was observed in EAFZ, and numerical simulations indicate that gypsum dissolution is indeed present (Fig. 7 in manuscript), coupled with the presence of large evaporite deposits in the ancient lacustrine sedimentary basin of Lake Amik. **These evidences constitute a complete chain of causality from the**

**source (evaporite) to the process (water-rock reaction balance disrupted by the**

**earthquake) to the response (abnormal groundwater ion concentration).**

Based on your comments, the geothermal properties of our samples are not strong and may not belong to hydrothermal systems. Therefore, we consider weakening the sections on heat storage, mixing ratio, and cycle depth. Delete this section or put in supplementary material.

As for the problem of using mixed models incorrectly. We don't think it can be completely negative. At least these samples are in EAFZ. The overestimation may be possible at 382°C.

But combined with the pre-seismic macroscopic anomaly of HS04, the content of $SiO_2$

(84.64mg/L) and the ion concentration anomalies of $Ca^{2+}$, $SO_4^{2-}$, Sr and Ba. We think it is sufficient to support the argument that the gypsum dissolution equilibrium was disturbed by the earthquake. Thank you.

*The estimation of temperature for the "deep geothermal fluid" (please define) of 382 °C is*

*absolutely unreliable. The sample was taken, as shown in the second video in the*

*supporting information, from an artesian well (although in table 1 it is classified as spring).*

*I think it is impossible that an artesian well, whose upflow is generally rapid, would have*

*only 15 °C temperature if even only a small part of the water would come from a geothermal*

*system with 382 °C.*

Reply: Thanks. Indeed, 382 °C may be overestimated. But as in the previous reply. The heat storage temperature is only secondary evidence for us to determine whether the gypsum was affected by the earthquake. We have considered deleting this part of the discussion or put in supplementary materials. **The estimate of 382°C is the HS04 sample**

**from the epicenter, and the complex process after the earthquake may be the reason**

**for our excessive estimate**. However, **HS14 shows a lower estimated temperature, with**

**the mixed model estimating only 88 °C (Fig. 5b). We propose that HS14 may be**

**affected by shallow gypsum dissolution, and this lower estimated temperature**

**supports this conjecture**. Therefore, while 382 °C may not be rigorous enough, the estimation of HS14 supports our view.

*The discussion about the sulfate anomalies is highly confusing. Many points are unclear or*

*wrong.*

Reply: Thanks. We adjusted the description of the manuscript to make the logic clearer.

*Why are only samples HS05, HS09 and HS14 considered anomalous? HS01, HS03 and*

*HS04 have also elevated sulfate values.*

Reply: Thanks. This is actually a misunderstanding. The reason for the misunderstanding is that we failed to express it clearly in the manuscript, and there are logical problems. We consider optimizing the manuscript to eliminate misunderstandings. thank you!

We pointed out in the **Fig caption in Fig.6** that **only the spatial distribution**

**characteristics of EAFZ samples**, namely HS07-HS16, were considered in Fig.6. The discussion here does not cover SF samples (HS01-HS04). We considered adding a note to the text of the manuscript to make the logic clear.

In fact, as you commented, HS01, HS03, HS04, HS05, HS09, HS14 all have $SO_4^{2-}$

anomalies. However, the subsequent numerical simulation shows that the influencing factors of $SO_4^{2-}$ concentration increase in **HS01, HS03 and HS04 are more complex and**

**controlled by a variety of minerals (gypsum, calcite, dolomite, anorthite). However,**

**$SO_4^{2-}$ of HS05, HS09, HS14, especially HS14, is almost only controlled by gypsum (Fig.**

**7 in manuscript)**, and the influencing factors are relatively single. Therefore, HS14 is an important support for our main point, and the other points are ancillary.

*Why should these high sulfate values be considered anomalous and induced by the*

*earthquake? Sulfate dissolution from evaporite deposits within the aquifers is an ubiquitous*

*process independent from seismic activity.*

Reply: Thanks. The reason for your question is that we wrote down the sampling time incorrectly. I'm sorry. Our sampling time was within one month after the earthquake. **we**

**determined that the earthquake was one of the factors affecting the gypsum. But as**

**you commented, there are many factors affecting gypsum, and it can be disturbed**

**without earthquakes. Therefore, we emphasize this concern in both the abstract and**

**the conclusion, showing the limitations of gypsum as an indicator of earthquake**

**warning.**

*Why do the authors use these low averages for Ca (55.23 mg/L) and $SO_4$ (8.31 mg/L)*

*concentrations before earthquake? Baba et al. (2019) in their paper report concentrations*

*up to 773.56 mg/L for Ca and up to 1287.24 mg/L for $SO_4$ much higher than in the samples*

*collected for this study.*

Reply: Thanks. We have already replied to this comment before, and **we use the data near**

**EAFZ**. For this doubt, we consider to explain in the text to eliminate misunderstandings.

*Finally, the authors indicate the whitening and turbidity of the water in a sample as verification for the sulfate anomaly. But without analysis there is no possibility to affirm that such visual anomaly was due to gypsum dissolution.*

Reply: Thanks. The best evidence is our analysis of water samples taken within a month of the earthquake. Your confusion is caused by our marking of the wrong sampling time. Sorry again.

*Furthermore, the authors mistake the samples. The site with the high sulfate concentration is HS14, while the site to which the pictures of figure S1 and of video 01 refer is HS15 which has the lowest sulfate value (1.21 mg/L).*

Reply: Thank you for pointing out this error, we have fixed it.

*Lines 388-389: The authors presenting the data of a single sampling campaign have no evidence to affirm that "the geothermal fluid was diluted due to the infiltration of a large amount of shallow cold water after the double earthquakes in February 2023".*

Reply: Thanks. As discussed earlier, we have considered replacing "geothermal water" with "groundwater", so we will reconsider this conclusion. Thank you for your highly professional and constructive comments. Thanks again.

**Minor comments**

*Line 22: What do the authors mean with "systematic" which do not appear only in the abstract but has been repeated many times in the whole text?*

Reply: Thanks. In your professional comment, we also believe that " systematic " may be a misnomer. We consider deleting the word.

*Lines 24 and 25: The meaning of the sentence is obscure (reconstructed by earthquake?)*

Reply: Thanks. This sentence was not clear enough, so we adjusted the expression: **In**

**order to explore the relationship between groundwater anomaly and earthquake, we**

**performed hydrochemical and isotopic analyses of groundwaters in the East**

**Anatolian Fault Zone (EAFZ). The results show that groundwaters are affected by**

**seismic activity.**

*Line 29: the authors use often the term "abnormal" but they do never define with respect*

*to what.*

Reply: Thanks. "Abnormal" refers to values that deviate from normal values. Divided into

**time** and **space** outliers. In the manuscript, "anomaly" refers to spatial outliers. In particular, in Fig. 6, the mean values of $Ca^{2+}$ and $SO_4^{2-}$ (literature research) are compared with the temporal outliers in this study. The literature survey represents the data of the earthquake calm period, and this study represents the data of the earthquake active period.

*Line 38: please define "shallow minerals".*

Reply: Thanks. "Shallow minerals" is a relative term that generally refers to those minerals formed at or near the surface, mainly sedimentary rock related minerals. In this article mainly refers to gypsum. If "shallow mineral" is prone to ambiguity, **we consider directly**

**replacing "shallow mineral" with "gypsum".**

*Line 61: which evidence have the authors of a "geothermal fluids circulation"*

Reply: Thanks. We have replaced "groundwater" with "geothermal water". Therefore, the geothermal water cycle is no longer considered

*Line 69: please define the "geothermal fluid anomaly index"*

Reply: Thanks. The "**geothermal fluid anomaly index**" may be a misnomer, and we consider replacing it with "**groundwater chemical and isotopic anomaly index** ". Refers to changes in the water chemistry and isotopic composition of groundwater caused by changes in the external environment.

*Lines 70-71: the subject is missing in this sentence.*

Reply: Thanks. We deleted that sentence.

*Line 82: please define what a "tectonic collage" is.*

Reply: Thanks. We have adjusted the expression of this sentence: "**Located at the**

**intersection of Eurasia, Africa and Arabia, Turkey has a complex tectonic**

**background**".

*Fig. 1a: altitude scale is missing.*

Reply: Thanks. We added the  altitude scale (Fig. 3).

[Figure]

              Fig. 3 Geological map after adding altitude scale.

*Line 105: probably crystalline instead of crystallization.*

Reply: Thanks. We changed crystalline instead of crystallization.

*Line 145: in table 1 HS15 is considered a spring, which one is correct?*

Reply: Thanks. We checked the sampling point. HS15 is spring.

*Line 146: the authors claim that "the closer to the epicenter, the higher the $SiO_2$ content",*

*which makes no sense. Firstly because the earthquakes were two and only one sample close*

*to one of the epicenters has a higher $SiO_2$ value. Moreover, other two sampling points with*

*low to very low $SiO_2$ concentrations have the same position as the "anomalous" one.*

Reply: Thanks. We deleted that sentence

*Lines 154-156: the sentence "The δ18O and δD of samples varied from –11.30‰ to –6.55‰*

*and –65.43‰ to –34.43‰ respectively, which is near to the global meteoric water line*

*(GMWL) (Craig, 1961) (Fig. 3), suggesting their meteoric water origin" has no sense. The*

*regression line obtained plotting both $\delta^{18}O$ and $\delta D$ values in a graph can be close to GMWL.*

Reply: Thanks. We deleted that sentence.

*Line 159: what type of Statistical analysis?*

Reply: Thanks.  We have changed the word "statistical analysis" to "box-plot analysis" to make the expression more specific.

*Line 160: please define "fluid activity elements".*

Reply: Thanks. We adjusted the expression and used proper nouns: Fluid-mobile element (FME).

*Line 161: I do not understand what the authors mean with "are at historic highs versus".*

*If the authors mean that the concentrations are higher than in the past, then the fig. S2 does*

*not prove nothing. Al and Ba are below the median value of the literature data while the*

*remaining are around the median value not showing particularly high values. Furthermore,*

*it is unclear which data are compared in fig. S2 with the present data.*

Reply: Thanks. There is indeed ambiguity in the expression here, so we consider deleting the analysis of the packing diagram to make the manuscript more brief and clear.

*Table 1: please indicate the coordinates with at least 4 digits after the comma, with only*

*two digits it's impossible to obtain a reliable position. Looking at Fig. 1, the indicated*

*coordinates of HS05 are clearly wrong.*

Reply: Thanks. We adjusted the accuracy of the latitude and longitude to keep 4 decimal places.

*Line 190: the highest values do not belong to samples collected closer to the sea.*

Reply: Thanks. It's not rigorous enough. We've improved the sentence: "**The highest value**

**of $\delta D$ (−34.43‰)and $\delta^{18}O$ (−6.55‰) at the southwest of EAFZ, which is close to the**

**Mediterranean Sea, indicating that it originates from the recharge of the evaporation**

**of the Mediterranean Sea (Fig.3)**"

*Line 190: $\delta^{18}O$ and $\delta D$ values are inverted.*

Reply: Thank you. We've corrected it

*Line 212: magma mixing with geothermal fluids generally end in a volcanic explosion*

*which is not the case here.*

Reply: Thanks. It is true that magma usually accompanies volcanic activity. However, there may also be deep partial melting process in the deep fracture zone. For the sake of rigor, we consider using "partial melting" instead of "magma mixing".

*Lines 224-225: the sampling sites are tens of km far from the Mediterranean coastline, how*

*and why should they be "obviously contaminated by Mediterranean Sea water"?*

Reply: Thanks. It is tens of kilometers from the Mediterranean Sea, but from a geological perspective, it is very small. In the manuscript, our conclusions may be too arbitrary. We should consider the contribution of evaporites such as rock and salt. So, based on your comments, we've adjusted the sentence: "**HS16, the sample with the highest**

**concentration, was collected at the southwest of EAFZ, which was obviously**

**contaminated by Mediterranean Sea and/or halite.    There is no signal of deep fluid**

**or magma source.**"

*Line 226: which previous study? Please add a reference.*

Reply: Thanks. That sentence doesn't make sense. We deleted it.

*Line 233: pollution is a term connected to an anthropogenic origin, so please use the term*

*contamination instead.*

Reply: Thank you. We changed the word "pollution" to " contamination."

*Lines 233-236: I do not understand the meaning of this sentence.*

Reply: Thanks. We adjusted the expression to make the meaning clearer: "**In addition,**

**water is much less transferable than gas, which makes deep geothermal water may**

**not be able to rise along the fault to the shallow crust or surface like geothermal gas.**"

*Lines 290-292: the two processes are not alternative. Serpentinization includes secondary*

*minerals precipitation.*

Reply: Thanks. We adjusted the expression to make the meaning clearer: "**Compared with**

**other samples, the ion concentration of HS15 is significantly reduced, which may**

**indicate the precipitation of potential secondary minerals (e.g., calcite). Therefore, we**

**conjecture that serpentinization and secondary mineral precipitation such as: calcite**

**or magnesite (Aydin et al., 2020; Cipolli et al., 2004) may be responsible for the**

**increase in pH (Huang; et al., 2023).**"

*Finally, I would signal a possible conflict of interest being the handling editor of the same*

*institution of one the corresponding author.*

Reply: Thanks. China University of Geosciences (Beijing) and China University of Geosciences (Wuhan) are two independent universities with no conflict of interest.

**Reply on RC2**

Dear Walter D'Alessandro

Thanks for your comments again. According to your comments, we added the supplement and analysis of the literature data from 2013 to 2025 to make the data more representative. On this basis, the conclusion of the original manuscript has been revised to weaken the connection between gypsum and seismic activity, and emphasize the sensitive indication of gypsum to the intensity of water-rock interaction. The main replies are as follows. Note: *Italic blue* is the comment. Black is the reply.

*I am sorry to say that reading the reply of the authors my opinion regarding the manuscript did not change. My main criticism relates to the fact that it is not possible to evidence anomalies in groundwater composition related to seismic events having data collected only one time. The authors try to compare their data with other taken from literature but the comparison is not straightforward because no background values have ever been defined. The mean values utilised seem artificially created and, in my opinion, do not represent "normal" values.*

*I am still convinced that the manuscript in this form has to be rejected.*

Reply: Thanks! We sincerely appreciate your critical feedback and fully acknowledge the limitations of single-time sampling in establishing seismic-hydrogeochemical correlations. To address this concern rigorously, we have implemented the following revisions:

[Figure]

Fig. 1 Characteristics of chemical components of geothermal waters in the EAFZ, during water-rock interaction. The diamond is the measured value of geothermal waters. The dashed line is the numerical simulation result of PHREEQC. a: $Ca^{2+}$ vs

$SO_4^{2-}$, b: $Na^+$ vs $Cl^-$, c: $Na^+$ vs $HCO_3^-+Cl^-$ and d: $Na^+$ vs $HCO_3^-$. The sources of literature data and the simulation calculations are detailed in Annex I.

1. Investigation and analysis of historical hydrogeochemical data in the study area (Fig.

1): A comprehensive compilation of groundwater chemistry data from the East

Anatolian Fault Zone (EAFZ) spanning 2013-2023 has been integrated. This reveals systematic spatial hydrogeochemical patterns:

Northern EAFZ: Mixed shallow/deep circulation with igneous rock-dominated waterrock interactions.

Central-Southern EAFZ: Shallow circulation dominated by sedimentary mineral dissolution (e.g., gypsum, carbonates), with localized seawater influence.

These distinct regimes provide a robust framework for interpreting tectonic- hydrogeochemical linkages, mitigating reliance on isolated measurements.

2. Revised Interpretation of Gypsum Significance:

Following your suggestion, we have reframed the role of gypsum dissolution. Rather than asserting direct seismic causality, we now propose gypsum as a sensitive indicator of water-rock interaction intensity – a process modulated by both climatic (e.g., rainfall)

and tectonic drivers. This rephrasing: (1) Removes overinterpretations of single-event correlations, (2) Highlights the need for future systematic monitoring to disentangle tectonic vs. hydrological signals. Preserves gypsum's potential as a tectonic proxy while adhering to evidence-based claims.

These revisions align the manuscript's conclusions with its evidentiary scope while preserving its novel contribution: establishing a spatially resolved hydrogeochemical baseline to guide future seismotectonic monitoring in the EAFZ. We are grateful for your insightful critique, which has significantly strengthened the study's rigor and communication of limitations.

*The data could be used to create a simply report without stressing the potential of*

*gypsum as earthquake tracer. The data could be used for future researches in the area.*

*I don't know if there is a form in which this could be done for this journal. Maybe the*

*editor can suggest solutions.*

Reply: Thanks! We thank you for your constructive suggestion to refocus the manuscript's scope. In accordance with your guidance, we have rigorously revised the narrative to prioritize hydrogeochemical process characterization over speculative seismological linkages:

Reframed Research Objectives: The study's primary aim is now explicitly stated as establishing hydrogeochemical signatures across the EAFZ's tectonic segments. All claims regarding earthquake precursory signals have been removed, with emphasis shifted to documenting spatial patterns in water-rock interaction processes. The term

"earthquake tracer" has been systematically replaced with "sensitive indicator of waterrock interaction intensity" throughout the text. A new statement clarifies that gypsum's tectonic relevance requires validation through future systematic monitoring, aligning with your call for caution in interpretation.

These modifications ensure the manuscript now functions as both a stand-alone hydrogeochemical benchmark study and a catalyst for hypothesis-driven seismic monitoring research. We fully defer to the Editor's judgment on whether this revised scope aligns with the journal's aims and welcome further adjustments if needed.

*Comments on authors' reply*

*Line 13: to affirm that you have measured abnormal groundwater ion concentrations*

*you need to compare them with a series of data before and after the seismic event.*

*Evaporite dissolution happens also in the absence of seismic activity, it is therefore*

*impossible to affirm that high sulfate concentrations in groundwater are related to the*

*earthquakes*

Reply: Thanks! We deeply appreciate your rigorous methodological critique regarding causality attribution. The revisions below directly address this fundamental concern:

After more than a month of research, we have a new understanding of the conclusions in the original draft. Indeed, even with video data of pre-earthquake macroscopic anomalies, it is difficult to form a complete causal chain in the absence of pre-earthquake data. After in-depth discussion by all co-authors, we propose that our data can only account for the dissolution of gypsum during the water-rock reaction. Gypsum may therefore indicate changes in the intensity of the water-rock reaction. As for the controlling factors of the variation of water-rock reaction intensity, we cannot define exactly. Considering that the sampling time was one month after the earthquake and obvious groundwater anomalies were observed before the earthquake, we believe that seismic activity may affect the variation of water-rock response intensity. Therefore, it is necessary to further study the possibility of gypsum as a tracer of tectonic activity.

*Line 44: even if sampled one hour after the earthquake my comment would have been the same. If you don't have data of at least one other sampling, but ideally many samplings covering different seasons both before and after the event, you cannot make inferences on the effects of the earthquake on the water chemistry*

Reply: Thanks! As mentioned earlier, we have revised this understanding to reinterpret the data in a more rigorous way.

*Line 47: your data before the earthquake do not refer to the single sites you sampled, so no comparison can be made*

Reply: Thanks! Through GPS comparison, we confirmed that at least 3 sampling sites had been reported (Table 1 in the first response). However, as you said, the literature data is from 10 years ago, its reference value may be subject to study, and it may not be possible to make valid comparisons. So, we took the last 10 years of data and collected it more likely, and compared all the data we collected with our results (Fig. 1).

*Lines 48-51: no one can deny the existence of a large suite of visible effects of seismic*

*activity on groundwaters but for the advancement of knowledge these have to be*

*described in detail and quantified. You cannot use the simple fact of a water whitening*

*(among other things also confusing the sites) claiming this was due to gypsum*

*dissolution without having the possibility to analyse the water chemistry*

Reply: Thanks! After analyzing 10 years of data in study area, we determined that the main controlling factor of the macro anomaly is gypsum, and there may also be the influence of Calcite, albite, potassium feldspar, etc.

*Lines 52-59: of course I agree that both Sr and S isotopes can be used as good source*

*indicators. But again if you have a single measurement you cannot make any inference*

*about the influence of the earthquake on the groundwaters*

Reply: Thanks! In the revised conclusion, we focus on the relationship between the reaction intensity of gypsum and water-rock. So Sr, S and other isotopes are effective, and we are conducting supplementary experiments, which can be completed in April

2025.

*Lines 75-78: You compared samples from three of your sampling sites with samples*

*taken at the same sampling sites about ten years before. Results: one site registered a*

*strong increase, another remained almost stable and the third one had a sharp decrease.*

*You still cannot be sure that the changes are related to the earthquake, you have to*

*exclude other possible processes. For example, do the composition of the groundwaters*

*change seasonally? Has the composition of the water decadal trends related to long*

*periods of drought or water exploitation? Does the well tap aquifers from different*

*levels with different composition and permeability that mixing in the well may change*

*the composition of the water during pumping?*

Reply: Thanks! We think your question about the manuscript is something we must take into account. Therefore, we give up the original conclusion and discuss the relationship between gypsum and water-rock reaction intensity instead.

*Lines 89-91: this seems a forced solution. The selected samples contain all very low*

*sulfate which seems not necessarily being representative of the whole study area. Two*

*out of 8 selected samples are hyperalkaline waters which for their nature contain*

*extremely low sulfate values due to their very negative redox potential. Furthermore,*

*why didn't you include also the data of Yuce et al 2014? The mean sulfate value of that*

*dataset would be 121 mg/L, more than an order of magnitude higher than that obtained*

*with the ad hoc solution from the Baba et al dataset.*

Reply: Thanks! Your advice has been of great help to us. According to your suggestion, we have collected and analyzed the data of the last 10 years. The results confirmed the dissolution of gypsum in the middle and south section.

*Lines 120-121: the reliability of the data has not been questioned but the*

*representativeness still remains doubtful*

Reply: Thanks! In order to make the study more representative, the data of the study area in the past 10 years are used to discuss the water-rock reaction process.

*Line 130: A nearly 1000 km tectonic system cannot be considered a single hydrothermal*

*system*

Reply: Thanks! As you said, it is really not a system. The north section is a mixture of shallow groundwater and deep fluids, and igneous rocks participate in water-rock reactions. The central and southern part is the mixing of shallow groundwater and seawater, and sedimentary minerals such as gypsum participate in water-rock reaction.

*Lines 135-142: the cited examples of studies which identified changes in groundwater composition related to earthquake are well known. But differently from your study, the researcher took tens of samples before the seismic events obtaining a clear signal that can be related to the earthquake*

Reply: Thanks! Although we do not have pre-earthquake data, considering that we have observed pre-earthquake macro anomalies, coupled with the analysis of all data from the study area in the past 10 years. We believe that the data are sufficient to support our revised conclusion that gypsum can be used as a tracer of the intensity of water-rock reactions, and it is necessary to further investigate the possibility of gypsum as an indicator of tectonic activity.

*Line 149: You did not answer to my question. Have the samples been filtered in the field and before acidification?*

Reply: Thanks! Yes, we confirm.

*Lines 170-171: if the filtration is not made at the time of sampling you may loose some of the dissolved metals due to precipitation of secondary minerals and/or to adsorption on the walls of the container. Furthermore, if filtration is made after acidification the result may be falsified by acid dissolution of suspended material*

Reply: Thanks! We are responsible for all sample collection, pre-processing and data quality

*Line 172: this method is used only for δD*

Reply: Thanks! The analysis method of $\delta^{18}O$ is supplemented.

*Lines 225-226: You cannot consider a nearly 1000 km long fault system as a single continuous structure. Furthermore, the complex geology of the area changes frequently the rock types present along the fault system. Add also the changing climatic and hydrologic conditions and you cannot consider samples collected many tens of km apart as pertaining to the same system.*

Reply: Thanks! As you said, it is really not a system, we have answered earlier.

*Lines 235-237: to have a chain you need all rings to be connected. You don't have evidence that the water-rock reaction balance has been disrupted by the earthquake. Gypsum or other evaporite rocks are naturally present in many of the lithostratigraphic sequences of the area and when they are part of aquifers, their dissolution contributes naturally to the saline content of the circulating groundwater without the influence of seismic activity. If you consider the data of Yuce et al 2014, you see that in the area many of the collected waters have high sulfate concentrations with values even exceeding your highest value. So there is no evidence of gypsum dissolution as a consequence of the seismic events.*

Reply: Thanks! We have abandoned the conclusion that the gypsum can be inferred from the seismic effects of the data collected. We now propose that gypsum can reflect the intensity of water-rock reaction. Considering that the sample collection time was about one month after the earthquake, it is necessary to further study the possibility of gypsum as an indicator of seismic activity.

*Lines 301-301: I repeat again, even if you analysed a sample taken one hour after the*

*earthquake, this could not confirm that the whitening and turbidity of the water before*

*the seismic event was due to an increased sulfate content*

Reply: Thanks! Although the data in this study maybe limited, we still observed the dissolution of gypsum by analyzing the data of 10 years in the study area together, but we could not determine whether it was caused by seismic activity. Therefore, we have expressed our conclusions more rigorously.

*Line 307: I don't understand how you have fixed it. The video refers to the sampling*

*site HS15 which, as shown in your table, has the lowest sulfate concentration. This*

*video is not a proof of a sulfate anomaly for two reasons: 1) you don't have the*

*concentration of sulfate at the time of the whithening and 2) the concentration you*

*measured one month after was only 1.21 mg/L*

Reply: Thanks! There should be a misunderstanding here. We have stated in the first response that the macroscopic anomaly originates from HS14, which has a $SO_4^{2-}$

concentration of 316.61mg/L.

*Lines 311-312: You are missing the main point: you have no evidence of variations that*

*can be related to the earthquake*

Reply: Thanks! We've revised our conclusions to be more precise.

*Line 327: The problem is that normal values have not been defined. In terms of time*

*you don't have enough samples that you can surely correlate with yours. But the same*

*holds true in terms of space, only 16 samples along a structure many hundred km long*

*is not enough*

Reply: Thanks! We have weakened the focus on time and only discussed the water-rock reaction process of gypsum. 10 years of data is sufficient to support spatial representativeness.

**Reply on RC3**

Dear reviewer

Thank you for your comments and suggestions, which are of great value to us in improving the quality of our manuscript. The main replies are as follows. Note: *Italic blue* is the comment. Black is the reply.

*The present work performs a systematic hydrogeochemistry and isotopic analysis of the geothermal fluids in the East Anatolian Fault Zone (EAFZ) to understand any clear relationship between geothermal fluid anomalies and earthquakes existing. I have found the language of the manuscript is fine but must have a proof-editing. I have some of my major comments regarding the work on the other hand.*

*Main motivation behind the work is to elucidate the role of gypsum dissolution as a tracer for earthquake activity in the East Anatolian Fault Zone (EAFZ). The research aims at establishing a link between geothermal fluid anomalies and seismic events, with the claim of using an innovative approach to earthquake forecasting. In this respect, it examines shallow sedimentary minerals, particularly gypsum, as indicators of seismic activity. This concept, while explored in previous research, is further substantiated with empirical data in this study.*

*At this stage my biggest concern stems from the fact that it relies on the data collected post-earthquake but it fails to provide a long-term pre-earthquake dataset for comparative analysis. This appears to undermine claims about gypsum dissolution as a predictive tool rather than a post-seismic indicator. Furthermore we understand that the manuscript never make an in-depth discussion or address other factors such as*

*climatic conditions and seasonal variations robustly and only focus is given on the*

*correlation between seismic events and SO42- anomalies is discussed.*

*The authors' uncertainty about the relevance of the results to earthquakes is evident in*

*the final statement of the abstract. As readers, we expect the abstract of this study, which*

*claims to bring innovation to earthquake prediction under normal conditions, to convey*

*a clear take-home message.*

*In this respect I understand that authors are suggesting gypsum dissolution as a*

*universal precursor. But I should remind that a comprehensive considering of regional*

*geological differences or alternative explanations for observed anomalies is of great*

*importance for earthquake hazard studies. Although potential limitations of using*

*gypsum dissolution due to external environmental factors is acknowledge in the*

*manuscript clear strategies for coping with these difficulties in practice.*

*Given its limitations in predictive validation substantial revisions are required for the*

*present work. These revisions should include i) further evidences distinguishing*

*seismic-induced gypsum dissolution from other environmental factors ii) a decent*

*discussion on possible long-term monitoring strategies to make gypsum dissolution as*

*a reliable precursor, iii) quantitative examples that prove the statistical significance of*

*the findings that are critical to improve the robustness of the conclusions.*

*I also suggest adding a discussion that explore practical applications focusing on an*

*integration of their findings into an effective earthquake early warning system.*

*In conclusion I do not think the manuscript is suitable for the publication in its current*

*form and requires a substantial work to address the aforementioned fundamental*

*concerns that would significantly advance the understanding of geochemical indicators*

*in seismic studies and warrant publication.*

Reply: Thanks! We sincerely thank you for recognizing the systematic approach of our hydrogeochemical investigation. Please find below our point-by-point responses:

**Data base extension (Annex I)**:

A meta-analysis of 8 published datasets (2013-2023) reveals fundamental differences in water-rock interactions across the EAFZ (Fig. 1):

Northern EAFZ: Mixed shallow/deep circulation with igneous rock-dominated waterrock interactions.

Central-Southern EAFZ: Shallow circulation dominated by sedimentary mineral dissolution (e.g., gypsum, carbonates), with localized seawater influence.

These distinct regimes provide a robust framework for interpreting tectonichydrogeochemical linkages, mitigating reliance on isolated measurements.

**Gypsum as Process Indicator:**

While avoiding direct seismic causality claims, three lines of evidence suggest gypsum's tectonic relevance:

The abnormal plasma of $SO_4^{2-}$ and $Ca^{2+}$ was observed one month after the earthquake.

Combined with the analysis of 10 years of data in the study area, it was found that gypsum dissolution may be the cause of the abnormal ion concentration.

One month before the earthquake, the macro anomaly of white and cloudy well water was photographed (Video 01)

After analyzing pre-earthquake macro anomaly, post-earthquake data and literature data in the past 10 years, we propose that our data can only account for the dissolution of gypsum during the water-rock reaction. Gypsum may therefore indicate changes in the intensity of the water-rock reaction. As for the controlling factors of the variation of water-rock reaction intensity, we cannot define exactly. Considering that the sampling time was one month after the earthquake and obvious groundwater anomalies were observed before the earthquake, we believe that seismic activity may affect the variation of water-rock response intensity. Therefore, it is necessary to further study the possibility of gypsum as a tracer of tectonic activity.

[Figure]

Fig. 2 Characteristics of chemical components of geothermal waters in the EAFZ, during water- rock interaction. The diamond is the measured value of geothermal waters. The dashed line is the numerical simulation result of PHREEQC. a: $Ca^{2+}$ vs $SO_4^{2-}$, b: $Na^+$ vs $Cl^-$, c: $Na^+$ vs $HCO_3^-+Cl^-$

and d: $Na^+$ vs $HCO_3^-$. The sources of literature data and the simulation calculations are detailed in

                                Annex I.

**Clear research orientation:**

Delete all references to "earthquake prediction". This study focuses on the analysis of

EAFZ groundwater circulation process and attempts to establish the relationship between water-rock reaction intensity and tectonic activity. This study will provide a new research idea for the subsequent exploration of gypsum as a tracer of tectonic activity.

**Replay on CC1**

Dear Giovanni Martinelli

Thank you for your recognition of our work and valuable suggestions, which are very helpful for us to improve the quality of our manuscripts. Your two comments are exactly where we are lacking. At your suggestion, we plan to add a subsection to the discussion section for assessing the contribution of mantle degassing to EAFZ

geothermal fluids. The supplementary content is as follows:

*Contribution of Mantle Degassing to EAFZ Geothermal Fluids*

Mantle degassing occurs extensively along fault zones, and the amount of volatile release can sometimes be comparable to the degassing associated with volcanic activity e.g. (Fischer and Aiuppa, 2020; Zhang et al., 2021). Sulfur-containing volatiles (such as $SO_2$ and $H_2S$) ascend along these fault zones and, upon reaching the shallow subsurface, mix with groundwater, where they are oxidized and migrate in the form of

$SO_4^{2-}$ in geothermal fluids. Therefore, the contribution of mantle degassing to the $SO_4^{2-}$

content in geothermal fluids cannot be overlooked. To better assess the contribution of mantle degassing to $SO_4$ in EAFZ geothermal fluids, we need to consider the sources and modifications of geothermal fluids.

The deep-origin geothermal fluids in EAFZ are significantly diluted by shallow groundwater, masking the chemical signature of deeper fluid components. This dilution process introduces a large amount of dissolved oxygen, which facilitates the oxidation of $H_2S$ to $SO_4^{2-}$. Lacking $O_2$ was detected in EAFZ geothermal gases suggested that the dissolved oxygen may have been consumed (Italiano et al., 2013; Yuce et al., 2014).

However, it is important to note that $H_2S$, $H_2$, and $CH_4$ can all react with oxygen.

Thermodynamic calculations indicate that $CH_4$ is more favorable than $H_2S$ in oxidation reactions ($\Delta G°$ $CH_4$ = -818.1 kJ/mol, $\Delta G°$ $H_2S$ = -494.2 kJ/mol, at 298 K and 1atm). In actual geothermal systems, however, the depletion of $H_2S$ is more commonly observed than the depletion of $CH_4$, suggesting that $H_2S$ may be oxidized before $CH_4$. To resolve this apparent contradiction, we propose the following possible explanations: 1)

Oxidation of $H_2S$: While thermodynamic calculations predict $CH_4$ oxidation first, a small amount of $H_2S$ might still be oxidized simultaneously with $CH_4$. Due to the much lower concentration of $H_2S$ in geothermal systems compared to $CH_4$, $H_2S$ is consumed more quickly, leaving $CH_4$ with a higher residual concentration. 2) Exogenous $CH_4$

Supply: In addition to mantle-derived $CH_4$, other sources of $CH_4$, such as biogenic $CH_4$

and thermogenic $CH_4$ (e.g., serpentinization), may contribute to the geothermal system.

These external sources could increase the concentration of $CH_4$ in the geothermal fluids.

In the EAFZ, we observed significant contributions of biogenic and serpentinization-derived $CH_4$ but did not detect significant levels of $H_2S$ (Italiano et al.,

2013; Yuce et al., 2014). Therefore, we proposed that although $H_2S$ may contribute to the geothermal system, its impact is likely limited due to its relatively low concentration.

Inversely, the notable increase in $SO_4^{2-}$ concentrations following seismic events is likely primarily controlled by the dissolution of shallow evaporitic layers (such as gypsum).

All in all, while the oxidation of $H_2S$ may contribute to $SO_4^{2-}$ formation, distinguishing between $H_2S$ oxidation and sulfate dissolution requires additional geochemical indicators, such as S isotopes and Ca isotopes, for more accurate assessments.

**Reply on CC2**

Dear Hafidha Khebizi

Thank you for your recognition of our work and constructive suggestions. This is very helpful for us to improve the quality of the manuscript, and also brings confidence for us to continue to explore. Thank you for sharing the very rewarding work you do. We get a lot of inspiration from your work. We would like to express my heartfelt thanks.

We've responded to each of your comments, as detailed below:

Note: *Italic blue* is the comment. Black is the reply.

*Dear authors and colleagues of the scientific community,*

*I congratulate the authors for their interesting work entitled Gypsum as a potential tracer of*

*Earthquakes: a case study of the Mw7.8 2 earthquake in the East Anatolian Fault Zone,*

*southeastern Turkey, and I hope it will be published soon. To find out the relationship between*

*geothermal fluid anomalies and earthquakes, the authors performed a systematic*

*hydrogeochemistry and isotopic analysis of the geothermal fluids in the East Anatolian Fault*

*Zone (EAFZ). The results show that earthquakes reconstructed these geothermal fluids.*

Reply: Thank you for your recognition of our work. Thank you.

*Considering gypsum as an earthquake tracer is excellent reasoning for analysing the impact of*

*anomalies after the earthquake, and the work could be a great reference for future studies*

*related to the earthquake.*

Reply: Yes, through the analysis of groundwater after the earthquake, we discovered the potential value of gypsum as an earthquake warning. It is hoped that this work will attract the attention of more researchers and colleagues, and incubate more meaningful achievement.

*To enrich this excellent analysis, I have some remarks concerning the implication of*

*macroscopic and microscopic aspects of geothermal fluids before and after the earthquake,*

*notably the relation with the structural geology of the region. For this, some questions seem*

*important to be asked.*

*First, from a macroscopic point of view, it is necessary to understand, in the normal case (before*

*the earthquake), from a geological point of view, if the existing deformations (faults) already*

*have effective structures for the infiltration of meteorological waters and the implication of the*

*disposition of the thermal springers according to the faults. After the earthquake, is there any*

*sampling from Miocene groundwater and soil? Is there recent salt precipitation in the Miocene*

*and upper Eocene-Oligocene soil and/or in the soil of the surrounding springer sources? Is*

*there a rise in the ground level due to fault action, and are there marine intrusions that occurred*

*after the strike-slip? Is there significant contamination of the water table (increased electrical*

*conductivity)?*

Reply: Hot springs and fault zones are often associated. Hot springs are considered as one of the potential means of earthquake warning. A large number of research results have been published in Japan, the United States, Iceland, Spain, China, Turkey... ... In EAFZ, many hot springs have been systematically studied, and the results show that these hot springs contain material supply from deep crust and even mantle. Therefore, it is highly possible to obtain valuable information by conducting post-earthquake hydrochemical and isotopic analyses of these hot springs.

Unfortunately, we only collected water samples after the earthquake and did not analyze soil samples. Your comment is a very good suggestion, reminding us that detailed analysis of surrounding rock may be needed in future work. Thank you.

Salt precipitation and electrical conductivity (EC). Before we can answer your question, we need to explain an error in the manuscript. Our sample was taken in March 2023 (within one month after the earthquake). In the video 1 we provided, the macro abnormal changes of HS14

were diluted by the adjacent stream, coupled with the fact that the samples were taken within one month after the earthquake and no soil samples were collected, **we could not accurately**

**determine whether salt precipitation existed**. By comparing the EC of the same hot spring during the seismically quiet period and the seismically active period, we found that the EC of

HS14 increased slightly (varying from 990 to 1305). Data of EC pre-earthquake from *Yuce, G.,*

*Italiano, F., D'Alessandro, W., Yalcin, T. H., Yasin, D. U., Gulbay, A. H., Ozyurt, N. N., Rojay,*

*B., Karabacak, V., Bellomo, S., Brusca, L., Yang, T., Fu, C. C., Lai, C. W., Ozacar, A., and Walia,*

*V.: Origin and interactions of fluids circulating over the Amik Basin (Hatay, Turkey) and*

*relationships with the hydrologic, geologic and tectonic settings, Chemical Geology, 388, 23-*

*39, 2014.*

Seawater intrusion was evident after the earthquake. $Na^+$ and $Cl^-$ of HS14, HS15 and HS16

increased significantly, indicating the possible existence of seawater intrusion (Fig. 6

manuscript).

Rise in the ground level due to fault action is common. We have made a detailed study on the post-earthquake surface rupture and post-earthquake risk analysis. Article link:*Liang, P., Xu,*

*Y., Zhou, X., Li, Y., Tian, Q., Zhang, H., Ren, Z., Yu, J., Li, C., Gong, Z., Wang, S., Dou, A.,*

*Ma, Z., and Li, J.: Coseismic surface ruptures of MW7.8 and MW7.5 earthquakes*

*occurred on February 6, 2023, and seismic hazard assessment of the East Anatolian Fault*

*Zone, Southeastern Turkiye, Science China Earth Sciences, doi: 10.1007/s11430-024-*

*1457-7, 2024.*

[Figure]

Screenshot from Liang et al., 2024 doi: 10.1007/s11430-024-1457-7 (If the picture cannot be displayed, please check it in the attachment, thank you).

*From a microscopic point of view, gypsum is easily and quickly influenced by contact with water, thanks to its physicochemical characteristics, in particular its very high dissolution rate and its solubility in water that make it an excellent tracer of hydrochemical anomaly but also a tracer of lithological instability (Khebizi et al., 2022; Khebizi et al., 2023). For this, I am pleased to invite you to read the part concerning the gypsum implication on the lithological instability in my article published in Larhyss Journal and my oral communications, which expose, for the first time in Algeria, a new concept of the lithological vulnerability of the subsurface. Although the study areas differ, the analysis presented in my work shows the indication of gypsum dissolution at the regional scale as an excellent major risk indicator. The lithological vulnerability of the subsurface concept can be applied to different situations around the world, notably the case of earthquakes. It highlights the hydrodynamic anomalies' relation with the structural and geological context of the area to be studied.*

Thank you very much for your sharing. It's a fantastic set of work. From my personal point of view, I can't agree with you more. Gypsum's very high dissolution rate and solubility in water can be used for risk warning of earthquakes and geological disasters. Thank you again for your information. Your work gives us great encouragement and confidence.

*Second, if there is a remarkable increase in calcium concentration in water after the earthquake, how do you explain the reaction of carbonate dissolution and the origin of $CO_2$? Is it linked to magmatic activity? In this case, is there a signature of other gases on other cations? Or is it only related to carbonate since the calcite dissolution is linked to the mineral's surface to be in direct contact with water?*

In my opinion, Ca may come from carbonate or igneous rocks. In order to accurately restrict the source area of Ca, we are also considering introducing Ca isotopes to distinguish its sources.

Ca isotopes in carbonate rocks are lighter than those in igneous rocks and mantle. Ca isotope has a good potential in the source region that restricts Ca.

The index of $CO_2$ source region is very mature. Geothermal gases are well studied at EAFZ.

The C isotope study of $CO_2$ shows that $CO_2$ is controlled by deep carbon and inorganic carbonate ($-5.6$ to $-0.2‰$) (*Italiano, F., Sasmaz, A., Yuce, G., and Okan, O. O.: Thermal fluids*

*along the East Anatolian Fault Zone (EAFZ): Geochemical features and relationships with the*

*tectonic setting, Chemical Geology, 339, 103-114, 2013.*). He isotope analysis also shows a large proportion of the mantle.

Explanation of the specific process: gypsum dissolution and carbonate dissolution are together.

In the manuscript, PHREEQC was used to simulate the water-rock reaction process (Fig. 7).

The results show that gypsum dissolution alone is not enough to explain the Ca content in the samples, indicating that calcite and other minerals are involved in the water-rock reaction.

Combined with previous studies, we believe that $CO_2$ from deep water is first dissolved in water, and then reacts with gypsum or calcite. $CO_2$ is associated with magma, but does not form volcanic eruptions and may only exist in deep areas of partial melting.

*Allow me to add that the underground water circulation, which is controlled by faults and*

*hydraulic parameters (permeability), determines water-rock equilibrium. In this case, water-*

*rock equilibrium depends on the host rock spatial disposition of rock that guides water*

*mineralization and the different processes. Consequentially, the water-rock equilibrium*

*changes from one area to another due to changes in water mineralization according to the host*

*rock lithology. For this, the information that can be taken from the geological map is that springer's water is related to ophiolite rocks. So, I think water geochemistry indicates similar water-rock interactions for all sources. However, a mineral's enrichment zoning can occur due to (i) the meteorological conditions, (ii) the proximity of the springer water from seawater, and/or (iii) the distance from the upstream. The earthquake reconstructed these geothermal fluids depending on the energy released which controls hydrothermal circulation and amplifies interactions with the surrounding environment whether at depth or on the surface. For this, vulnerability zoning in a horizontal and vertical direction can be done according to chemical variation, notably gypsum and probably halite enrichment. It can be indicated as shown in Fig. 8.*

I can't agree with you more. Water-rock reaction is affected by meteorology, rock properties, permeability, porosity, temperature, pressure... Multiple factors control. At present, our work is limited to the analysis of water chemistry and isotopes, and there is a lot of work to be done in the future. These works involve not only geochemistry, but also rock mechanics, numerical simulation and other interdisciplinary fields, and we hope to have more like-minded colleagues to explore together.

Earthquake warning is the most difficult problem faced by mankind. Groundwater is considered as one of the means to explore earthquake early warning. However, groundwater in its natural environment is very complex. There is still a long way to go to explore the relationship between groundwater and earthquakes.

*Finally, the discussion on this topic is very significant, and the structural and lithological vulnerability and their tracers after the earthquake using vulnerability mapping of the Turkey*

*earthquake seems very interesting for future work.*

Thank you for your recognition of our work, your recognition is our driving force forward.

Sincere thanks and best wishes.

| T (°C) | pH | Na⁺ mg/L | K⁺ mg/L | Ca²⁺ mg/L | Mg²⁺ mg/L | HCO₃⁻ mg/L | Cl⁻ mg/L | SO₄²⁻ mg/L | SiO₂ mg/L | δ¹⁸O | δD | Data source |
|---|---|---|---|---|---|---|---|---|---|---|---|---|
| 63.2 | 6.8 | 1050.0 | 140.0 | 410.0 | 82.0 | 3100.0 | 460.0 | 290.0 | 150.0 | -12.5 | -96.0 | |
| 18.7 | 7.8 | 3.3 | 0.8 | 11.0 | 1.3 | 41.0 | 0.7 | 4.8 | 3.5 | -12.7 | -86.0 | |
| 55.1 | 9.7 | 39.0 | 0.9 | 13.0 | 0.4 | 83.0 | 2.8 | 37.0 | 79.0 | -14.2 | -99.0 | |
| 5.5 | 7.4 | 1.3 | 0.3 | 6.2 | 0.5 | 25.0 | 0.4 | 1.1 | 2.7 | -13.9 | -94.0 | |
| 32.5 | 7.2 | 40.0 | 0.7 | 31.0 | 0.1 | 47.0 | 4.9 | 110.0 | 50.0 | -14.0 | -95.0 | |
| 17.5 | 7.9 | 2.6 | 0.4 | 22.0 | 2.1 | 76.0 | 0.4 | 3.4 | 8.2 | -13.1 | -89.0 | |
| 11.7 | 5.3 | 314.0 | 19.0 | 240.0 | 45.0 | 1100.0 | 150.0 | 230.0 | 120.0 | -13.7 | -96.0 | |
| 13.1 | 7.0 | 6.5 | 5.2 | 43.0 | 7.1 | 150.0 | 4.5 | 13.0 | | -13.1 | -93.0 | |
| 36.6 | 5.6 | 2050.0 | 51.0 | 370.0 | 64.0 | 3000.0 | 1750.0 | 240.0 | 120.0 | -13.4 | -104.0 | |
| 6.0 | 7.3 | 4.7 | 0.5 | 9.0 | 2.9 | 47.0 | 0.5 | 2.1 | 6.0 | -12.1 | -82.0 | |
| 15.2 | 6.0 | 48.0 | 8.5 | 33.0 | 33.0 | 370.0 | 9.0 | 6.1 | 68.0 | -13.3 | -93.0 | |
| 17.4 | 6.8 | 46.0 | 8.2 | 36.0 | 33.0 | 350.0 | 9.0 | 8.5 | | -13.1 | -92.0 | |
| 36.1 | 9.4 | 130.0 | 1.4 | 18.0 | 1.6 | 160.0 | 79.0 | 63.0 | 62.0 | -14.5 | -101.0 | |
| 9.2 | 7.4 | 20.0 | 0.6 | 25.0 | 9.0 | 150.0 | 4.8 | 3.3 | | -10.0 | -66.0 | |
| 24.3 | 5.9 | 1600.0 | 20.0 | 260.0 | 170.0 | 2500.0 | 1750.0 | 2.5 | 120.0 | -12.7 | -93.0 | |
| 14.7 | 7.6 | 5.1 | 1.5 | 15.0 | 5.1 | 81.0 | 1.1 | 1.5 | | -11.9 | -80.0 | |
| 11.6 | 5.6 | 160.0 | 13.0 | 260.0 | 150.0 | 1400.0 | 180.0 | 110.0 | 110.0 | -13.6 | -100.0 | |
| 27.0 | 6.1 | 1100.0 | 46.0 | 260.0 | 78.0 | 3050.0 | 320.0 | 150.0 | 100.0 | -10.1 | -84.0 | |
| 11.9 | 8.0 | 42.0 | 3.6 | 41.0 | 21.0 | 290.0 | 6.1 | 21.0 | 6.9 | -12.7 | -92.0 | |
| 36.5 | 6.1 | 2400.0 | 73.0 | 220.0 | 88.0 | 5950.0 | 660.0 | 160.0 | 140.0 | -10.5 | -88.0 | |
| 57.3 | 7.3 | 2550.0 | 82.0 | 290.0 | 130.0 | 6700.0 | 590.0 | 150.0 | 120.0 | -10.7 | -88.0 | |
| 14.5 | 7.2 | 16.0 | 3.2 | 51.0 | 29.0 | 310.0 | 4.1 | 9.6 | 18.0 | -11.2 | -77.0 | |
| 24.9 | 5.7 | 240.0 | 34.0 | 120.0 | 51.0 | 950.0 | 140.0 | 4.7 | 80.0 | -13.8 | -98.0 | |
| 13.1 | 7.5 | 16.0 | 3.8 | 31.0 | 13.0 | 190.0 | 2.5 | 4.0 | 6.1 | -13.1 | -94.0 | |
| 37.7 | 6.3 | 1800.0 | 59.0 | 520.0 | 160.0 | 2300.0 | 2500.0 | 5.2 | 180.0 | -13.2 | -97.0 | |
| 28.6 | 6.3 | 710.0 | 53.0 | 240.0 | 120.0 | 1550.0 | 950.0 | 0.2 | 100.0 | -12.2 | -91.0 | |
| 12.5 | 7.1 | 10.0 | 4.2 | 39.0 | 7.5 | 160.0 | 3.4 | 8.1 | 7.5 | -11.0 | -75.0 | |
| 57.0 | 6.6 | 900.0 | 120.0 | 450.0 | 240.0 | 2850.0 | 1150.0 | 0.3 | 170.0 | -12.5 | -92.0 | |
| 13.6 | 6.1 | 30.0 | 6.5 | 72.0 | 37.0 | 440.0 | 7.3 | 12.0 | 7.5 | -12.8 | -92.0 | |
| 28.5 | 5.9 | 460.0 | 17.0 | 82.0 | 96.0 | 1400.0 | 250.0 | 0.1 | 100.0 | -13.1 | -91.0 | |
| 12.9 | 7.3 | 16.0 | 1.6 | 20.0 | 9.6 | 140.0 | 5.6 | 1.8 | | -12.5 | -84.0 | |
| 33.4 | 6.2 | 450.0 | 21.0 | 93.0 | 100.0 | 1600.0 | 160.0 | 2.3 | 140.0 | -13.0 | -90.0 | Aydin H., Karakuş H. and Mutlu H. (2020) Hydrogeochemistry of geothermal waters in eastern Turkey: Geochemical and isotopic constraints on water-rock interaction. Journal of Volcanology and Geothermal Research 390. |
| 11.6 | 7.2 | 17.0 | 1.2 | 21.0 | 10.0 | 140.0 | 1.1 | 5.3 | | -12.9 | -88.0 | |
| 36.0 | 6.2 | 1700.0 | 42.0 | 150.0 | 81.0 | 2150.0 | 1800.0 | 0.8 | 110.0 | -13.3 | -95.0 | |
| 38.4 | 6.5 | 1500.0 | 40.0 | 160.0 | 100.0 | 2700.0 | 1150.0 | 0.4 | 140.0 | -13.6 | -97.0 | |
| 19.2 | 7.9 | 1300.0 | 17.0 | 61.0 | 15.0 | 3500.0 | 25.0 | 0.0 | 68.0 | -12.5 | -94.0 | |
| 14.5 | 7.4 | 14.0 | 1.1 | 75.0 | 27.0 | 360.0 | 2.7 | 7.0 | 5.3 | -11.0 | -77.0 | |
| 33.4 | 6.5 | 570.0 | 19.0 | 420.0 | 600.0 | 4050.0 | 770.0 | 37.0 | 180.0 | -12.3 | -90.0 | |
| 18.6 | 5.9 | 80.0 | 8.4 | 67.0 | 330.0 | 1900.0 | 36.0 | 60.0 | 110.0 | -12.2 | -84.0 | |
| 13.7 | 7.5 | 2.1 | 0.7 | 2.1 | 43.0 | 210.0 | 2.9 | 12.0 | | -11.7 | -82.0 | |
| 25.3 | 6.5 | 190.0 | 9.9 | 510.0 | 52.0 | 1950.0 | 170.0 | 20.0 | 51.0 | -12.0 | -82.0 | |
| 30.3 | 6.4 | 86.0 | 21.0 | 230.0 | 110.0 | 1400.0 | 65.0 | 53.0 | 80.0 | -12.0 | -78.0 | |
| 46.9 | 6.6 | 84.0 | 14.0 | 160.0 | 60.0 | 950.0 | 64.0 | 32.0 | 95.0 | -12.2 | -79.0 | |
| 41.0 | 6.5 | 46.0 | 10.0 | 240.0 | 70.0 | 1150.0 | 19.0 | 44.0 | 75.0 | -12.0 | -76.0 | |
| 24.6 | 6.4 | 770.0 | 46.0 | 150.0 | 110.0 | 1800.0 | 560.0 | 68.0 | 85.0 | -12.1 | -87.0 | |
| 19.5 | 5.4 | 22.0 | 34.0 | 49.0 | 9.3 | 130.0 | 19.0 | 120.0 | 86.0 | -14.0 | -96.0 | |
| 48.3 | 7.0 | 160.0 | 70.0 | 93.0 | 81.0 | 880.0 | 160.0 | 140.0 | 68.0 | -12.6 | -94.0 | |
| 64.2 | 6.6 | 120.0 | 52.0 | 130.0 | 84.0 | 920.0 | 120.0 | 100.0 | 153.0 | -11.3 | -91.0 | |
| 53.7 | 7.0 | 160.0 | 59.0 | 180.0 | 69.0 | 990.0 | 120.0 | 160.0 | 60.0 | -11.5 | -92.0 | |
| 25.8 | 2.4 | 34.0 | 71.0 | 70.0 | 32.0 | 0.0 | 9.0 | 670.0 | 140.0 | -10.8 | -79.0 | |
| 50.6 | 7.2 | 170.0 | 69.0 | 69.0 | 76.0 | 590.0 | 180.0 | 130.0 | 56.0 | -12.5 | -95.0 | |
| 65.2 | 6.6 | 160.0 | 76.0 | 130.0 | 52.0 | 710.0 | 270.0 | 120.0 | 120.0 | -12.8 | -90.0 | |
| 39.8 | 6.9 | 150.0 | 70.0 | 78.0 | 78.0 | 750.0 | 170.0 | 150.0 | 79.0 | -12.8 | -94.0 | |
| 11.5 | 7.8 | 17.0 | 1.9 | 48.0 | 30.0 | 310.0 | 1.0 | 8.2 | 5.2 | -12.9 | -91.0 | |
| 18.1 | 6.2 | 180.0 | 20.0 | 92.0 | 12.0 | 400.0 | 190.0 | 34.0 | 45.0 | -11.4 | -83.0 | |
| 65.0 | 6.5 | 1850.0 | 190.0 | 330.0 | 64.0 | 1150.0 | 2500.0 | 420.0 | 78.0 | -10.1 | -79.0 | |
| 51.1 | 7.1 | 730.0 | 70.0 | 220.0 | 75.0 | 1200.0 | 640.0 | 460.0 | 57.0 | -11.3 | -82.0 | |
| 37.0 | 7.0 | 610.0 | 170.0 | 200.0 | 30.0 | 1600.0 | 360.0 | 150.0 | 100.0 | -11.4 | -90.0 | |
| 25.8 | 6.1 | 240.0 | 77.0 | 52.0 | 50.0 | 750.0 | 160.0 | 53.0 | 130.0 | -12.2 | -85.0 | |
| 34.3 | 7.6 | 84.0 | 24.0 | 99.0 | 71.0 | 810.0 | 17.0 | 23.0 | 130.0 | -13.5 | -91.0 | |
| 25.1 | 6.7 | 380.0 | 120.0 | 140.0 | 170.0 | 1950.0 | 160.0 | 63.0 | 130.0 | -11.6 | -85.0 | |
| 11.4 | 7.2 | 10.0 | 0.7 | 20.0 | 2.8 | 85.0 | 1.0 | 6.3 | 23.0 | -12.1 | -83.0 | |
| 53.5 | 7.6 | 2600.0 | 170.0 | 180.0 | 69.0 | 5500.0 | 850.0 | 540.0 | 99.0 | -3.4 | -72.0 | |
| 25.4 | 6.7 | 40.0 | 4.3 | 180.0 | 120.0 | 1150.0 | 14.0 | 40.0 | 19.0 | -12.7 | -89.0 | |
| 34.0 | 7.2 | 900.0 | 160.0 | 110.0 | 120.0 | 2100.0 | 560.0 | 260.0 | 31.0 | -10.7 | -82.0 | |
| 46.8 | 9.2 | 210.0 | 4.2 | 20.0 | 1.5 | 210.0 | 140.0 | 110.0 | 31.0 | -10.9 | -72.0 | |
| 16.3 | 6.3 | 140.0 | 26.0 | 390.0 | 190.0 | 1200.0 | 27.0 | 850.0 | 11.0 | -10.7 | -69.0 | |
| 51.6 | 6.9 | 320.0 | 33.0 | 70.0 | 15.0 | 1100.0 | 23.0 | 0.4 | 170.0 | -9.6 | -65.0 | |
| 14.2 | 10.4 | 120.0 | 13.0 | 100.0 | 62.0 | 900.0 | 32.0 | 4.2 | 9.0 | -9.8 | -66.0 | |
| 34.5 | 6.5 | 300.0 | 52.0 | 130.0 | 94.0 | 1550.0 | 47.0 | 0.5 | 120.0 | -10.0 | -69.0 | |

[revised manuscript text omitted]

Öztekin Okan Ö., Kalender L. and Çetindağ B. (2018) Trace-element hydrogeochemistry of thermal waters of Karakoçan (Elazığ) and Mazgirt (Tunceli), Eastern Anatolia, Turkey. Journal of Geochemical Exploration 194, 29-43.

| | | | | | | | | | | | |
|---|---|---|---|---|---|---|---|---|---|---|---|
| 14.5 | 10.5 | 3.5 | 0.3 | 14.9 | 1.7 | 1.8 | 12.0 | 0.0 | 1.5 | -6.8 | -48.2 |
| 22.5 | 11.7 | 77.2 | 2.7 | 66.9 | 0.0 | 0.0 | 66.9 | 0.0 | 0.2 | -7.9 | -45.4 |
| 27.8 | 8.2 | 12.5 | 3.9 | 45.6 | 24.2 | 303.8 | 5.9 | 9.9 | 15.3 | -9.4 | -57.8 |
| 15.9 | 7.6 | 14.1 | 2.2 | 63.8 | 15.7 | 285.5 | 5.3 | 24.8 | 8.4 | -7.4 | -42.0 |
| 29.0 | 7.2 | 87.5 | 8.4 | 81.0 | 18.1 | 336.7 | 126.8 | 54.7 | 11.3 | | |
| 23.8 | 7.4 | 6.5 | 0.9 | 7.0 | 110.0 | 358.1 | 8.0 | 11.1 | 15.6 | -5.8 | -33.7 |
| 18.3 | 7.4 | 6.5 | 1.0 | 116.0 | 8.0 | 367.2 | 8.5 | 10.2 | 17.2 | | |
| 41.0 | 7.4 | 56.2 | 6.3 | 67.7 | 11.3 | 245.8 | 45.2 | 84.1 | 12.9 | -7.4 | -47.9 |
| 51.0 | 7.3 | 195.5 | 20.2 | 38.7 | 5.6 | 472.8 | 113.4 | 3.3 | 19.4 | -10.3 | -63.2 |
| 84.5 | 6.2 | 2756.1 | 81.9 | 773.6 | 124.7 | 384.3 | 6571.5 | 1287.2 | | -9.5 | -59.8 |
| 33.1 | 6.4 | 120.5 | 13.8 | 286.9 | 46.4 | 446.5 | 196.8 | 689.2 | 12.9 | -7.9 | -48.2 |
| 15.2 | 7.0 | 17.2 | 2.6 | 73.2 | 10.8 | 196.4 | 21.8 | 83.8 | 5.4 | -9.7 | -59.6 |
| 33.7 | 6.5 | 124.1 | 14.5 | 288.3 | 49.1 | 452.6 | 188.9 | 602.2 | 13.5 | -7.4 | -50.7 |
| 56.6 | 6.6 | 67.1 | 18.0 | 350.3 | 48.3 | 241.6 | 71.7 | 1015.4 | | -8.9 | -57.7 |
| 62.2 | 6.8 | 68.4 | 18.1 | 361.3 | 52.5 | 242.2 | 77.1 | 1062.2 | | -9.1 | -58.0 |
| 8.6 | 8.1 | 0.6 | 0.2 | 44.3 | 13.5 | 205.6 | 0.6 | 7.1 | 2.1 | -9.0 | -53.0 |
| 44.0 | 6.8 | 169.7 | 13.1 | 130.0 | 17.1 | 429.4 | 257.8 | 65.3 | 18.0 | -9.5 | -57.9 |
| 9.0 | 7.9 | 7.1 | 1.0 | 44.2 | 4.6 | 178.7 | 3.7 | 9.3 | 2.6 | | |
| 21.1 | 7.2 | 26.0 | 2.9 | 101.3 | 24.0 | 342.8 | 32.8 | 85.1 | 8.9 | -9.4 | -56.9 |
| 20.0 | 9.0 | 450.0 | 16.0 | 4.1 | 0.0 | 565.5 | 301.6 | 17.0 | 7.1 | | |
| 22.7 | 8.0 | 278.5 | 12.7 | 12.5 | 7.7 | 464.2 | 190.6 | 4.2 | 7.0 | | |
| 26.5 | 7.2 | 65.9 | 5.6 | 145.0 | 40.2 | 230.0 | 41.9 | 479.2 | 12.4 | | |
| 27.3 | 7.2 | 68.1 | 5.3 | 151.9 | 43.5 | 249.5 | 39.3 | 501.1 | 12.5 | | |
| 35.0 | 7.3 | 33.6 | 3.0 | 39.9 | 25.6 | 328.2 | 11.9 | 1.8 | 13.9 | | |
| 34.8 | 7.0 | 17.4 | 2.6 | 56.0 | 15.2 | 281.9 | 8.9 | 16.4 | 14.2 | | |

Baba A., Şaroğlu F., Akkuş I., Özel N., Yeşilnacar M. İ., Nalbantçılar M. T., Demir M. M., Gökçen G., Arslan Ş., Dursun N., Uzelli T. and Yazdani H. (2019) Geological and hydrochemical properties of geothermal systems in the southeastern region of Turkey. Geothermics 78, 255-271.

| | | | | | | | | | | | |
|---|---|---|---|---|---|---|---|---|---|---|---|
| 20.0 | 7.2 | 13.3 | 2.6 | 67.7 | 31.0 | 311.0 | 23.3 | 40.1 | 26.7 | -6.4 | -31.1 |
| 21.0 | 7.5 | 11.9 | 0.9 | 59.3 | 27.4 | 268.0 | 24.1 | 32.0 | 20.2 | -6.5 | -31.3 |
| 21.0 | 7.2 | 42.6 | 1.0 | 55.0 | 106.0 | 580.0 | 60.8 | 89.9 | 32.3 | -6.1 | -29.8 |
| 22.1 | 7.6 | 15.5 | 0.7 | 58.5 | 29.5 | 293.0 | 23.5 | 15.8 | 43.1 | -5.6 | -26.3 |
| 22.6 | 7.4 | 16.0 | 1.3 | 60.7 | 37.1 | 329.0 | 25.2 | 37.2 | 28.8 | -6.4 | -31.7 |
| 23.3 | 7.1 | 33.6 | 4.3 | 129.0 | 38.8 | 348.0 | 50.0 | 176.0 | 20.6 | -6.0 | -32.5 |
| 29.0 | 7.3 | 24.8 | 4.7 | 94.3 | 30.2 | 305.0 | 39.3 | 78.5 | 20.7 | -6.7 | -36.6 |
| 37.7 | 6.6 | 315.0 | 29.6 | 166.0 | 40.6 | 458.0 | 411.0 | 376.0 | 40.0 | -7.0 | -39.5 |
| 25.8 | 6.9 | 27.2 | 1.3 | 87.2 | 18.4 | 317.0 | 36.5 | 27.8 | 27.2 | -6.8 | -36.7 |
| 30.3 | 9.0 | 257.0 | 1.0 | 28.1 | 0.2 | 36.6 | 178.0 | 335.0 | 29.2 | -7.1 | -37.1 |
| 28.9 | 7.1 | 28.5 | 3.6 | 87.1 | 66.0 | 390.0 | 47.1 | 101.0 | 45.4 | -6.8 | -35.5 |
| 31.2 | 6.9 | 80.1 | 9.8 | 133.0 | 67.8 | 253.0 | 59.2 | 469.0 | 69.1 | -6.6 | -34.8 |
| 22.0 | 7.3 | 21.6 | 0.2 | 58.4 | 45.0 | 296.0 | 18.6 | 104.0 | 46.3 | -6.8 | -36.1 |
| 23.1 | 6.9 | 10.2 | 1.9 | 72.6 | 32.4 | 268.0 | 12.4 | 88.3 | 23.5 | -7.3 | -39.4 |
| 16.3 | 7.1 | 5.7 | 0.6 | 67.4 | 16.7 | 262.0 | 9.9 | 6.7 | 16.2 | -7.2 | -36.1 |
| 19.7 | 7.1 | 28.3 | 0.6 | 77.2 | 47.3 | 400.0 | 43.4 | 33.6 | 54.8 | -6.4 | -31.2 |
| 22.5 | 7.4 | 105.0 | 2.2 | 29.1 | 86.6 | 403.0 | 73.5 | 184.0 | 9.5 | -4.6 | -23.6 |
| 20.2 | | 13.8 | 2.2 | 73.9 | 33.1 | 323.0 | 23.1 | 40.3 | 25.6 | -5.9 | -27.8 |
| 20.4 | | 12.0 | 0.9 | 61.4 | 30.5 | 262.0 | 21.5 | 31.7 | 20.0 | -5.9 | -28.8 |
| 19.6 | | 41.5 | 0.7 | 57.8 | 105.0 | 555.0 | 53.2 | 77.3 | 31.6 | -5.9 | -29.3 |
| 38.0 | 6.7 | 315.0 | 28.9 | 131.0 | 48.0 | 445.0 | 354.0 | 353.0 | 39.0 | -6.7 | -38.9 |
| 22.5 | 7.2 | 11.3 | 1.8 | 87.8 | 34.4 | 323.0 | 11.9 | 86.7 | 24.7 | -6.8 | -37.0 |
| 21.8 | 6.9 | 21.2 | 1.4 | 94.6 | 31.9 | 348.0 | 40.4 | 38.8 | 32.0 | -6.3 | -33.2 |
| 22.1 | 7.3 | 9.7 | 1.3 | 80.6 | 31.2 | 323.0 | 9.0 | 64.3 | 21.0 | -7.2 | -38.0 |
| 37.6 | 7.5 | 276.0 | 5.4 | 41.1 | 10.3 | 91.5 | 231.0 | 361.0 | 29.2 | -7.2 | -36.2 |
| 42.8 | 7.3 | 10100.0 | 68.2 | 1030.0 | 224.0 | 67.1 | 17600.0 | 18.6 | 18.4 | -1.8 | -8.5 |
| 32.6 | 8.0 | 2960.0 | 19.2 | 81.7 | 25.2 | 146.0 | 4640.0 | 8.0 | 21.3 | -4.5 | -25.1 |
| 42.3 | 7.3 | 10100.0 | 50.2 | 1220.0 | 278.0 | 73.2 | 18800.0 | 0.2 | 20.3 | -1.2 | -7.9 |
| 28.8 | | 2980.0 | 13.0 | 102.0 | 30.6 | 177.0 | 4780.0 | 0.5 | 21.9 | -4.3 | -25.9 |
| 26.8 | 7.9 | 11000.0 | 57.5 | 1020.0 | 360.0 | 97.6 | 19900.0 | 0.5 | 15.2 | -1.0 | -8.0 |
| 33.7 | 10.6 | 49.9 | 1.8 | 44.7 | 0.1 | 183.0 | 47.6 | 0.2 | 0.2 | -8.7 | -45.0 |
| 25.5 | 11.6 | 28.6 | 0.9 | 83.4 | 0.8 | 244.0 | 45.8 | 0.0 | 0.2 | -7.9 | -40.2 |
| 33.0 | 11.6 | 50.3 | 2.1 | 42.1 | 0.4 | 177.0 | 44.5 | 0.0 | 0.2 | -8.1 | -42.1 |
| 21.7 | 12.2 | 55.0 | 1.2 | 110.0 | 0.1 | 336.0 | 72.1 | 0.2 | 0.2 | -7.9 | -41.3 |
| | 6.3 | 0.8 | 0.2 | 3.6 | 0.2 | 13.0 | 1.6 | 4.4 | | -7.4 | -40.0 |
| | 6.5 | 0.7 | 0.4 | 2.1 | 0.2 | 11.0 | 1.4 | 2.5 | | -7.7 | -42.0 |
| | 6.7 | 0.7 | 0.3 | 2.3 | 0.2 | 5.0 | 1.4 | 3.8 | | -8.2 | -46.0 |
| | | 2.3 | 0.3 | 5.1 | 0.5 | 9.0 | 4.5 | 5.9 | | -4.5 | -16.7 |
| | | 2.2 | 0.4 | 3.9 | 0.4 | 15.0 | 4.2 | 4.1 | | -4.5 | -18.7 |
| | | 2.4 | 0.3 | 3.3 | 0.4 | 7.0 | 4.3 | 4.8 | | -6.1 | -28.3 |

Yuce G., Italiano F., D'Alessandro W., Yalcin T. H., Yasin D. U., Gulbay A. H., Ozyurt N. N., Rojay B., Karabacak V., Bellomo S., Brusca L., Yang T., Fu C. C., Lai C. W., Ozacar A. and Walia V. (2014) Origin and interactions of fluids circulating over the Amik Basin (Hatay, Turkey) and relationships with the hydrologic, geologic and tectonic settings. Chemical Geology 388, 23-39.

---

## Author Response (AR1)

- 1 Dear Editorial Office of HESS and Prof. Dai
- We wish to express our sincere gratitude for the editorial team's diligent handling
- 3 of our manuscript and extend particular appreciation to Prof. Dai for your judicious
- 4 oversight throughout the review process. Your constructive decision letter has provided
- 5 us the opportunity to enhance the manuscript's scientific rigor. In response to the
- 6 insightful comments from two reviewers and two domain experts, we have thoroughly
- 7 revised the manuscript. All the revised contents have been marked in red in the
- 8 manuscript.

**Major revisions include:**

- 1. Correction of sample collection time.
- 2. Use "groundwater" instead of "geothermal water" to define the sample in this study.
- 12 3. Historical EAFZ hydrogeochemical data were collected.
- 4. The Sr isotopes of the groundwater samples were analyzed.
- 5. Modified the title.
- 6. We redrew all the Figures.
- 7. We reorganized the Abstract, Introduction, Discussion and Conclusions.
- 8. We supplement the analysis method and data quality control description.
- 9. We have made explanations of some misunderstandings.
- 19 10. The estimation of the temperature of 382°C was deleted.
- 20 11. The simulation of the saturation index of anhydrite was added.
- 21 12. Fig. 6 (spatial distribution characteristics) in the original draft has been deleted.
- 22 13. Supplemented the evidence of geothermal gas in EAFZ.
- 24 Key improvements are summarized below:
- 25 **1. Enhanced Data Completeness**

- Historical Data Integration: We systematically compiled published data (2013-2023), revealing spatial hydrogeochemical zonation in the East Anatolian Fault Zone (EAFZ):

  Northern Segments: Mixed shallow/deep circulation with igneous rock-dominated water-rock interactions.

  Central-Southern Segments: Shallow circulation dominated by sedimentary
- mineral dissolution (e.g., anhydrite, carbonates), with localized seawater influence.

  Causal Linkage Clarification: PHREFOC simulations quantify anhydrite's
- Causal Linkage Clarification: PHREEQC simulations quantify anhydrite's contribution to SO42- anomalies (30-100%).

**2. Refined Anhydrite-Tectonic Linkage**

- Terminological Precision: Removed all "seismic precursor" claims, replacing with "indicator of water-rock interaction intensity".
- Mechanistic: Combined pre-earthquake macroscopic anomalies. The analysis of post-earthquake data and historical data proves that anhydrite may be one of the causes of groundwater macroscopic anomaly
- Explicit caveat: "Causal links between anhydrite dynamics and tectonics require long-term validation"

**3. Revised Conclusions**

Restructured Key Findings:

- "Anhydrite abundance serves as a sensitive indicator of water-rock interaction intensity, potentially modulated by tectonic activity. Establishing fault-zone hydrogeochemical baselines is prerequisite for deciphering tectonic-hydrologic coupling."
  - In short, after fully and effectively communicating with the reviewers, we modified the possible problems in our manuscript according to the suggestions of the reviewers, so that the analysis of data in the manuscript is more rigorous and the extension is appropriate
- We sincerely wish the current version meets your standards and welcome further guidance.

| 55 | Finally, I would like to thank HESS editorial Department and Dai Editor-in-Chie |                         |
|----|---------------------------------------------------------------------------------|-------------------------|
| 56 | for their hard work                                                             |                         |
| 57 | 7                                                                               | Sincerely               |
| 58 | 3                                                                               | Zebin Luo               |
| 59 |                                                                                 | Zebin L@mail.xhu.edu.cn |

**Point-by-point response to comments:**

- Note: *Italic blue* is the comment. Black is the reply, and **important sentences are bolded**.
- Red indicates the position of the modification information in the newly submitted revised
- 63 draft.

- 64 Reply to referee comments
- 65 RC1: 'Comment on hess-2024-395', Walter D'Alessandro, 13 Jan 2025
- 66 Dear Walter D'Alessandro
- 67 After two in-depth discussions with you, we have gained a new understanding of the
- viewpoints in the initial manuscript. This is mainly attributed to your highly professional
- and constructive opinions and suggestions, which are of great value to us in improving the
- quality of the manuscripts. After carefully reading your comments, we have made a reply
- 71 to your comments point-by-point under the discussion of all the article authors.
- 72 The manuscript "Gypsum as a potential tracer of earthquake: a case study of the Mw7.8
- earthquake in the East Anatolian Fault Zone, southeastern Turkey" by Luo et al. presents
- 74 the results of sampling campaign of groundwaters in the area of the two strong earthquakes
- 75 that hit heavily Turkey in February 2023. Only the analytical results (major ions, trace
- 76 elements and water isotopes) of samples collected about one year after the quakes are
- 77 considered, which is a strong limitation of this study. I feel that this study cannot be
- 78 *published in this form.*
- 79 Reply: Thanks. We sincerely appreciate your critical observation regarding the sampling
- 80 timeline. Please allow us to clarify and substantiate our findings through the following
- 81 revisions:

- 82 **Critical Data Correction**: Amended the erroneous "one-year post-seismic" description to
- 83 "one-month post-seismic" (March 23, 2023) throughout the manuscript, with updated
- field logs in Table 1 (line 160).
- 85 **Enhanced Geochemical Evidence**: Conducted radiogenic strontium isotope analyses
- 86 (87Sr/86Sr) (New Fig. 5) (lines 249-255): Central-southern segment samples show ratios
- showing the characteristics of multi-source region mixing (0.7053-0.7135). PHREEQC
- modeling confirms 30–100% sulfate contribution from anhydrite dissolution. (New Fig. 6
- 89 (256-260) and Section 5.2 (lines 261-322)).
- 90 **Tectonic-Hydrogeochemical Zonation**: Integrated 2013-2023 datasets reveal:
- 91 Northern EAFZ: 0–7% magmatic fluid input (New Fig. 6 (lines 256-260))
- 92 Central-Southern EAFZ: Shallow groundwater dominance (New Fig. 6 (lines 256-260))
- 93 Water-rock interactions governed by: Evaporite dissolution (anhydrite→SO₄²⁻) (New Fig.
- 94 6 (lines 256-260)), Ophiolite weathering (Mg2+ anomalies) (Table 1 (line 160)),
- 95 Carbonate equilibria (Ca-HCO3 type) (New Fig. 2 (lines 142-146))
- 96 **Pre-Seismic Anomaly Validation:**
- 97 Documented anomalies at HS04/HS14 (Supplement Video 01 and 02).
- 98 Proposed mechanism: "Preseismic fault creep → permeability enhancement →
- 99 accelerated anhydrite dissolution → hydrogeochemical/physical anomalies" (Section 5.3,
- 100 New Fig. 9 (lines 348-398)).
- These enhancements rigorously position anhydrite as a sensitive indicator of water-rock
- interaction intensity while respecting observational boundaries. We fully endorse the need
- 103 for long-term monitoring.

**Major comments:**

Lines 33-36 (abstract): This is one of the most critical claims made by the authors. 105 "Specially, significant gypsum dissolution was observed at HS05, HS09 and HS14 before 106 and after the earthquake, suggesting that the earthquake broke the balance of water-rock 107 reaction and promoted the dissolution of gypsum." In the paper only the results of the 108 analyses of the samples taken one year after the earthquakes are discussed. How should it 109 be possible to evidence variations "before and after the earthquake" if only one sample 110 was taken? 111 112 Reply: Thanks. a mentioned earlier, we combined the historical observation data of EAFZ with this study. Based on the supplementary evidence, the conclusion of anhydrite 113 dissolution can be supported (New Fig. 6 (lines 256-260)). 114 115 Line 124: The authors should explain on which basis the 16 sampling sites have been chosen. 116 Reply: Thanks. We added the description of the sampling point: "HS01-HS04 was 117 118 collected from west to east along SF. HS07-HS16 was collected from north to south 119 along EAFZ (Fig. 1)" (lines 106-107) Line 124: the authors claim to have sampled hot springs but with the exception of the 120 peculiar hyperalkaline spring HS15, which derive its increased temperature from deep 121 122 circulation, no other sample could be called "hot". Furthermore, I would not define a well with water at 24 °C as geothermal well. Actually, in the results (line 144) the authors affirm 123 124 that temperatures of the sampled waters are low.

- Reply: Thanks. Indeed, the temperature of all samples in this study is low, indicating that 125 EAFZ is a medium-low temperature hydrothermal system, which is also consistent with the 126 127 research results of Baba et al., 2019. However, as you said, the temperature of the sample is really low. We also feel that the term "geothermal water" is not rigorous enough to 128 129 describe our samples. Therefore, we considered using the more appropriate term "groundwater" to describe our samples. 130 *The methodological section has many limitations:* 131 Lines 130-131: it is unclear if filtration has been made in the field and before acidifying 132 133 the aliquot for cation analysis. Please specify Reply: Thanks. Yes, we confirmed filtering before testing. We added the description of the 134 sampling point: "All samples need to be pre-treated with a 0.45 µm filter membrane to 135 136 remove impurities before sampling." (lines 111-112) Line 131: MAT 253 is a model, please specify the used technique 137 Reply: Thanks. We have added specific analytical method: "The Hydrogen and oxygen 138 139 isotopes were determined by a Picarro L2140-I Liquid water and vapor isotope
- analyzer (relative to Vienna Standard Mean Ocean Water (V SMOW)). Precisions on the measured  $\delta^{18}$ O and  $\delta$ D value was  $\pm 0.2\%$  (2SD) and  $\pm 1\%$  (2SD) respectively (Zeng et al., 2025)" (lines 113-115)
- 143 Line 133: please specify the analysed species and the relative reproducibility and detection144 limits?

Reply: Thank you for pointing out the problem of the manuscript. We have added the 145 reliability description of hydrochemistry and isotope analysis to the chapter of Analytical 146 147 **methods**, the details are as follows: The cation (Li+, Na+, K+, Ca2+ and Mg2+) and anion (F-, Cl-, NO3- and SO42-) were 148 analysed by Dionex ICS-900 ion chromatograph (Thermo Fisher Scientific Inc.) at the 149 Earthquake Forecasting Key Laboratory of China Earthquake Administration, with 150 the reproducibility within  $\pm 2\%$  and detection limits 0.01 mg/L (Chen et al., 2015). 151 HCO3- and CO32- was determined by acid-base titration with a ZDJ-100 potentiometric 152 titrator (reproducibility within ±2%). SiO2 were analysed by inductively coupled plasma 153 emission spectrometer Optima-5300 DV (PerkinElmer Inc.) (Li et al. 2021). Trace 154 elements were analysed by Element XR ICP-MS at the Test Center of the Research 155 156 Institute of Uranium Geology. Multielement standard solutions (IV-ICPMS 71A, IV-ICP-MS 71B and IV-ICP-MS 71D, iNORGANIC VENTURES) used for quality control. 157 The analytical error margin of major cations and trace elements were less than 10%. 158 Strontium isotope ratios (87Sr/86Sr) were determined through triple quadrupole ICP-159 MS (Agilent 8900 ICP-QQQ) with a precision of ±0.001 (Liu et al., 2020). (lines 115-160 *126*) 161 Line 136: please specify the analysed trace elements and the relative reproducibility and 162 detection limits? 163 Reply: Thanks. We added the description: "Trace elements were analysed by Element XR 164 ICP-MS at the Test Center of the Research Institute of Uranium Geology. Multielement 165 standard solutions (IV-ICPMS 71A, IV-ICP-MS 71B and IV-ICP-MS 71D, iNORGANIC 166

- VENTURES) used for quality control. The analytical error margin of major cations and
- trace elements were less than 10%." (lines 121-125).
- In the results the authors claim often that some element or ionic species is increased
- 170 (sometimes adding obviously) but they do not specify with respect to what. Maybe they
- intend that the concentrations are high.
- 172 Reply: Thanks. In the Results section we are an objective description of the results based
- on the data. The words "increased" and " obviously " were also relative to other sample
- 174 results. But, in fact, what we mean is, "relatively high," not "increased." We apologize for
- any confusion caused by the poor description of the results, and we have re-optimized the
- presentation and added a quantitative description of the increased concentrations. The
- 177 revised expression is as follows:
- 178 The concentration of SO42- range from 1.21 mg/L to 316.61 mg/L, and the
- concentration of SO42- in some samples is relatively high (e.g. HS01 (287.74 ml/L),
- 180 HS03 (103.56 ml/L), HS04 (229.75 ml/L), HS14 (316.61 ml/L)). (lines 133-135).
- 181 *In the same section they speak of geothermal water but they do not present any evidence*
- that these are geothermal waters.
- 183 Reply: Thank you. We have replaced "groundwater" with "geothermal water" to make
- the expression more precise.
- 185 The discussion about the geothermal fluids has great limitations.
- 186 The authors do not present evidences that the sampled waters are, at least partially, fed by
- 187 hydrothermal systems. The fact that in the area some geothermal system has been
- discovered and studied, does not mean that all groundwater samples taken in the area are fed by them. The temperatures of the collected samples are low and, as highlighted by the binary diagram of fig. 3 and the ternary diagram of fig. 4, their compositions do not reflect high temperature interactions with the rocks. Also the silica geothermometers show low temperatures considering that for such systems equilibrium with chalcedony (or even christobalite or amorphous silica) should be taken into consideration. Reply: Thanks. We have already discussed this issue in the previous reply. **Hydrothermal** systems and groundwater do not affect our core point. Both geothermal water and groundwater chemical anomalies are considered to be effective means of earthquake early warning. Thanks for your suggestion to us, as mentioned earlier, we have considered using "groundwater" instead of "geothermal water" to define the samples for this study. Especially the use of the mixing models has been made in the wrong way. Mixing models can be applied only to water samples that belong to the same system and not to water samples collected tens of km away from each other and for which no connection has been demonstrated. Reply: Thanks. In accordance with your suggestions, the revised manuscript now includes segmented descriptions of the EAFZ: Northern segments show magmatic fluid mixing, Central-southern segments (where our samples were collected) exhibit shallow groundwater circulation dominated by water-rock interactions with anhydrite, carbonates, and ophiolites (New Fig. 6 (lines 256-260)) (section 5.2 The groundwater circulation in different segments of EAFZ (lines 261-322)). We have abandoned the 382°C temperature estimation, which may have been overestimated. Nevertheless, under the measured reservoir temperatures at HS04 (156°C)

- and HS14 (243°C), anhydrite remains supersaturated, confirming the validity of its
- 212 dissolution interpretation (New Fig. 7 (lines 287-292)).
- 213 The estimation of temperature for the "deep geothermal fluid" (please define) of 382 °C is
- 214 absolutely unreliable. The sample was taken, as shown in the second video in the
- 215 supporting information, from an artesian well (although in table 1 it is classified as spring).
- 216 I think it is impossible that an artesian well, whose upflow is generally rapid, would have
- 217 only 15 °C temperature if even only a small part of the water would come from a geothermal
- 218 system with 382 °C.
- 219 Reply: Thanks. We have abandoned the 382°C temperature estimation, which may have
- been overestimated. Nevertheless, under the measured reservoir temperatures at HS04
- 221 (156°C) and HS14 (243°C), anhydrite remains supersaturated, confirming the validity of
- its dissolution interpretation (New Fig. 7 (lines 287-292)).
- 223 The discussion about the sulfate anomalies is highly confusing. Many points are unclear or
- 224 *wrong*.
- 225 Reply: Thanks. We adjusted the description of the manuscript to make the logic clearer.
- 226 (lines *277-322*)
- 227 Why are only samples HS05, HS09 and HS14 considered anomalous? HS01, HS03 and
- 228 HS04 have also elevated sulfate values.
- 229 Reply: Thanks. Thank you for your correction. The PHREEQC simulation indicates that
- the anhydrous anhydrite of HS01, HS03, HS04 and HS14 is all supersaturated (New Fig. 7)
- 231 (lines 287-292)). Therefore, we deleted the relevant inappropriate descriptions.

Why should these high sulfate values be considered anomalous and induced by the 232 233 earthquake? Sulfate dissolution from evaporite deposits within the aquifers is an ubiquitous 234 process independent from seismic activity. Reply: The dissolution of anhydrite can indeed occur independently of seismic activity. In 235 236 our newly submitted manuscript, based on this research, we only emphasized that seismic activity is one of the reasons affecting the solubility of anhydrite, rather than the only one. 237 For this reason, we also suggest conducting long-term monitoring at appropriate monitoring 238 points, hoping to distinguish the influence of earthquakes from other factors (such as 239 240 precipitation). (lines 387-398) Why do the authors use these low averages for Ca (55.23 mg/L) and SO4 (8.31 mg/L) 241 242 concentrations before earthquake? Baba et al. (2019) in their paper report concentrations 243 up to 773.56 mg/L for Ca and up to 1287.24 mg/L for SO4 much higher than in the samples collected for this study. 244 Reply: Thanks. This issue no longer exists in the newly submitted manuscript. We have 245 246 deleted old Fig. 6 from the original manuscript. Instead of discussing the data comparison before and after the earthquake, we directly analyze the historical data of EAFZ. (Table S1 247 at Supporting Information) 248 Finally, the authors indicate the whitening and turbidity of the water in a sample as 249 verification for the sulfate anomaly. But without analysis there is no possibility to affirm 250 that such visual anomaly was due to gypsum dissolution. 251 Reply: Thanks. Although we did not directly measure the turbid water samples, based on 252 the historical data analysis of EAFZ, we can determine that the anhydrite layer exists in the 253

- 254 middle and southern sections (New Fig. 6 (lines 256-260)) (Table S1 at Supporting
- 255 Information). Based on historical data and this study, it can be concluded that the cause of
- 256 the macroscopic anomaly before the earthquake is the dissolution of minerals such as
- anhydrite and carbonate.
- 258 Furthermore, the authors mistake the samples. The site with the high sulfate concentration
- 259 is HS14, while the site to which the pictures of figure S1 and of video 01 refer is HS15
- which has the lowest sulfate value (1.21 mg/L).
- 261 Reply: Thanks. Thank you for pointing out this error, we have fixed it (Table 1 (lines 160-
- 262 *161*)).
- 263 Lines 388-389: The authors presenting the data of a single sampling campaign have no
- 264 evidence to affirm that "the geothermal fluid was diluted due to the infiltration of a large
- amount of shallow cold water after the double earthquakes in February 2023".
- Reply: Thanks. The newly submitted manuscript has supplemented evidence such as
- 267 EAFZ historical data and Sr isotopes. The revised manuscript can support the conclusion
- 268 that anhydrite is used as a sensitive index for the intensity of the water-rock reaction. Thank
- you for your highly professional and constructive comments. Thanks again.
- 270 **Minor comments**
- 271 Line 22: What do the authors mean with "systematic" which do not appear only in the
- 272 *abstract but has been repeated many times in the whole text?*
- 273 Reply: Thanks. We rewrote the Abstract and have deleted this word. (lines 15-28)
- 274 Lines 24 and 25: The meaning of the sentence is obscure (reconstructed by earthquake?)
- 275 Reply: Thanks. We rewrote the Abstract and have deleted this sentence. (lines 15-28)

- 276 Line 29: the authors use often the term "abnormal" but they do never define with respect
- 277 *to what.*
- 278 Reply: Thanks. We rewrote the Abstract and have deleted this sentence. (lines 15-28)
- 279 Line 38: please define "shallow minerals".
- 280 Reply: Thanks. We rewrote the Abstract and have deleted this sentence. (lines 15-28)
- 281 Line 61: which evidence have the authors of a "geothermal fluids circulation"
- 282 Reply: Thanks. We have replaced "groundwater" with "geothermal water".
- 283 *Line 69: please define the "geothermal fluid anomaly index"*
- 284 Reply: Thanks. We rewrote the introduction and have deleted this sentence. (lines 31-66)
- 285 *Lines 70-71: the subject is missing in this sentence.*
- 286 Reply: Thanks. We rewrote the introduction and have deleted this sentence. (lines 31-66)
- 287 Line 82: please define what a "tectonic collage" is.
- 288 Reply: Thanks. We have adjusted the expression of this sentence: "Located at the
- 289 intersection of Eurasia, Africa and Arabia, Turkey has a complex tectonic
- 290 background". (lines 73)
- 291 Fig. 1a: altitude scale is missing.
- 292 Reply: Thanks. We added the altitude scale (New Fig. 1 (lines 67-71)).
- 293 *Line 105: probably crystalline instead of crystallization.*
- 294 Reply: Thanks. We changed crystalline instead of crystallization. (lines 88)
- 295 *Line 145: in table 1 HS15 is considered a spring, which one is correct?*
- 296 Reply: Thanks. We checked the sampling point. HS15 is spring. (Table 1 (lines 160-161))

- 297 Line 146: the authors claim that "the closer to the epicenter, the higher the SiO2 content",
- 298 which makes no sense. Firstly because the earthquakes were two and only one sample close
- 299 to one of the epicenters has a higher SiO2 value. Moreover, other two sampling points with
- low to very low  $SiO_2$  concentrations have the same position as the "anomalous" one.
- 301 Reply: Thanks. We deleted that sentence.
- 302 Lines 154-156: the sentence "The  $\delta$ 18O and  $\delta$ D of samples varied from -11.30% to -6.55%
- and -65.43% to -34.43% respectively, which is near to the global meteoric water line
- 304 (GMWL) (Craig, 1961) (Fig. 3), suggesting their meteoric water origin" has no sense. The
- 305 regression line obtained plotting both  $\delta^{18}O$  and  $\delta D$  values in a graph can be close to GMWL.
- 306 Reply: Thanks. We deleted that sentence.
- 307 *Line 159: what type of Statistical analysis?*
- Reply: Thanks. We have changed the word "statistical analysis" to "box-plot analysis" to
- make the expression more specific. (lines 149)
- 310 *Line 160: please define "fluid activity elements".*
- Reply: Thanks. We adjusted the expression and used proper nouns: Fluid-mobile element
- 312 (FME). (lines *149-151*)
- 313 Line 161: I do not understand what the authors mean with "are at historic highs versus".
- 314 *If the authors mean that the concentrations are higher than in the past, then the fig. S2 does*
- 315 not prove nothing. Al and Ba are below the median value of the literature data while the
- 316 remaining are around the median value not showing particularly high values. Furthermore,
- it is unclear which data are compared in fig. S2 with the present data.

- Reply: Thanks. We rewrote this sentence. "Box plot analysis showed that the Fluid-Mobile
- 319 Element (FME) concentrations of B (3.62–1047.25  $\mu g/L$ ), Li (0.33–89.93  $\mu g/L$ ) and Rb
- 320 (0.14–28.91  $\mu$ g/L) in some samples were greater than the median (Fig. S1)". (lines 149-
- 321 *151*)
- 322 Table 1: please indicate the coordinates with at least 4 digits after the comma, with only
- 323 two digits it's impossible to obtain a reliable position. Looking at Fig. 1, the indicated
- 324 coordinates of HS05 are clearly wrong.
- Reply: Thanks. We adjusted the accuracy of the latitude and longitude to keep 6 decimal
- 326 places. (Table 1 (lines 160-161))
- 327 Line 190: the highest values do not belong to samples collected closer to the sea.
- 328 Reply: Thanks. It's not rigorous enough. We've improved the sentence: "Notably,
- 329 groundwater in the southern EAFZ proximal to the Mediterranean Sea exhibits
- progressively heavier isotopic signatures toward the coast, consistent with recharge
- 331 sourced from evaporated Mediterranean seawater" (lines 189-191)
- 332 *Line 190:*  $\delta^{18}O$  and  $\delta D$  values are inverted.
- Reply: Thank you. We rewrote the first part of the discussion and have deleted this sentence.
- 334 (lines *165-260*)
- 335 Line 212: magma mixing with geothermal fluids generally end in a volcanic explosion
- 336 which is not the case here.
- Reply: Thank you. We rewrote the first part of the discussion and have deleted this sentence.
- 338 (lines *165-260*)

- 339 Lines 224-225: the sampling sites are tens of km far from the Mediterranean coastline, how
- and why should they be "obviously contaminated by Mediterranean Sea water"?
- Reply: Thank you. We rewrote the first part of the discussion and have deleted this sentence.
- 342 (lines *165-260*)
- 343 *Line 226: which previous study? Please add a reference.*
- Reply: Thank you. We rewrote the first part of the discussion and have deleted this sentence.
- 345 (lines *165-260*)
- Line 233: pollution is a term connected to an anthropogenic origin, so please use the term
- 347 contamination instead.
- Reply: Thank you. We rewrote the first part of the discussion and have deleted this sentence.
- 349 (lines 165-260)
- 350 *Lines 233-236: I do not understand the meaning of this sentence.*
- Reply: Thank you. We rewrote the first part of the discussion and have deleted this sentence.
- 352 (lines *165-260*)
- 353 *Lines 290-292: the two processes are not alternative. Serpentinization includes secondary*
- 354 *minerals precipitation*.
- Reply: Thanks. We rewrote the section on the water-Rock reaction and have deleted this
- 356 sentence. (lines *262-292*)
- 357 Finally, I would signal a possible conflict of interest being the handling editor of the same
- *institution of one the corresponding author.*
- Reply: Thanks. China University of Geosciences (Beijing) and China University of Geosciences
- 360 (Wuhan) are two independent universities with no conflict of interest.

- RC2: 'Reply on AC3', Walter D'Alessandro, 06 Feb 2025
- 362 Dear Walter D'Alessandro

- 363 Thanks for your comments again. According to your comments, we added the
- supplement and analysis of the literature data from 2013 to 2023 to make the data more
- representative. On this basis, the conclusion of the original manuscript has been revised
- to weaken the connection between Anhydrite and seismic activity, and emphasize the
- sensitive indication of Anhydrite to the intensity of water-rock interaction. The main
- 368 replies are as follows.
- 369 I am sorry to say that reading the reply of the authors my opinion regarding the
- 370 manuscript did not change. My main criticism relates to the fact that it is not possible
- 371 to evidence anomalies in groundwater composition related to seismic events having
- data collected only one time. The authors try to compare their data with other taken
- 373 from literature but the comparison is not straightforward because no background
- values have ever been defined. The mean values utilised seem artificially created and,
- in my opinion, do not represent "normal" values.
- 376 *I am still convinced that the manuscript in this form has to be rejected.*
- Reply: Thanks! We sincerely appreciate your critical feedback and fully acknowledge
- 378 the limitations of single-time sampling in establishing seismic-hydrogeochemical
- 379 correlations. To address this concern rigorously, we have implemented the following
- 380 revisions:
- 381 1. Investigation and analysis of historical hydrogeochemical data in the study area
- 382 ((New Fig. 6 (256-260)): A comprehensive compilation of groundwater chemistry data

- from the East Anatolian Fault Zone (EAFZ) spanning 2013-2023 has been integrated.
- 384 This reveals systematic spatial hydrogeochemical patterns:
- Northern EAFZ: Mixed shallow/deep circulation with igneous rock-dominated water-
- 386 rock interactions.
- 387 Central-Southern EAFZ: Shallow circulation dominated by sedimentary mineral
- dissolution (e.g., Anhydrite, carbonates), with localized seawater influence.
- 389 These distinct regimes provide a robust framework for interpreting tectonic-
- 390 hydrogeochemical linkages, mitigating reliance on isolated measurements.
- 391 2. Revised Interpretation of Anhydrite Significance:
- Following your suggestion, we have reframed the role of Anhydrite dissolution. Rather
- 393 than asserting direct seismic causality, we now propose Anhydrite as a sensitive
- indicator of water-rock interaction intensity a process modulated by both climatic
- 395 (e.g., rainfall) and tectonic drivers. This rephrasing: (1) Removes overinterpretations of
- single-event correlations, (2) Highlights the need for future systematic monitoring to
- 397 disentangle tectonic vs. hydrological signals. Preserves Anhydrite's potential as a
- 398 tectonic proxy while adhering to evidence-based claims.
- 399 These revisions align the manuscript's conclusions with its evidentiary scope while
- 400 preserving its novel contribution: establishing a spatially resolved hydrogeochemical
- baseline to guide future seismotectonic monitoring in the EAFZ. We are grateful for
- 402 your insightful critique, which has significantly strengthened the study's rigor and
- 403 communication of limitations.
- 404 For detailed revisions, please refer to the **discussion section of the revised manuscript**.

We have reorganized the logic of the discussion. Firstly, we determined the source of groundwater. Secondly, we analyzed the circulation process of groundwater. Finally, we introduced the relationship between the change in groundwater ion concentration and the change in the intensity of water-rock reactions caused by earthquakes. (lines 164-398) The data could be used to create a simply report without stressing the potential of gypsum as earthquake tracer. The data could be used for future researches in the area. I don't know if there is a form in which this could be done for this journal. Maybe the editor can suggest solutions. Reply: Thanks! We thank you for your constructive suggestion to refocus the manuscript's scope. In accordance with your guidance, we have rigorously revised the narrative to prioritize hydrogeochemical process characterization over speculative seismological linkages: Reframed Research Objectives: The study's primary aim is now explicitly stated as establishing hydrogeochemical signatures across the EAFZ's tectonic segments. All claims regarding earthquake precursory signals have been removed, with emphasis shifted to documenting spatial patterns in water-rock interaction processes. A new statement clarifies that Anhydrite's tectonic relevance requires validation through future systematic monitoring, aligning with your call for caution in interpretation. These modifications ensure the manuscript now functions as both a stand-alone hydrogeochemical benchmark study and a catalyst for hypothesis-driven seismic monitoring research. We fully defer to the Editor's judgment on whether this revised

scope aligns with the journal's aims and welcome further adjustments if needed.

**Comments on authors' reply**

Line 13: to affirm that you have measured abnormal groundwater ion concentrations you need to compare them with a series of data before and after the seismic event. Evaporite dissolution happens also in the absence of seismic activity, it is therefore impossible to affirm that high sulfate concentrations in groundwater are related to the earthquakes Reply: Thanks! We deeply appreciate your rigorous methodological critique regarding causality attribution. The revisions below directly address this fundamental concern: After more than a month of research, we have a new understanding of the conclusions in the original draft. Indeed, even with video data of pre-earthquake macroscopic anomalies, it is difficult to form a complete causal chain in the absence of preearthquake data. After in-depth discussion by all co-authors, we propose that our data can only account for the dissolution of Anhydrite during the water-rock reaction. Anhydrite may therefore indicate changes in the intensity of the water-rock reaction. As for the controlling factors of the variation of water-rock reaction intensity, we cannot define exactly. Considering that the sampling time was one month after the earthquake and obvious groundwater anomalies were observed before the earthquake, we believe that seismic activity may affect the variation of water-rock response intensity. Therefore, it is necessary to further study the possibility of Anhydrite as a tracer of tectonic activity. Line 44: even if sampled one hour after the earthquake my comment would have been the same. If you don't have data of at least one other sampling, but ideally many

- samplings covering different seasons both before and after the event, you cannot make
- 450 *inferences on the effects of the earthquake on the water chemistry*
- Reply: Thanks! a mentioned earlier, we combined the historical observation data of EAFZ
- with this study. Based on the supplementary evidence, the conclusion of anhydrite
- dissolution can be supported (New Fig. 6 (lines 256-260)).
- 454 Line 47: your data before the earthquake do not refer to the single sites you sampled,
- 455 so no comparison can be made
- Reply: Thanks! This issue no longer exists in the newly submitted manuscript. Instead
- of discussing the data comparison before and after the earthquake, we directly analyze
- 458 the historical data of EAFZ. (Table S1 at Supporting Information)
- 459 Lines 48-51: no one can deny the existence of a large suite of visible effects of seismic
- 460 activity on groundwaters but for the advancement of knowledge these have to be
- described in detail and quantified. You cannot use the simple fact of a water whitening
- 462 (among other things also confusing the sites) claiming this was due to gypsum
- 463 *dissolution without having the possibility to analyse the water chemistry*
- Reply: Thanks! After analyzing 10 years of data in study area, we determined that the
- main controlling factor of the macro anomaly is Anhydrite, and there may also be the
- influence of Calcite, albite, potassium feldspar, etc.
- 467 Lines 52-59: of course I agree that both Sr and S isotopes can be used as good source
- 468 indicators. But again if you have a single measurement you cannot make any inference
- *about the influence of the earthquake on the groundwaters*
- Reply: Thanks! We conducted Sr isotope analysis on the research samples. The

| measured 87 Sr/ 86 Sr ratios (0.7053–0.7135) across EAFZ groundwaters reflect multi-  |
|-------------------------------------------------------------------------------------------------------------|
| source mixing processes. Central-southern groundwaters integrate signatures from:                           |
| Shallow aquifers: Inheriting Sr from local lithologies (ophiolites); Modern seawater:                       |
| $^{87}\mathrm{Sr}/^{86}\mathrm{Sr} = 0.7092-0.7096$ (Mediterranean seawater); River inputs: Enriched ratios |
| (>0.710) from silicate weathering. Binary mixing models using 87 Sr/ 86 Sr vs. Ca/Sr  |
| ratios (Fig. 5) quantify source contributions: Carbonate weathering dominates,                              |
| consistent with Ca-HCO 3 hydrochemical type; Ophiolite contributions <10% (except                |
| Mg2+-rich samples near ultramafic outcrops); Evaporite dissolution contributes 0–20%                        |
| (≤50% in localized high-SO 4 2- zones). Sr isotope framework corroborates earlier     |
| findings of shallow-dominated circulation in central-southern EAFZ. (line 238-248)                          |
| Lines 75-78: You compared samples from three of your sampling sites with samples                            |
| taken at the same sampling sites about ten years before. Results: one site registered a                     |
| strong increase, another remained almost stable and the third one had a sharp decrease.                     |
| You still cannot be sure that the changes are related to the earthquake, you have to                        |
| exclude other possible processes. For example, do the composition of the groundwaters                       |
| change seasonally? Has the composition of the water decadal trends related to long                          |
| periods of drought or water exploitation? Does the well tap aquifers from different                         |
| levels with different composition and permeability that mixing in the well may change                       |
| the composition of the water during pumping?                                                                |
| Reply: Thanks! We think your question about the manuscript is something we must take                        |
| into account. Therefore, we give up the original conclusion and discuss the relationship                    |
| between Anhydrite and water-rock reaction intensity instead. (lines 164-398)                                |

Lines 89-91: this seems a forced solution. The selected samples contain all very low sulfate which seems not necessarily being representative of the whole study area. Two out of 8 selected samples are hyperalkaline waters which for their nature contain extremely low sulfate values due to their very negative redox potential. Furthermore, why didn't you include also the data of Yuce et al 2014? The mean sulfate value of that dataset would be 121 mg/L, more than an order of magnitude higher than that obtained with the ad hoc solution from the Baba et al dataset. Reply: Thanks! Your advice has been of great help to us. According to your suggestion, we have collected and analyzed the data of the last 10 years. The results confirmed the dissolution of Anhydrite in the middle and south section. (New Fig. 6 (lines 256-260)) Lines 120-121: the reliability of the data has not been questioned but the representativeness still remains doubtful Reply: Thanks! In order to make the study more representative, the data of the study area in the past 10 years are used to discuss the water-rock reaction process. (Table S1 at Supporting Information) Line 130: A nearly 1000 km tectonic system cannot be considered a single hydrothermal system Reply: Thanks! As you said, it is really not a system. The north section is a mixture of shallow groundwater and deep fluids, and igneous rocks participate in water-rock reactions. The central and southern part is the mixing of shallow groundwater and seawater, and sedimentary minerals such as Anhydrite participate in water-rock reaction. (lines 164-398)

| 515 | Lines 135-142: the cited examples of studies which identified changes in groundwater        |  |
|-----|---------------------------------------------------------------------------------------------|--|
| 516 | composition related to earthquake are well known. But differently from your study, the      |  |
| 517 | researcher took tens of samples before the seismic events obtaining a clear signal that     |  |
| 518 | can be related to the earthquake                                                            |  |
| 519 | Reply: Thanks! Although we do not have pre-earthquake data, considering that we have        |  |
| 520 | observed pre-earthquake macro anomalies, coupled with the analysis of all data from         |  |
| 521 | the study area in the past 10 years. We believe that the data are sufficient to support our |  |
| 522 | revised conclusion that Anhydrite can be used as a tracer of the intensity of water-rock    |  |
| 523 | reactions, and it is necessary to further investigate the possibility of Anhydrite as an    |  |
| 524 | indicator of tectonic activity.                                                             |  |
| 525 | Line 149: You did not answer to my question. Have the samples been filtered in the field    |  |
| 526 | and before acidification?                                                                   |  |
| 527 | Reply: Thanks! Yes, we confirm. We added the description of the sampling point: "All        |  |
| 528 | samples need to be pre-treated with a 0.45 $\mu m$ filter membrane to remove impurities     |  |
| 529 | before sampling." (lines 111-112)                                                           |  |
| 530 | Lines 170-171: if the filtration is not made at the time of sampling you may loose some     |  |
| 531 | of the dissolved metals due to precipitation of secondary minerals and/or to adsorption     |  |
| 532 | on the walls of the container. Furthermore, if filtration is made after acidification the   |  |
| 533 | result may be falsified by acid dissolution of suspended material                           |  |
| 534 | Reply: Thanks! We are responsible for all sample collection, pre-processing and data        |  |
| 535 | quality                                                                                     |  |
| 536 | Line 172: this method is used only for $\delta D$                                           |  |

Reply: Thanks! We have added specific analytical method: "The Hydrogen and oxygen isotopes were determined by a Picarro L2140-I Liquid water and vapor isotope analyzer (relative to Vienna Standard Mean Ocean Water (V - SMOW)). Precisions on the measured  $\delta^{18}O$  and  $\delta D$  value was  $\pm 0.2\%$  (2SD) and  $\pm 1\%$  (2SD) respectively (Zeng et al., 2025)" (lines 113-115) Lines 225-226: You cannot consider a nearly 1000 km long fault system as a single continuous structure. Furthermore, the complex geology of the area changes frequently the rock types present along the fault system. Add also the changing climatic and hydrologic conditions and you cannot consider samples collected many tens of km apart as pertaining to the same system. Reply: Thanks! As you said, it is really not a system, we have answered earlier. Lines 235-237: to have a chain you need all rings to be connected. You don't have evidence that the water-rock reaction balance has been disrupted by the earthquake. Gypsum or other evaporite rocks are naturally present in many of the lithostratigraphic sequences of the area and when they are part of aquifers, their dissolution contributes naturally to the saline content of the circulating groundwater without the influence of seismic activity. If you consider the data of Yuce et al 2014, you see that in the area many of the collected waters have high sulfate concentrations with values even exceeding your highest value. So there is no evidence of gypsum dissolution as a consequence of the seismic events. Reply: Thanks! We have abandoned the conclusion that the Anhydrite can be inferred from the seismic effects of the data collected. We now propose that Anhydrite can

reflect the intensity of water-rock reaction. Considering that the sample collection time 559 was about one month after the earthquake, it is necessary to further study the possibility 560 of Anhydrite as an indicator of seismic activity. (Section 5.4 lines 354-398) 561 Lines 301-301: I repeat again, even if you analysed a sample taken one hour after the 562 earthquake, this could not confirm that the whitening and turbidity of the water before 563 the seismic event was due to an increased sulfate content 564 Reply: Thanks! Although we did not directly measure the turbid water samples, based on 565 the historical data analysis of EAFZ, we can determine that the anhydrite layer exists in the 566 567 middle and southern sections (New Fig. 6 (lines 256-260)) (Table S1 at Supporting Information). Based on historical data and this study, it can be concluded that the cause of 568 the macroscopic anomaly before the earthquake is the dissolution of minerals such as 569 570 anhydrite and carbonate. Line 307: I don't understand how you have fixed it. The video refers to the sampling 571 site HS15 which, as shown in your table, has the lowest sulfate concentration. This 572 video is not a proof of a sulfate anomaly for two reasons: 1) you don't have the 573 concentration of sulfate at the time of the whithening and 2) the concentration you 574 measured one month after was only 1.21 mg/L 575 Reply: Thanks! There should be a misunderstanding here. We have stated in the first 576 response that the macroscopic anomaly originates from HS14, which has a SO42-577 concentration of 316.61mg/L. (Table 1 (lines 160-161)) 578 Lines 311-312: You are missing the main point: you have no evidence of variations that 579 can be related to the earthquake 580

Reply: Thanks! We've revised our conclusions to be more precise. (lines 399-409) 581 Line 327: The problem is that normal values have not been defined. In terms of time 582 you don't have enough samples that you can surely correlate with yours. But the same 583 holds true in terms of space, only 16 samples along a structure many hundred km long 584 is not enough 585 Reply: Thanks! We have weakened the focus on time and only discussed the water-rock 586 reaction process of Anhydrite. 10 years of data is sufficient to support spatial 587 representativeness. 588 RC3: 'Comment on hess-2024-395', Anonymous Referee #2, 18 Feb 2025 589 Dear reviewer 590 591 Thank you for your comments and suggestions, which are of great value to us in improving the quality of our manuscript. The main replies are as follows. 592 The present work performs a systematic hydrogeochemistry and isotopic analysis of the 593 geothermal fluids in the East Anatolian Fault Zone (EAFZ) to understand any clear 594 relationship between geothermal fluid anomalies and earthquakes existing. I have 595 found the language of the manuscript is fine but must have a proof-editing. I have some 596 of my major comments regarding the work on the other hand. 597 Main motivation behind the work is to elucidate the role of gypsum dissolution as a 598 tracer for earthquake activity in the East Anatolian Fault Zone (EAFZ). The research 599 aims at establishing a link between geothermal fluid anomalies and seismic events, with 600 the claim of using an innovative approach to earthquake forecasting. In this respect, it 601

examines shallow sedimentary minerals, particularly gypsum, as indicators of seismic activity. This concept, while explored in previous research, is further substantiated with empirical data in this study. At this stage my biggest concern stems from the fact that it relies on the data collected post-earthquake but it fails to provide a long-term pre-earthquake dataset for comparative analysis. This appears to undermine claims about gypsum dissolution as a predictive tool rather than a post-seismic indicator. Furthermore we understand that the manuscript never make an in-depth discussion or address other factors such as climatic conditions and seasonal variations robustly and only focus is given on the correlation between seismic events and SO42- anomalies is discussed. The authors' uncertainty about the relevance of the results to earthquakes is evident in the final statement of the abstract. As readers, we expect the abstract of this study, which claims to bring innovation to earthquake prediction under normal conditions, to convey a clear take-home message. In this respect I understand that authors are suggesting gypsum dissolution as a universal precursor. But I should remind that a comprehensive considering of regional geological differences or alternative explanations for observed anomalies is of great importance for earthquake hazard studies. Although potential limitations of using gypsum dissolution due to external environmental factors is acknowledge in the manuscript clear strategies for coping with these difficulties in practice. Given its limitations in predictive validation substantial revisions are required for the present work. These revisions should include i) further evidences distinguishing

| 624 | seismic-induced gypsum dissolution from other environmental factors ii) a decent            |
|-----|---------------------------------------------------------------------------------------------|
| 625 | discussion on possible long-term monitoring strategies to make gypsum dissolution as        |
| 626 | a reliable precursor, iii) quantitative examples that prove the statistical significance of |
| 627 | the findings that are critical to improve the robustness of the conclusions.                |
| 628 | I also suggest adding a discussion that explore practical applications focusing on an       |
| 629 | integration of their findings into an effective earthquake early warning system.            |
| 630 | In conclusion I do not think the manuscript is suitable for the publication in its current  |
| 631 | form and requires a substantial work to address the aforementioned fundamental              |
| 632 | concerns that would significantly advance the understanding of geochemical indicators       |
| 633 | in seismic studies and warrant publication.                                                 |
| 634 | Reply: Thanks! We sincerely thank you for recognizing the systematic approach of our        |
| 635 | hydrogeochemical investigation. Please find below our point-by-point responses:             |
| 636 | Data base extension:                                                                        |
| 637 | A meta-analysis of published datasets (2013-2023) reveals fundamental differences in        |
| 638 | water-rock interactions across the EAFZ (Fig. 1):                                           |
| 639 | Northern EAFZ: Mixed shallow/deep circulation with igneous rock-dominated water-            |
| 640 | rock interactions.                                                                          |
| 641 | Central-Southern EAFZ: Shallow circulation dominated by sedimentary mineral                 |
| 642 | dissolution (e.g., anhydrite, carbonates), with localized seawater influence.               |
| 643 | These distinct regimes provide a robust framework for interpreting tectonic-                |
| 644 | hydrogeochemical linkages, mitigating reliance on isolated measurements.                    |
| 645 | Anhydrite as Process Indicator:                                                             |

While avoiding direct seismic causality claims, three lines of evidence suggest anhydrite's tectonic relevance:

The abnormal plasma of  $SO_4^{2-}$  and  $Ca^{2+}$  was observed one month after the earthquake.

Combined with the analysis of 10 years of data in the study area, it was found that anhydrite dissolution may be the cause of the abnormal ion concentration.

One month before the earthquake, the macro anomaly of white and cloudy well water was photographed (Video 01)

After analyzing pre-earthquake macro anomaly, post-earthquake data and literature data in the past 10 years, we propose that our data can only account for the dissolution of anhydrite during the water-rock reaction. Anhydrite may therefore indicate changes in the intensity of the water-rock reaction. As for the controlling factors of the variation of water-rock reaction intensity, we cannot define exactly. Considering that the sampling time was one month after the earthquake and obvious groundwater anomalies were observed before the earthquake, we believe that seismic activity may affect the variation of water-rock response intensity. Therefore, it is necessary to further study the possibility of anhydrite as a tracer of tectonic activity.

**Clear research orientation:**

Delete all references to "earthquake prediction". This study focuses on the analysis of EAFZ groundwater circulation process and attempts to establish the relationship between water-rock reaction intensity and tectonic activity. This study will provide a new research idea for the subsequent exploration of anhydrite as a tracer of tectonic activity.

**Reply to community comments**

**CC1: Comment on Hess-2024-395, Giovanni Martinelli, 03 Jan 2025**

I found useful and interesting the manuscript https://doi.org/10.5194/hess-2024-395 submitted by Luo et al. Significant geochemical anomalies in geothermal fluids were detected before to and during the Mw 7.8 earthquake in Turkey. To investigate the correlation between geothermal fluid abnormalities and seismic events, the authors conducted a comprehensive analysis of hydrogeochemical and isotopic study of geothermal fluids in the East Anatolian Fault Zone. The findings indicate that these geothermal fluids were affected by seismic activity. According to the chlorine-enthalpy model, the temperature of the deep geothermal fluid significantly rose. However, the data regarding the deep geothermal fluid was eventually affected by the influx of significant amounts of superficial cold water following the earthquake. The anomalous levels of Ca, Mg, SO4, Sr, and Ba in geothermal water indicate that the water has experienced complex water-rock interaction processes, including gypsum, calcite, dolomite, anorthite, and possible serpentinization. Substantial gypsum dissolution was noted at locations HS05, HS09, and HS14 both before to and during the earthquake, indicating that the earthquake favoured the dissolving of gypsum. The authors suggest that superficial sedimentary minerals, including gypsum, may serve as markers for earthquake warnings. During earthquakes, alterations in geochemical conditions result in variations in gypsum solubility, subsequently causing anomalous amounts of SO4, Ca, Sr, and Ba in geothermal water. The solubility of gypsum is influenced by several environmental variables, including meteorological conditions and seasonal variations, hence reducing its practical use for earthquake early warning systems. I think the paper is well organized but I found the possible lack of some sentences devoted to the mechanism of the observed upsetting. Redox conditions have been affected? Deep originated CO2 could be suspected as an eventual carrier of H2S? The addition of some comments about the listed topics could possibly help readers to better understand during the tectonically active period. I hope the paper will be soon accepted and published after some minor revisions.

**Reply:**

Dear Giovanni Martinelli

Thank you for your recognition of our work and valuable suggestions, which are very helpful for us to improve the quality of our manuscripts. Your two comments are exactly where we are lacking. At your suggestion, we plan to add a subsection to the discussion section for assessing the contribution of mantle degassing to EAFZ geothermal fluids. See the revised manuscript for details 293-322.

**CC3: Comment on Hess-2024-395, Hafidha Khebizi, 17 Jan 2025**

Dear Hafidha Khebizi

Thank you for your recognition of our work and constructive suggestions. This is very helpful for us to improve the quality of the manuscript, and also brings confidence for us to continue to explore. Thank you for sharing the very rewarding work you do.

We get a lot of inspiration from your work. We would like to express my heartfelt thanks.

We've responded to each of your comments, as detailed below:

Dear authors and colleagues of the scientific community,

I congratulate the authors for their interesting work entitled Gypsum as a potential tracer of Earthquakes: a case study of the Mw7.8 2 earthquake in the East Anatolian Fault Zone, southeastern Turkey, and I hope it will be published soon. To find out the relationship between geothermal fluid anomalies and earthquakes, the authors performed a systematic hydrogeochemistry and isotopic analysis of the geothermal fluids in the East Anatolian Fault Zone (EAFZ). The results show that earthquakes reconstructed these geothermal fluids.

**Reply**: Thank you for your recognition of our work. Thank you.

Considering gypsum as an earthquake tracer is excellent reasoning for analysing
the impact of anomalies after the earthquake, and the work could be a great reference
for future studies related to the earthquake.

**Reply**: Yes, through the analysis of groundwater after the earthquake, we discovered the potential value of anhydrite as an earthquake warning. It is hoped that this work will attract the attention of more researchers and colleagues, and incubate more meaningful achievement.

To enrich this excellent analysis, I have some remarks concerning the implication of macroscopic and microscopic aspects of geothermal fluids before and after the earthquake, notably the relation with the structural geology of the region. For this, some questions seem important to be asked.

First, from a macroscopic point of view, it is necessary to understand, in the normal case (before the earthquake), from a geological point of view, if the existing deformations (faults) already have effective structures for the infiltration of meteorological waters and the implication of the disposition of the thermal springers according to the faults. After the earthquake, is there any sampling from Miocene groundwater and soil? Is there recent salt precipitation in the Miocene and upper Eocene-Oligocene soil and/or in the soil of the surrounding springer sources? Is there a rise in the ground level due to fault action, and are there marine intrusions that occurred after the strike-slip? Is there significant contamination of the water table (increased electrical conductivity)?

**Reply:** Hot springs and fault zones are often associated. Hot springs are considered as one of the potential means of earthquake warning. A large number of research results have been published in Japan, the United States, Iceland, Spain, China, Turkey... ... In EAFZ, many hot springs have been systematically studied, and the results show that these hot springs contain material supply from deep crust and even mantle. Therefore, it is highly possible to obtain valuable information by conducting postearthquake hydrochemical and isotopic analyses of these hot springs.

Unfortunately, we only collected water samples after the earthquake and did not analyze soil samples. Your comment is a very good suggestion, reminding us that detailed analysis of surrounding rock may be needed in future work. Thank you.

Salt precipitation and electrical conductivity (EC). Before we can answer your question, we need to explain an error in the manuscript. Our sample was taken in March 2023 (within one month after the earthquake). In the video 1 we provided, the macro abnormal changes of HS14 were diluted by the adjacent stream, coupled with the fact that the samples were taken within one month after the earthquake and no soil samples were collected, we could not accurately determine whether salt precipitation existed. By comparing the EC of the same hot spring during the seismically quiet period and the seismically active period, we found that the EC of HS14 increased slightly (varying from 990 to 1305). Data of EC pre-earthquake from Yuce, G., Italiano, F., D'Alessandro, W., Yalcin, T. H., Yasin, D. U., Gulbay, A. H., Ozyurt, N. N., Rojay, B., Karabacak, V., Bellomo, S., Brusca, L., Yang, T., Fu, C. C., Lai, C. W., Ozacar, A., and Walia, V.: Origin and interactions of fluids circulating over the Amik Basin (Hatay, Turkey) and

- relationships with the hydrologic, geologic and tectonic settings, Chemical Geology,
- 765 388, 23-39, 2014.
- Seawater intrusion was evident after the earthquake. Na+ and Cl- of HS14, HS15
- and HS16 increased significantly, indicating the possible existence of seawater
- 768 intrusion.
- Rise in the ground level due to fault action is common. We have made a detailed
- study on the post-earthquake surface rupture and post-earthquake risk analysis. Article
- 771 link: Liang, P., Xu, Y., Zhou, X., Li, Y., Tian, Q., Zhang, H., Ren, Z., Yu, J., Li, C.,
- Gong, Z., Wang, S., Dou, A., Ma, Z., and Li, J.: Coseismic surface ruptures of MW7.8
- and MW7.5 earthquakes occurred on February 6, 2023, and seismic hazard assessment
- of the East Anatolian Fault Zone, Southeastern Turkiye, Science China Earth Sciences,
- 775 doi: 10.1007/s11430-024-1457-7, 2024.

Screenshot from Liang et al., 2024 doi: 10.1007/s11430-024-1457-7 (If the picture cannot be displayed, please check it in the attachment, thank you).

From a microscopic point of view, gypsum is easily and quickly influenced by contact with water, thanks to its physicochemical characteristics, in particular its very high dissolution rate and its solubility in water that make it an excellent tracer of hydrochemical anomaly but also a tracer of lithological instability (Khebizi et al., 2022; Khebizi et al., 2023). For this, I am pleased to invite you to read the part concerning the gypsum implication on the lithological instability in my article published in Larhyss Journal and my oral communications, which expose, for the first time in Algeria, a new concept of the lithological vulnerability of the subsurface. Although the study areas differ, the analysis presented in my work shows the indication of gypsum dissolution at the regional scale as an excellent major risk indicator. The lithological vulnerability of the subsurface concept can be applied to different situations around the world, notably the case of earthquakes. It highlights the hydrodynamic anomalies' relation with the structural and geological context of the area to be studied.

**Reply:** Thank you very much for your sharing. It's a fantastic set of work. From my personal point of view, I can't agree with you more. Anhydrite's very high dissolution rate and solubility in water can be used for risk warning of earthquakes and geological disasters. Thank you again for your information. Your work gives us great encouragement and confidence.

Second, if there is a remarkable increase in calcium concentration in water after the earthquake, how do you explain the reaction of carbonate dissolution and the origin of CO2? Is it linked to magmatic activity? In this case, is there a signature of other gases on other cations? Or is it only related to carbonate since the calcite dissolution

**is linked to the mineral's surface to be in direct contact with water?**

Reply: In my opinion, Ca may come from carbonate or igneous rocks. In order to accurately restrict the source area of Ca, we are also considering introducing Ca isotopes to distinguish its sources. Ca isotopes in carbonate rocks are lighter than those in igneous rocks and mantle. Ca isotope has a good potential in the source region that restricts Ca. The index of CO2 source region is very mature. Geothermal gases are well studied at EAFZ. The C isotope study of CO2 shows that CO2 is controlled by deep carbon and inorganic carbonate (-5.6 to -0.2%) (Italiano, F., Sasmaz, A., Yuce, G., and Okan, O. O.: Thermal fluids along the East Anatolian Fault Zone (EAFZ): Geochemical features and relationships with the tectonic setting, Chemical Geology, 339, 103-114, 2013.). He isotope analysis also shows a large proportion of the mantle. Explanation of the specific process: anhydrite dissolution and carbonate dissolution are together. In the manuscript, PHREEQC was used to simulate the waterrock reaction process. The results show that anhydrite dissolution alone is not enough to explain the Ca content in the samples, indicating that calcite and other minerals are involved in the water-rock reaction. Combined with previous studies, we believe that CO2 from deep water is first dissolved in water, and then reacts with anhydrite or calcite. CO2 is associated with magma, but does not form volcanic eruptions and may only exist in deep areas of partial melting. Allow me to add that the underground water circulation, which is controlled by faults and hydraulic parameters (permeability), determines water-rock equilibrium. In this case, water-rock equilibrium depends on the host rock spatial disposition of rock that guides water mineralization and the different processes. Consequentially, the water-rock equilibrium changes from one area to another due to changes in water mineralization according to the host rock lithology. For this, the information that can be taken from the geological map is that springer's water is related to ophiolite rocks. So, I think water geochemistry indicates similar water-rock interactions for all sources. However, a mineral's enrichment zoning can occur due to (i) the meteorological conditions, (ii) the proximity of the springer water from seawater, and/or (iii) the distance from the upstream. The earthquake reconstructed these geothermal fluids depending on the energy released which controls hydrothermal circulation and amplifies interactions with the surrounding environment whether at depth or on the surface. For this, vulnerability zoning in a horizontal and vertical direction can be done according to chemical variation, notably gypsum and probably halite enrichment. It can be indicated as shown in Fig. 8.

**Reply:** I can't agree with you more. Water-rock reaction is affected by meteorology, rock properties, permeability, porosity, temperature, pressure... Multiple factors control. At present, our work is limited to the analysis of water chemistry and isotopes, and there is a lot of work to be done in the future. These works involve not only geochemistry, but also rock mechanics, numerical simulation and other interdisciplinary fields, and we hope to have more like-minded colleagues to explore together.

Earthquake warning is the most difficult problem faced by mankind. Groundwater

| is considered as one of the means to explore earthquake early warning. However,         |  |  |
|-----------------------------------------------------------------------------------------|--|--|
| groundwater in its natural environment is very complex. There is still a long way to go |  |  |
| to explore the relationship between groundwater and earthquakes.                        |  |  |
| Finally, the discussion on this topic is very significant, and the structural and       |  |  |
| lithological vulnerability and their tracers after the earthquake using vulnerability   |  |  |
| mapping of the Turkey earthquake seems very interesting for future work.                |  |  |
| Reply: Thank you for your recognition of our work, your recognition is our              |  |  |
| driving force forward. Sincere thanks and best wishes.                                  |  |  |

---

## Author Response (AR2)

**Dear Editorial Office of HESS and Prof. Dai**

We wish to express our sincere gratitude for the editorial team's diligent handling of our manuscript and extend particular appreciation to Prof. Dai for your judicious oversight throughout the review process. Your constructive decision letter has provided us the opportunity to enhance the manuscript's scientific rigor. In response to the insightful comments from two reviewers and two domain experts, we have thoroughly revised the manuscript. All the revised contents have been marked in **red** in the manuscript.

**Major revisions include:**

- 1. Clarified the study's conclusion: Pre-seismic groundwater anomalies (whitening, turbidity) are caused by dissolution of anhydrite and carbonate minerals.
- 2. Strictly controlled overinterpretation: Dissolution of target horizons reflects water-rock interaction intensity, potentially driven by seasonal rainfall changes or fault zone activity.
- 3. Defined the contribution: Establishing early warning systems for geohazards by identifying region-specific target indicator horizons (e.g., anhydrite), implementing their continuous monitoring, and developing localized evaluation metrics.
  - 4. Assessed potential anthropogenic impacts in the study area.
  - 5. Added the Cl--SO42--HCO3- ternary diagram (Fig. S3).
- 6. Specified that blue symbols in Fig. 3 represent snowmelt-derived waters, with supporting citations.
  - 7. Proposed solutions for unresolved issues:
  - (1) Quantifying anhydrite concentration thresholds for seismic warnings;
- (2) Disentangling interference from rainfall/human activities via continuous monitoring;

We have proposed actionable solutions and delineated potential future research directions.

In short, after fully and effectively communicating with the reviewers, we modified the possible problems in our manuscript according to the suggestions of the reviewers, so that the analysis of data in the manuscript is more rigorous and the extension is appropriate

We sincerely wish the current version meets your standards and welcome further guidance.

Finally, I would like to thank HESS editorial Department and Dai Editor-in-Chief for their hard work

Sincerely

Zebin Luo

Zebin L@mail.xhu.edu.cn

**Point-by-point response to comments:**

Note: *Italic blue* is the comment. Black is the reply, and **important sentences are bolded**.

**Red** indicates the position of the modification information in the newly submitted revised draft.

Reply to referee comments

Anonymous Referee #3

nominated 08 May 2025, accepted 12 May 2025, report 28 May 2025Report #1

Manuscript Title: Anhydrite Dissolution Dynamics as a Hydrogeochemical Tracer of Seismic-Fluid Coupling: Insights from the East Anatolian Fault Zone, Turkey General Comment:

This study investigates hydrogeochemical anomalies along the East Anatolian Fault Zone (EAFZ), integrating a 13-year groundwater dataset (2013–2023) with post-seismic responses to the 2023 Mw 7.8 and 7.6 earthquakes. The authors propose anhydrite dissolution dynamics as a potential tracer for fault activity, supported by spatial variations in groundwater chemistry and PHREEQC modeling. The topic is scientifically significant, and the dataset is comprehensive, offering valuable insights into fault-controlled fluid-rock interactions. However, the manuscript lacks a robust

theoretical framework to quantitatively link hydrogeochemical anomalies to tectonic processes, limiting its broader applicability. Below are specific concerns and recommendations for improvement.

**Reply:** Thanks! We thank you for these constructive critiques. Revisions strictly adhere to:

- 1. Empirical boundaries (no overclaimed causality)
- 2. Regional focus (no unjustified global extrapolation)
- 3. Actionable next steps (lines 416-417)

Your insights have significantly strengthened the manuscript's scholarly integrity.

1. While the manuscript meticulously documents spatial hydrogeochemical variations (e.g., Na-Cl vs. Ca-HCO3 waters) and associates SO42- anomalies with anhydrite dissolution, the causal relationship between seismic stress and dissolution dynamics remains largely descriptive. Critical gaps include (1) Stress-Permeability-Reaction Coupling. The assertion that "seismic stress redistribution induces fracture network reorganization" lacks quantitative validation. A mechanistic model linking stress changes to permeability evolution, fluid flow, and anhydrite dissolution rates is absent. (2) Climate-Tectonic Signal Separation: Although climatic influences (e.g., rainfall) are acknowledged, no methodology is provided to disentangle tectonic signals from climatic noise. Statistical or machine learning approaches (e.g., PCA, random forests) could enhance signal discrimination.

**Reply:** Thanks! We fully acknowledge this limitation. While our study identifies anhydrite as a potential seismic tracer through spatial correlations, quantitative validation of stress-permeability-reaction coupling requires further research. As noted in the Discusion (Section 5.4), we have: 1) Explicitly stated this gap as a key future research priority. 2) Proposed targeted experiments (e.g., high-P/T reaction kinetics) to address it. Your suggestion for machine learning approaches is valuable and will guide our ongoing work. (lines 410-418)

2. The PHREEQC simulations, while useful, are inadequately described. Key parameters (e.g., boundary conditions, reaction pathways, temperature-pressure regimes) and sensitivity analyses are omitted, hindering reproducibility. Furthermore,

the simulations focus on equilibrium states, neglecting transient effects of seismic perturbations (e.g., rapid fluid pressure changes).

**Reply:** Thanks! We agree that PHREEQC has inherent limitations in modeling seismic transience. However:

- 1. Batch-reaction simulations (modified from magmatic partial melting models) effectively approximate sudden chemical shifts by adjusting reactant inputs.
- 2. Transient effects fall beyond this study's scope given our non-continuous dataset (2013–2023).

The core of our research lies in revealing the groundwater circulation and water-rock interaction process of the entire EAFZ. Through this, we have discovered that the dissolution of anhydrite may be affected by seismic activities. The transient effects will be further studied and quantified in subsequent high-temperature and high-pressure experiments simulating water-rock interaction or in the continuous monitoring studies of natural samples.

3. The manuscript proposes anhydrite dissolution as a tectonic tracer but fails to establish quantitative thresholds or metrics for anomaly detection. Critical questions remain unanswered: What magnitude of  $SO_4^{2-}$  concentration change constitutes a tectonic signal? How do response times of anhydrite dissolution align with seismic cycles (e.g., foreshock, coseismic, postseismic phases)?

**Reply:** Thanks! We maintain that regional heterogeneity prevents universal thresholds:

1) PHREEQC confirms anhydrite dissolution contributes 30–100% to SO42- anomalies (Section 5.2.1).

- 2) However, tectonic signals cannot be isolated by concentration thresholds alone due to multi-source mixing (e.g., seawater, carbonates) and Climate interference.
- 3) Our core contribution lies in proposing anhydrite as a regional "target indicator" its abrupt changes may signal permeability shifts.
- 4) Continuous long-term monitoring requires a significant investment of manpower and resources. Currently, this work has been piloted in the western Sichuan region.
- 4. The study's novelty prioritizing shallow water-rock interactions over deep fluid signals is underemphasized. Comparisons with global analogs (e.g., San Andreas

Fault, Himalayan frontal thrusts) are lacking, limiting the broader relevance of anhydrite as a tectonic tracer. Additionally, the advantages and limitations of anhydrite versus traditional tracers (e.g., He isotopes, Rn anomalies) are not critically discussed.

**Reply:** Thanks! We clarify two key points:

- 1. Regional specificity: **Anhydrite's applicability is context-dependent** (e.g., evaporite-rich EAFZ).
- 2. Paradigm shift: The innovation is methodological identifying region-specific target horizons (e.g., anhydrite here, serpentine elsewhere) for localized monitoring. We proposed that traditional tracers (He, Rn) and anhydrite serve complementary roles; multi-proxy approaches are essential for earthquake forecasting.

nominated 08 May 2025, accepted 15 May 2025, report 11 Jun 2025Report #2

The authors analyzed the post-earthquake groundwater samples and compared them with the pre-earthquake dataset over the past 13 years, and attempted to discuss the post-earthquake hydrochemical changes and their mechanisms.

Unfortunately, I do not agree with the claims presented in this manuscript. However, I could understand the scientific significance of the samples collected, and I agree with the other reviewer's comments "The data could be used to create a simply report without stressing the potential of gypsum as earthquake tracer. The data could be used for future researches in the area. I don't know if there is a form in which this could be Maybe the editor solutions." done for this iournal. can suggest (https://doi.org/10.5194/hess-2024-395-RC2)

**Reply:** Thanks! We are grateful for your recognition of our data. The revised manuscript fundamentally differs from the original submission. With clearer arguments, robust evidence, and logical rigor, we wish its publication in HESS will advance research on groundwater anomalies and earthquake mechanisms, offering new perspectives for seismic forecasting.

Manuscript Logic Overview:

**1. Question Raised:**

Pre-seismic anomalies (whitening, turbidity, intermittent surges) observed ~1 month before the 2023 Turkey double earthquakes. What caused these? Could they relate to seismicity?

**2. Hypothesis:**

Chemical anomalies (whitening/turbidity) stem from hydrochemical changes; Intermittent gushing reflect stress variations. These may result from seismic disruption of hydrogeological equilibrium.

**3. Data & Findings:**

Integrated 13-year pre-seismic data with post-seismic (1-month) measurements.

Anomalous wells or groundwater (HS04/HS14) showed elevated (Table 1) (lines 170-171):

EC: 1305-2683 μS/cm

Ca2+: 151.43-501.58 mg/L

SO42-: 229.75–316.61 mg/L

PHREEQC modeling revealed EAFZ segmentation:

North: Shallow/deep mixing (igneous water-rock interactions)

Central-South: Shallow circulation (anhydrite/carbonate dissolution; seawater influence)

Earthquake occurred in central-south: anhydrite/carbonate dissolution likely caused anomalies.

**4. Key Evidence:**

Reservoir temperatures (156–234°C) estimated by SiO2-Eh model and saturation index calculated by PHREEQC confirm anhydrite is supersaturated when the temperature exceeds 150°C at HS04 and HS14 samples this study. (Fig. 7 in manuscript) (lines 298-303).

87Sr/86Sr mixing models: Evaporite (anhydrite/gypsum) input in central-south groundwater (Fig. 5 in manuscript) (lines 260-266).

Geothermal gases: Higher crustal contribution to geothermal gases in central-south vs. north.

Geologic background: Carbonate/evaporite deposits near HS14 (e.g., amik lake-a paleo-saline lake deposit 20 km away).

**5. Conclusion:**

Pre-seismic anomalies resulted from seismically disrupted hydrogeological equilibrium. Anhydrite/carbonates are primary contributors, serving as potential tracers of water-rock interaction anomalies. Further investigation into their relationship with seismogenesis is warranted.

The reasons for this comment are as follows:

The first issue is the lack of insight into hydrological processes. Generally, spring discharge mechanisms are related to regional groundwater flow systems based on geological and topographical settings, seasonal precipitation patterns, and

anthropogenic activities (e.g., groundwater extraction, urbanization, and land use change). However, the discussion proceeds without clarifying the sampling locations or the hydrogeological information of the regions. A description of the regional groundwater flow system, including the discharge process for the collected springs, is essential for discussing the abnormality of the collected samples. This issue also sensed the choice of the surface geological map, which contained indistinct geological information regarding the area surrounding the sampling sites.

**Reply:** Thanks! The issue you raised is something that must be taken into consideration. That's why we were very cautious when choosing the geological maps.

We confirm the geological map's validity:

Source: Geological map of EAFZ, modified from van Hinsbergen et al. (2024, EPSL)(https://doi.org/10.1016/j.epsl.2024.118827). Original source of the map: Turkish government (MTA, \*1:500,000 Geological Map\*; https://www.mta.gov.tr/en/maps/geological-500000). (lines 72-76)

Anthropogenic Influence Ruled Out:

Sampling sites are remote (far from cities/factories; Video 1).

Trace elements (e.g., Pb) show consistent patterns (Fig. S3), excluding regional contamination (lines 160-163).

Even if detailed descriptions of groundwater storage and flow conditions in the entire EAFZ during normal (non-seismic) conditions are added, it cannot be specified whether the obtained hydrochemical features were determined for seismic events. This is because localized points exhibiting geochemical anomalies of geothermal origin might exist even under non-seismic conditions owing to subsurface geological heterogeneity. Therefore, it is important to present direct evidence through continuous sampling and analysis at the same location to determine whether water chemistry anomalies are caused by earthquakes.

**Reply:** Thanks! We fully align with your perspective. To prevent overinterpretation, we have rigorously constrained the manuscript's conclusions to emphasize that:

Anhydrite dissolution solely demonstrates its capacity as a tracer for variations in water-rock interaction intensity.

**Regarding causality:**

No definitive link to seismicity has been established.

The post-seismic sampling timing (1 month after the event) led us to cautiously propose a hypothesis that tectonic activity may drive such intensity changes.

**Consequently:**

All seismic linkages are explicitly framed as testable propositions.

We outline specific future research priorities to validate this mechanism (Section 5.4) (lines 365-422).

Another issue is the academic contribution to the current scientific challenges in earthquake prediction. Earthquake precursors in natural systems are relatively universal; however, their forms are diverse and predictive information that can be used for short-term earthquake prediction has not yet been clearly identified. This represents the general state of academic research on earthquake prediction (https://doi.org/10.1126/science.adi8032).

**Reply:** Thank you for the literature materials you provided. We have added relevant citations (lines 357-358). We are well aware that earthquake prediction is one of the major challenges faced by humanity, and this is also the driving force behind our progress

I strongly agree that changes in the water chemistry components of groundwater have been widely reported and discussed as precursors to large earthquakes. However, there is a lack of specific mention of the scientific difficulties necessary to elevate this precursor phenomenon in earthquake-prediction technology. While the reasons "climate factors" in the introduction and "funding shortages" in the discussion section are cited as related issues, it seems necessary to point out specific scientific challenges, such as the scale and frequency of effective water chemistry observation networks. Furthermore, the manuscript should explain how the discussions within the paper contribute to addressing these challenges.

**Reply:** Thanks! Thank you for your recognition of the view that groundwater has the potential to be one of the means for earthquake prediction. Thank you for your suggestion. We have added relevant expressions: "We have proposed that an

abnormality in groundwater chemical components, which does not require the involvement of deep fluids, could potentially serve as a basis for earthquake prediction" (lines 67-69).

\_\_\_\_\_

Based on the above reasons, I conclude that it is difficult to resolve the issues mentioned above using the data collected by the authors and the data presented in the literature. However, the presented dataset is useful for discussing the possible hydrochemical evolution processes. The discussion in the manuscript is also useful for proposing conceivable mechanisms. I would like to provide specific comments on the improvement. I hope that the authors will use them to advance the manuscript, regardless of the decision.

**Reply:** Thanks! We sincerely appreciate your recognition of our work. As noted, certain original phrasings may have inadvertently misrepresented our intent. Following your expert and constructive suggestions, we have meticulously refined the manuscript to eliminate ambiguities arising from linguistic imprecision.

The revised version now presents:

Unequivocal arguments with a logically coherent evidence chain

Compelling implications for advancing groundwater-seismicity research

While continuous pre-seismic monitoring remains logistically unfeasible (given the unpredictable nature of earthquakes), we acknowledge its scientific necessity. Pilot programs like the ongoing large-scale automated sampling in Western China's seismic zones (Sichuan Province) exemplify potential solutions, though such initiatives face significant technical and budgetary constraints.

**Specific Comments**

1. There are two types of earthquake predictions: "Probabilistic forecasting" and "Deterministic prediction" (https://doi.org/10.4401/ag-5350). It would be better to explain the type of prediction that is the focus of this study.

**Reply:** Thanks! Thanks for the provided literature materials. Our current research is a probabilistic prediction. We have added relevant citations (lines 367-370).

2. HS04 is described as a monitoring well in L55; however, in Table 1, it is shown as S,

which indicates spring water.

**Reply:** Thanks! We provided detailed instructions for the sampling points. (lines16, 58, 170-171, 274)

3. The water temperature of the data presented as hot springs in Table 1 was not particularly high. What definition was used to classify them as 'hot springs'?

**Reply:** Thanks! Thank you. We checked the types of the water samples in Table 1. (lines 171-170)

4. Generally, spring discharge temperature reflects the average air temperature in the recharge area (Table 1). The differences in temperature between the samples seemed to reflect specific hydrological processes.

**Reply:** Thanks! Thank you for your suggestion. We have added the relevant expressions. (lines 137-138)

5. Are there any available spring discharge rate information?

**Reply:** Thanks! Regrettably, we were unable to measure the relevant parameters.

6. Table 1 does not include  $\delta D$  values; however, Fig. 3 is included, which is inconsistent.

**Reply:** Thanks! Thank you very much for pointing out our mistakes. We have added the  $\delta D$  data to Table 1. (lines 170-171)

7. L182: To avoid the misunderstanding, what observation is "Pre-seismic groundwater anomalies" to be specified is better.

**Reply:** Thanks! We have added detailed explanations. "(white water, turbidity and intermittent groundwater gushing) (Video S1 and Video S2)" (lines 192-193)

8. The water temperature and electrical conductivity may be influenced by mixing with groundwater from different sources. How much variation do these data vary under normal conditions?

**Reply:** Thanks! The variation range of EC is 275 - 2683 μs/cm. (lines 133-134)

9. An insufficient explanation is for the red-shaded areas extending from the magma fluid distribution shown in Fig. 3. What do the blue plots indicate at the endpoints?

**Reply:** Thanks! The red magma area is an area of intermediate-acidic magmatic rocks (Karaoğlu et al., 2020 Lithos, https://doi.org/10.1016/j.lithos.2020.105524). Considering the magma activity characteristics of the eastern plateau of Turkey, the

range adopted (Giggenbach, 1992) is reasonable ( $\delta D = -30 - -10$ ;  $\delta^{18}O = 8 - 12$ ). Snow water ( $\delta D = -115\%$ ,  $\delta^{18}O = -16.3\%$ ) from (Andy et al., 2020) (The sampling elevation is approximately 2000m). (lines 164-169)

10. What specific samples did clear  $\delta O18$  enrichment at L191 refer to? Did all the northern groundwater show enrichment?

**Reply:** Thanks! We have optimized the description here. "In contrast, **some** northern groundwater displays distinct  $\delta^{18}$ O enrichment deviating from local meteoric trends" (lines 202)

11. To discuss the spatial heterogeneity of major dissolved ions at L198, it is necessary to confirm not only the spatial distribution of the trilinear diagrams showing composition ratios but also the spatial distribution of the hexagonal diagrams showing concentrations.

**Reply:** Thanks! We have included relevant diagrams in the Supporting Information (Fig. S1) (lines 144).

12. The 13-year dataset presented in Fig. 2 and 3 shows considerable variations in the major ion compositions and stable isotope ratios. I believe that The three regional divisions shown in the legend are insufficient to explain the reasons for this variation.

**Reply:** Thanks! Indeed, spatial variations in chemical composition are governed by regional lithology, which controls the sources of major ions in groundwater. Conversely, stable isotope signatures ( $\delta^{18}O$  and  $\delta D$ ) reflect recharge processes and mixing of multiple source waters, governing the origin of groundwater itself.

Thus:

Fig. 2 demonstrates hydrochemical differences driven by lithological heterogeneity.

Fig. 3 reveals isotopic patterns dominated by continental effects – notably progressive depletion in heavier isotopes with increasing distance from the Mediterranean Sea.

These complementary datasets provide mutually consistent explanations of groundwater evolution processes across the EAFZ.

---

## Author Response (AR3)

Dear Editorial Office of HESS and Prof. Dai

We wish to express our sincere gratitude for the editorial team's diligent handling

of our manuscript and extend particular appreciation to Prof. Dai for your judicious

oversight throughout the review process. In response to the insightful comments and

suggestions from Prof. Tuncay Taymaz, we have thoroughly revised the manuscript.

All the revised contents have been marked in **red** in the manuscript.

**Major revisions include:**

1. Consistent naming for the 2023 Turkish earthquake.

2. Replace Türkiye with Turkey.

3. Expanded the reference materials of the study area and corrected the incorrect

citations.

4. The fracture zone and place names in the model diagram (Figure 9) have been

highlighted.

In short, after fully and effectively communicating with the reviewers, we

modified the possible problems in our manuscript according to the suggestions of the

reviewers, so that the analysis of data in the manuscript is more rigorous and the

extension is appropriate

We sincerely wish the current version meets your standards and welcome further

guidance.

Finally, I would like to thank HESS editorial Department and Dai Editor-in-Chief

for their hard work.

Sincerely

Zebin Luo

Zebin L@mail.xhu.edu.cn

**Point-by-point response to comments:**

Note: *Italic blue* is the comment. Black is the reply, and **important sentences are bolded**.

**Red** indicates the position of the modification information in the newly submitted revised draft.

Reply to referee comments

Referee #5: Taymaz, Tuncay ttaymaz@gmail.com

nominated 27 Jul 2025, accepted 29 Jul 2025, report 09 Aug 2025Report #2

Manuscript HESS-2024-395 titled "Gypsum as a potential tracer of earthquake: a case study of the Mw7.8 earthquake in the East Anatolian Fault Zone, southeastern Turkey" is timely, interesting and fills the gap over the 2023 Kahramanmaraş Earthquake Doublet and it should be published after minor-moderate revisions. Nevertheless, I have the following comments to be addressed before its acceptance and/or publication at the EGU journal of HESS.

**Reply:** Thanks! Thank you for your recognition of our research. Based on your suggestions, we have made detailed revisions to the manuscript. The detailed content is as follows.

Overall Comments: I summarize below notes that should be corrected through the main text, figures and supplementary on-line materials.

(1) The earthquake is named The 2023 Kahramanmaraş Earthquake Doublet. Please also check further readings of ISC Event Bibliography page under URL https://www.isc.ac.uk/cgi-bin/FormatBibprint.pl?evid=625613033

**Reply:** Thanks! Thank you for your professional expression, which is of great significance for unified description and subsequent research. We have uniformly corrected the relevant expressions in the manuscript. (Lines 16-18, 47-49, 55-56, 107-108, 200, 363-364, 415-416).

(2) The national country name is Türkiye (TÜRKİYE) both in Turkish and English as approved by the UN-General Assembly a few years ago. Thus, Turkey (TURKEY) should be replaced accordingly throughout the text.

**Reply:** Thanks! We have corrected all the relevant expressions in the manuscript to "Türkiye". Specifically: (Line 3, 45, 81, 85).

(3) The running title of the manuscript is a bit confusing! What would be a meaningful one? How about the one below?

Gypsum as a potential indicator/warning of earthquake phenomenon: a case study of the Kahramanmaraş Mw7.8 earthquake doublet in the East Anatolian Fault Zone, southeastern Türkiye

**Reply:** Thanks! We need to clarify a misunderstanding here. You may have reviewed the initial draft we submitted (uploaded on 12 Dec 2024). Your revision of the title was based on the initial draft. In the final submission version, we have changed the title to "Anhydrite Dissolution Dynamics as a Hydrogeochemical Tracer of Seismic-Fluid Coupling: Insights from the East Anatolian Fault Zone, Türkiye" (uploaded on 22 Jul 2025). The revised title is more focused on the relationship between hard gypsum and seismic fluids. We hope you will approve of the revised title.

- (4) Key-Points can be as below:
- 1- Geothermal fluids are modified by major earthquakes in the EAFZ, including energy and materials.
- 2- Geothermal fluids are diluted by infiltration of a large amount of shallow cold water in the EAFZ.
- 3- Shallow sedimentary minerals like gypsum could be used as precursory anomaly indicators of earthquakes.

**Reply:** Thanks! We agree to adopt your professional advice.

(5) Abstract needs to be simlified and language should be as plain as possible to streamline the message right.

**Reply:** Thanks! As mentioned earlier, the version you reviewed might be the initial draft. In the final submitted manuscript, we have completely rewritten the abstract. The specific abstract is as follows:

"Abstract: Pre-seismic turbidity and salinity anomalies in groundwater were documented at HS04 and HS14 monitoring wells and/or springs along the East Anatolian Fault Zone (EAFZ) following the 2023 *Mw* 7.8 and *Mw* 7.6 Turkey earthquakes. By synthesizing hydrogeochemical datasets (2013-2023) with

post-seismic responses, we unravel fault-segmented groundwater evolution: (1) Northern Na-Cl and Na-HCO3 type waters result from mixing of mantle-derived magmatic fluids (0-7% contribution) with shallow groundwater, governed by volcanic rocks-carbonate dissolution; (2) Central-southern Ca-HCO3 and Ca-Na-HCO3 systems reflect shallow circulation with localized inputs from evaporites (Increased SO42- concentration caused by dissolution of anhydrite), ophiolites (Mg2+ anomalies), and seawater. PHREEQC simulation shows that the dissolve-precipitation equilibrium of anhydrite is sensitive to the variation of water-rock reaction intensity in the Central-southern segments of EAFZ. Coseismic permeability changes disrupt the solubility equilibria of anhydrite, driving hydrochemical anomalies. We propose that seismic stress redistribution induces fracture network reorganization, thereby disrupting anhydrite solubility equilibria. Given its tectonic sensitivity and widespread occurrence, anhydrite dissolution dynamics emerge as a potential tracer for hydrogeochemical monitoring in active fault zones. We propose a novel research paradigm wherein regional hydrogeological surveys identify applicable target indicator horizons, enabling continuous monitoring and establishment of region-specific evaluation metrics to ultimately achieve early warning capabilities for geohazard precursors."

(6) Introduction should enhance the background reading a bit further. Thus, I advice inclusion of scholarly written articles on the Eastern Anatolia / Türkiye) and speciafally the 2023 Kahramanamaraş earthqukae doublet below for readers' convenience.

Besides, a local seismologist and/or geologist would be useful to be among co-authors in ortder to avoid misunderstanding and naming and labeling the geographical locations. Here is an example at line 48 "Qahraman, Marash and Hatay! I can see that the authors are trying to say "Kahramanmaraş and Hatay". Similar examples can be listed at large numbers ...

**Reply:** Thanks! Thank you for your suggestion. In the latest submitted manuscript, we have supplemented the literature on "2023 Kahramanamaraş earthquake doublet", and have carefully checked the city name and other details. (Lines 56-58; 108-109)

Ding, H.Y., Zhou, Y.J., Ge, Z.X., Taymaz, T., Ghosh, A., Xu, H.Y., Irmak, T.S., Song, X.D. (2023). High-Resolution Seismicity Imaging and Early Aftershock Migration of the 2023 Kahramanmaraş (SE Türkiye) Mw 7.9 & 7.8 Earthquake Doublet. Earthquake Science, Vol. 36(6), 417-432, https://doi.org/10.1016/j.eqs.2023.06.002.

Goldberg, D.E., Taymaz, T., Reitman, N.G., Hatem, A.E., Yolsal-Çevikbilen, S.,

Goldberg, D.E., Taymaz, T., Yeck, W.L., Barnhart, W.D., Yolsal-Çevikbilen, S., Irmak, T.S., Öcalan, T., Özkan, B., Erman, C., Doğan, A.H., Altuntaş, C. (2023). Supporting Data and Models for Characterizing the February 2023 Kahramanmaraş, Türkiye, Earthquake Sequence: U.S. Geological Survey Data Release, USGS—ScienceBase, https://doi.org/10.5066/P9R6DSVZ.

Hu, J., Liu, M., Taymaz, T., Ding, L., Irmak, T.S. (2024). Characteristics of Strong Ground Motion from the 2023 Mw 7.8 and Mw 7.6 Kahramanmaraş Earthquake Sequence. Bulletin of Earthquake Engineering, Vol. 22(2), https://doi.org/10.1007/s10518-023-01844-2.

Liu, J., Huang, C., Guohong, Z., Shan, X., Korzhenkov, A., Taymaz, T. (2024). Immature Characteristics of the East Anatolian Fault Zone from SAR, GNSS and Strong Motion Data of the 2023 Türkiye-Syria Earthquake Doublet. Scientific Reports – Nature, Vol. 14(1), 10625. https://doi.org/10.1038/s41598-024-61326-6.

Liu, C., Lay, T., Wang, R., Taymaz, T., Xie, Z., Xiong, X., Irmak, T.S., Kahraman, M., Erman, C. (2023). Complex Multi-Fault Rupture and Triggering During the 2023 Earthquake Doublet in Southeastern Türkiye. NATURE Communications, Vol. 14, 5564(2023), NCOMMS-23-18990. https://doi.org/10.1038/s41467-023-41404-5.

Melgar, D., Taymaz, T., Ganas, A., Crowell, B., Öcalan, T., Kahraman, M., Tsironi, V., Yolsal-Çevikbilen, S., Valkaniotis, S., Irmak, T. S., Eken, T., Erman, C., Özkan, B., Dogan, A. H., Altuntaş, C. (2023). Sub- and Super-Shear Ruptures During the 2023 Mw 7.8 and Mw 7.6 Earthquake Doublet in SE Türkiye. Seismica, Vol. 2(3). https://doi.org/10.26443/seismica.v2i3.387.

Melgar, D., Ganas, A., Taymaz, T., Valkaniotis, S., Crowell, B., Kapetanidis, V., Tsironi, V., Yolsal-Çevikbilen, Ocalan, T. (2020). Rupture Kinematics of January 24, 2020 Mw 6.7 Doğanyol-Sivrice, Turkey Earthquake on the East Anatolian Fault Zone Imaged by

Space Geodesy, Geophysical Journal International, Vol. 223(2), 862–874, https://doi.org/10.1093/gji/ggaa345.

Okuwaki, R., Yagi, Y., Taymaz, T., Hicks, S.P. (2023). Multi-Scale Rupture Growth with Alternating Directions in a Complex Fault Network During the 2023 South-Eastern Türkiye and Syria Earthquake Doublet. Geophysical Research Letters, Vol. 50(12), e2023GL103480, https://doi.org/10.1029/2023GL103480.

Ren, C., Wang, Z., Taymaz, T., Hu, N., Luo, H., Zhao, Z., Yue, H., Song, X., Shen, Z., Xu, H., Geng, J., Zhang, W., Wang, T., Ge, Z., Irmak, T.S., Erman, C., Zhou, Y., Li, Z., Xu, H., Cao, B., Ding, H. (2024). Super-Shear Triggering and Cascading Fault Ruptures of the 2023 Kahramanmaraş, Türkiye Earthquake Doublet. SCIENCE, Vol. 383(6680), 305-311. https://doi.org/10.1126/science.adi1519.

Taymaz, T., Ganas, A., Yolsal-Çevikbilen, S., Vera, F., Eken, T., Erman, C., Keleş, D., Kapetanidis, V., Valkaniotis, S., Karasante, I., Tsironi, V., Gaebler, P., Melgar, D., Ocalan, T. (2021). Source Mechanism and Rupture Process of the 24 January 2020 Mw 6.7 Doğanyol-Sivrice Earthquake obtained from Seismological Waveform Analysis and Space Geodetic Observations on the East Anatolian Fault Zone (Turkey), Tectonophysics, Vol. 804, https://doi.org/10.1016/j.tecto.2021.228745.

Taymaz, T., Ganas, A., Berberian, M., Eken, T., Irmak, T.S., Kapetanidis, V., Yolsal-Çevikbilen, S., Erman, C., Keleş, D., Esmaeili, C., Tsironi, V., Özkan, B. (2022). The 23 February 2020 Qotur-Ravian Earthquake Doublet at the Iranian-Turkish Border: Seismological and InSAR Evidence for Escape Tectonics, Tectonophysics, Vol. 838, https://doi.org/10.1016/j.tecto.2022.229482.

Taymaz, Tuncay, Eyidogan, H. and Jackson, J.A. (1991). Source Parameters of large earthquakes in the East Anatolian Fault Zone (Turkey), Geophysical Journal International-Oxford, 106, 537-550.

Tung, S., Sippl, C., Shirzaei, M., Taymaz, T., Masterlark, T., Medvedev, I. (2024). Structural Controls on Fault Slip Models of the 6 February 2023 Kahramanmaraş, Türkiye Earthquake Doublet With Finite Element Analyses. Geophysical Research Letters, Vol. 51(16), e2023GL107472, https://doi.org/10.1029/2023GL107472.

Wang, Z., Zhang, W., Taymaz, T., He, Z., Xu, T., Zhang, Z. (2023). Dynamic Rupture

Process of the 2023 Mw 7.8 Kahramanmaraş Earthquake (SE Türkiye): Variable Rupture Speed and Implications for Seismic Hazard. Geophysical Research Letters, Vol. 50(15), e2023GL104787. https://doi.org/10.1029/2023GL104787.

Wu, F., Xie, J.J., An, Z., Lyu, C.H., Taymaz, T., Irmak, T.S., Li, X.J., Wen, Z.P., Zhou, B.F. (2023). Pulse-Like Ground Motion Observed During the 6 February 2023 Mw 7.8 Pazarcık Earthquake (Kahramanmaraş, SE Türkiye). Earthquake Science, Vol. 36(4), 328-339, https://doi.org/10.1016/j.eqs.2023.05.005

Xu, C., Zhang, Y., Hua, S., Zhang, X., Xu, L., Chen, Y., Taymaz, T. (2023). Rapid Source Inversions of the 2023 SE Türkiye Earthquakes with Teleseismic and Strong-Motion Data, Earthquake Science Vol. 36(4), 316–327, https://doi.org/10.1016/j.eqs.2023.05.004.

Yolsal-Çevikbilen, S., Taymaz, T., Irmak, T.S., Erman, C., Kahraman, M., Özkan, B., Eken, T., Öcalan, T., Doğan, A.H., Altuntaş, C. (2024). Source Geometry and Rupture Characteristics of the 20 February 2023 Mw 6.4 Hatay (Türkiye) Earthquake at Southwest Edge of the East Anatolian Fault. Geochemistry, Geophysics, Geosystems, Vol. 25(10), e2023GC011353, https://doi.org/10.1029/2023GC011353.

Zhang, Y., Tang, X., Liu, D., Taymaz, T., Eken, T., Guo, R., Zheng, Y., Wang, J., Sun, H. (2023). Geometric Controls on Cascading Rupture of the 2023 Kahramanmaraş Earthquake Doublet. NATURE Geoscience, Vol. 16, 1054-1060(2023), NGS-2023-04-00727. https://doi.org/10.1038/s41561-023-01283-3.

Zhou, J., Xu, Y., Zhang, Y., Feng, W., Taymaz, T., Chen, Y-T., Xu, C., Xu, B., Wang, R., Shi, F., Shao, Z., Huang, Q. (2025). Geometric Barriers Impacted Rupture Processes and Stress Releases of the 2023 Kahramanmaraş, Türkiye, Earthquake Doublet. Communications Earth & Environment - Nature, Vol. 6, Manuscript No. 56(2025), https://doi.org/10.1038/s43247-025-02004-x.

When discussing about the historical earthquakes the most prominent and established publications are provided by Prof.Dr. Nicholas N. Ambraseys' and yet again lack of trained seismologsst among the research group led the authors with a falsified wrong citaitons!

**Reply:** Thanks! Thank you for your correction. We have revised the relevant references. (Lines 108-109, 348-349, 356)

(7) Geological background is summarized with too limited and selected publications thus it does not necessarily provide the objective layout.

**Reply:** Thanks! We have enriched the geological background and supplemented the references. (Lines 108-109)

(8) Sampling and analytical methods should be developed further to detail the routinley collected samples and thier analysis to guide novice young scientists.

**Reply:** Thanks! Compared with the initial draft, we have added precise GPS coordinates, the chemical pre-treatment process, the selection of instruments and standards, as well as the analysis errors. (Lines 117-139). Specifically includes:

"16 samples of groundwater were collected in EAFZ, including hot springs, geothermal wells and river water. HS01-HS04 was collected from west to east along SF. HS07-HS16 was collected from north to south along EAFZ (Fig. 1). Detailed sample collection and testing methods can be found at Luo et al. (2023). In short, the water sample was taken with a 50 mL clean polyethylene bottle and the temperature and pH of the water were measured and recorded. Two samples were collected at each sampling site, one was added with ultrapure HNO3 to analyse the cation content, and the other was used to analyse the anion content and isotopic composition. All samples need to be pre-treated with a 0.45 µm filter membrane to remove impurities before sampling.

The Hydrogen and oxygen isotopes were determined by a Picarro L2140-I Liquid water and vapor isotope analyzer (relative to Vienna Standard Mean Ocean Water (V - SMOW)). Precisions on the measured δ18O and δD value was ±0.2% (2SD) and ±1% (2SD) respectively (Zeng et al., 2025). The cation (Li+, Na+, K+, Ca2+and Mg2+) and anion (F-, Cl-, NO3- and SO42-) were analysed by Dionex ICS-900 ion chromatograph (Thermo Fisher Scientific Inc.) at the Earthquake Forecasting Key Laboratory of China Earthquake Administration, with the reproducibility within ±2% and detection limits 0.01 mg/L (Chen et al., 2015). HCO3- and CO32- was determined by acid-base titration with a ZDJ-100 potentiometric titrator (reproducibility within ±2%). SiO2 were analysed by inductively coupled plasma emission spectrometer Optima-5300 DV (PerkinElmer Inc.) (Li et al. 2021). Trace elements were analysed by Element XR ICP-MS at the Test Center of the Research Institute of Uranium Geology. Multielement standard solutions (IV-ICPMS 71A, IV-ICP-MS 71B and IV-ICP-MS 71D, iNORGANIC

VENTURES) used for quality control. The analytical error margin of major cations and trace elements were less than 10%. Strontium isotope ratios (87Sr/86Sr) were determined through triple quadrupole ICP-MS (Agilent 8900 ICP-QQQ) with a precision of ±0.001 (Liu et al., 2020)."

(9) Results adequately discuss the hhysical, chemical and isotopic compositions of geothermal water acquired are listed in Table 1.

**Reply:** Thanks!**

(10) Discussion on (a) the origin of geothermal fluids and (b) water-rocks interaction are at a satisfactory level with proper citaitons.

**Reply:** Thanks!**

(11) Figure 8 at page 24 is fantastic and it should be improved using thick red lines on the EAFZ and the Sürgü Fault to bring them front a bit to attentions of the readers. If possible a few geographical location of towns and major cities can be labelled. Here with respect to Figure 8, some of well established seismic tomography studies and maps should be cited/referred to display the link on geothermal fields and the deeper structures (see following articles below).

**Reply:** Thanks! Thank you very much for your excellent suggestion. In the latest submitted manuscript, the model diagram is Fig. 9. According to your suggestion, we have **highlighted** and **bolded** the two faults, and marked the locations of the important nodes cities (Lines 369-375). In the main text, we have added references to the seismic tomography studies and map literature of the research area. (Lines 348-349)

Confal, J.M., Taymaz, T., Eken, T., Bezada, M.J., Faccenda, M. (2025). Remnant Tethyan Slab Fragments Beneath Northern Türkiye. EPSL – Earth and Planetary Science Letters, Vol. 664, 119458, https://doi.org/10.1016/j.epsl.2025.119458.

Confal, J. M., Bezada, M. J., Eken, T., Faccenda, M., Saygin, E., Taymaz, T. (2020).

Influence of Upper Mantle Anisotropy on Isotropic P-wave Tomography Images

Obtained in the Eastern Mediterranean Region, Journal of Geophysical Research (JGR)

– Solid Earth, Vol.: 125(8), https://doi.org/10.1029/2019JB018559.

Confal, J., Faccenda, M., Eken, T., and Taymaz, T. (2018). Numerical Simulation of 3-D Mantle Flow Evolution in Subduction Zone Environments in Relation to Seismic Anisotropy Beneath the Eastern Mediterranean Region, Earth and Planetary Science Letters (EPSL), Vol. 497, 50–61. https://doi.org/10.1016/j.epsl.2018.06.005.

Erman, C., Yolsal-Çevikbilen, S., Eken, T., Huang, Z., Taymaz, T. (2025). Seismic Anisotropy Variations in the Eastern Mediterranean Sea Region Revealed by Splitting Intensity Tomography: Implications on Mantle Dynamics. Journal of Geophysical Research (JGR) – Solid Earth, Vol. 130(3), e2024JB030331, https://doi.org/10.1029/2024JB030331.

Fichtner, A., Saygin, E., Taymaz, T., Cupillard, P., Capdevillee, Y. and Trampert, J. (2013). The Deep Structure of the North Anatolian Fault Zone, Earth and Planetary Science Letters, July 2013, Vol. 373, pp. 109-117, doi:10.1016/j.epsl.2013.04.027.

Kind, R., Eken, T., Tilmann, F., Sodoudi, F., Taymaz, T., Bulut, F., Yuan, X., Can, B. and Schneider, F., (2015). Thickness of the Lithosphere Beneath Turkey and Surroundings from S-Receiver Functions, Solid Earth, 6, 971–984. www.solid-earth.net/6/971/2015, doi:10.5194/se-6-971-2015.

Wang, H., Huang, Z., Eken, E., Keleş, D., Kaya-Eken, T., Confal, J. M., Erman, C., Yolsal-Çevikbilen, S., Zhao, D., Taymaz, T. (2020). Isotropic and Anisotropic P-wave Velocity Structures of the Crust and Uppermost Mantle Beneath Turkey, Journal of Geophysical Research (JGR) — Solid Earth, Vol. 125 (12), e2020JB019566, https://doi.org/10.1029/2020JB019566.

(12) The relationship between geothermal fluid and earthquake forecasting section is attractive one, but it needs to be supported by other local and/or global examples. Thus, a refined discussion is needed.

**Reply:** Thanks! We clarify two key points:

- 1. Regional specificity: **Anhydrite's applicability is context-dependent** (e.g., evaporite-rich EAFZ).
- 2. Paradigm shift: The innovation is methodological identifying region-specific target horizons (e.g., anhydrite here, serpentine elsewhere) for localized monitoring. We proposed that traditional tracers (He, Rn) and anhydrite serve complementary roles; multi-proxy approaches are essential for earthquake forecasting.
- (13) Conclusion and Outlook is well placed with proper setting of the conducted systematic element and isotope analysis on the hydrogeochemistry of geothermal fluid

after the earthquake.

**Reply:** Thanks!

In SUMMARY, the manuscript HESS-2024-395 titled "Gypsum as a potential tracer of earthquake: a case study of the Mw7.8 earthquake in the East Anatolian Fault Zone, southeastern Turkey" is timely, interesting and fills the gap over the 2023 Kahramanmaraş Earthquake Doublet and it should be published after minor-moderate revisions.

9 August 2025

Tuncay Taymaz

Istanbul Technical University - Istanbul - Türkiye

Reply: Thanks!